# Fair Streaming Principal Component Analysis: Statistical and Algorithmic Viewpoint

**Junghyun Lee**[*]   **Hanseul Cho**[*]   **Se-Young Yun**   **Chulhee Yun**
Kim Jaechul Graduate School of AI, KAIST
{jh_lee00, jhs4015, yunseyoung, chulhee.yun}@kaist.ac.kr

## Abstract

Fair Principal Component Analysis (PCA) is a problem setting where we aim to perform PCA while making the resulting representation fair in that the projected distributions, conditional on the sensitive attributes, match one another. However, existing approaches to fair PCA have two main problems: theoretically, there has been no statistical foundation of fair PCA in terms of learnability; practically, limited memory prevents us from using existing approaches, as they explicitly rely on full access to the entire data. On the theoretical side, we rigorously formulate fair PCA using a new notion called *probably approximately fair and optimal* (PAFO) learnability. On the practical side, motivated by recent advances in streaming algorithms for addressing memory limitation, we propose a new setting called *fair streaming PCA* along with a memory-efficient algorithm, fair noisy power method (FNPM). We then provide its *statistical* guarantee in terms of PAFO-learnability, which is the first of its kind in fair PCA literature. Lastly, we verify the efficacy and memory efficiency of our algorithm on real-world datasets.

## 1   Introduction

Algorithmic fairness ensures that machine learning algorithms do not propagate nor exacerbate bias, which may lead to discriminatory decision-making (Barocas and Selbst, 2016) and thus has been a very active area of research. This has direct implications in our everyday life, including but not limited to criminal justice (Kirchner et al., 2016), education (Kizilcec and Lee, 2021), and more. See Mehrabi et al. (2021) for a comprehensive survey of bias and fairness in machine learning.

Often, one needs to consider fairness for a large number of high-dimensional data points. One of the standard tools for dealing with such high-dimensional data is PCA (Hotelling, 1933; Pearson, 1901), a classical yet still one of the most popular algorithms for performing interpretable dimensionality reduction. It has been adapted as a baseline and/or standard tool in exploratory data analysis, whose application ranges from natural sciences, engineering (Abdi and Williams, 2010; Jolliffe and Cadima, 2016), and even explainable AI (Li et al., 2023; Tjoa and Guan, 2021). Due to its ubiquity and wide applicability, several works study defining fairness in PCA and developing a fair variant of it. A recent line of research (Kleindessner et al., 2023; Lee et al., 2022; Olfat and Aswani, 2019) defines PCA fairness in the context of fair representation (Zemel et al., 2013) in that the projected group conditional distributions should match.

However, existing fair PCA approaches suffer from two problems. Theoretically, they provide no statistical foundation of fair PCA or guarantees for their algorithms. By statistical foundation, we mean the usual PAC-learnability (Shalev-Schwartz and Ben-David, 2014) guarantees, e.g., sample complexity for ensuring optimality in explained variance and fairness constraint with high probability.

---

[*]Equal contributions

37th Conference on Neural Information Processing Systems (NeurIPS 2023).

On top of that, the second problem arises from a practical viewpoint: memory limitation. All the aforementioned fair PCA algorithms assume that the learner can store the entire data points and incurs memory complexity of order at least $\mathcal{O}(d \max(N, d))$, where $d$ is the dimensionality of the data and $N$ is the number of data points. As memory limitation is often a critical bottleneck in deploying machine learning algorithms (Mitliagkas et al., 2013), as much as fairness is important, it is also paramount that imposing fairness to PCA does not incur too much memory overhead. A popular approach to mitigate such memory limitation for PCA is to consider the one-pass, streaming setting. In this setting, each data point is revealed to the learner sequentially, each point is irretrievably gone unless she explicitly stores it, and she can use only $\mathcal{O}(dk)$ memory, with $k$ being the target dimension of projection. Indeed, without the fairness constraint, streaming PCA has been studied extensively; see Balzano et al. (2018) and references therein.

In this work, we address both problems in a principled manner. Our contributions are as follows:

- We provide an alternative formulation of fair PCA based on the "Null It Out" approach (Section 3). Based on the new formulation, we introduce the concept of *probably approximately fair and optimal* (PAFO)-learnability to formalize the problem of fair PCA (Section 4).
- To address the memory limitation, we propose a new problem setting called *fair streaming PCA*, as well as *fair noisy power method* (FNPM), a simple yet memory-efficient algorithm based on the noisy power method. (Section 5). We note that our algorithm incurs a much lower memory complexity even compared to the most efficient variant of fair PCA proposed by Kleindessner et al. (2023).
- We then prove that our algorithm achieves the PAFO-learnability for fair streaming PCA (Section 6). Such statistical guarantee is the first of its kind in fair PCA literature.
- Lastly, we empirically validate our algorithm on CelebA and UCI datasets. Notably, we run FNPM on the original *full-resolution* CelebA dataset on which existing fair PCA algorithms fail due to high memory requirements. It shows turning such a non-streaming setting into a streaming setting and applying our algorithm allows one to bypass the memory limitation (Section 7).

## 2 Preliminaries

**Notation.** For $\ell \geq 1$, let $\boldsymbol{I}_\ell$ be the identity matrix of size $\ell \times \ell$. For $k < d$, we bring the *Stiefel manifold* $\mathrm{St}(d, k) = \{\boldsymbol{A} \in \mathbb{R}^{d \times k} : \boldsymbol{A}^\intercal \boldsymbol{A} = \boldsymbol{I}_k\}$, which is the collection of all rank-$k$ semi-orthogonal matrices. We denote an orthonormal column basis of a (full column rank) matrix $\boldsymbol{M} \in \mathbb{R}^{d \times k}$ obtained by QR decomposition as $\mathrm{QR}(\boldsymbol{M}) \in \mathrm{St}(d, k)$ and denote its column space by $\mathrm{col}(\boldsymbol{M})$. Also, for $\boldsymbol{A} \in \mathrm{St}(d, k)$, we denote the orthogonal projection matrix to $\mathrm{col}(\boldsymbol{A})^\perp = \mathrm{null}(\boldsymbol{A}^\intercal)$ as $\boldsymbol{\Pi}_{\boldsymbol{A}}^\perp = \boldsymbol{I}_d - \boldsymbol{A}\boldsymbol{A}^\intercal = \boldsymbol{I}_d - \boldsymbol{\Pi}_{\boldsymbol{A}}$. Moreover, we denote the collection of all possible $d$-dimensional probability distributions as $\mathcal{P}_d$. For a zero-mean random matrix $\boldsymbol{Z}$, its (scalar-valued) variance is defined as $\mathrm{Var}(\boldsymbol{Z}) = \max\left(\|\mathbb{E}\left[\boldsymbol{Z}\boldsymbol{Z}^\intercal\right]\|_2, \|\mathbb{E}\left[\boldsymbol{Z}^\intercal\boldsymbol{Z}\right]\|_2\right)$. In general, $\mathrm{Var}(\boldsymbol{Z}) = \mathrm{Var}(\boldsymbol{Z} - \mathbb{E}[\boldsymbol{Z}])$. Lastly, we use the usual $\mathcal{O}$, $\Omega$, and $\Theta$ notations for asymptotic analyses, where tildes ($\tilde{\mathcal{O}}$, $\tilde{\Omega}$, and $\tilde{\Theta}$, resp.) are used for hiding logarithmic factors.

**Setup.** Assume that the sensitive attribute variable, which we will be imposing fairness, is binary, denoted by $a \in \{0, 1\}$. For each group $a$, let $\mathcal{D}_a$ be a $d$-dimensional distribution of mean $\boldsymbol{\mu}_a$ and covariance $\boldsymbol{\Sigma}_a$, both of which are assumed to be well-defined. We often call them group-conditional mean and covariance, respectively. With a fixed, *unknown* mixture parameter $p \in (0, 1)$, let us denote the total data distribution as $\mathcal{D} := p\mathcal{D}_0 + (1 - p)\mathcal{D}_1$. Equivalently, the sensitive attribute follows $a \sim \mathrm{Bernoulli}(p)$, and the conditional random variable $\boldsymbol{x}|a$ is sampled from $\mathcal{D}_a$. In that case, $\boldsymbol{\mu}_a = \mathbb{E}[\boldsymbol{x}|a]$ and $\boldsymbol{\Sigma}_a = \mathbb{E}[\boldsymbol{x}\boldsymbol{x}^\intercal|a] - \boldsymbol{\mu}_a\boldsymbol{\mu}_a^\intercal$. We often write $p_0 = 1 - p$ and $p_1 = p$ for brevity. We also define the mean difference $\boldsymbol{f} := \boldsymbol{\mu}_1 - \boldsymbol{\mu}_0$ and the second moment difference $\boldsymbol{S} := \mathbb{E}[\boldsymbol{x}\boldsymbol{x}^\intercal|a = 1] - \mathbb{E}[\boldsymbol{x}\boldsymbol{x}^\intercal|a = 0] = \boldsymbol{\Sigma}_1 - \boldsymbol{\Sigma}_0 + \boldsymbol{\mu}_1\boldsymbol{\mu}_1^\intercal - \boldsymbol{\mu}_0\boldsymbol{\mu}_0^\intercal$. Accordingly, denote the true mean and covariance of $\mathcal{D}$ as $\boldsymbol{\mu}$ and $\boldsymbol{\Sigma}$, respectively. For simplicity, let us assume that $\mathcal{D}$ is centered, i.e., $\boldsymbol{\mu} = \boldsymbol{0}$; note that this does *not* mean that the group conditional distributions $\mathcal{D}_a$'s are centered.

**PCA.** In the *offline* setting, the full covariance matrix $\boldsymbol{\Sigma}$ is given which is often a sample covariance matrix $\frac{1}{n}\sum_{i=1}^n \boldsymbol{x}_i\boldsymbol{x}_i^\top$ for $n$ data points $\boldsymbol{x}_1, \ldots, \boldsymbol{x}_n$. The goal of vanilla (offline) PCA (Hotelling, 1933; Pearson, 1901) is to compute the loading matrix $\boldsymbol{V} \in \mathbb{R}^{d \times k}$ that preserves as much variance as possible after projecting $\boldsymbol{\Sigma}$ via $\boldsymbol{V}$, *i.e.*, maximize $\mathrm{tr}(\boldsymbol{\Pi}_{\boldsymbol{V}}\boldsymbol{\Sigma}) = \mathrm{tr}(\boldsymbol{V}^\intercal\boldsymbol{\Sigma}\boldsymbol{V})$. Here, $k < d$ is the

target dimension to which the data's dimensionality $d$ is to be reduced and is chosen by the learner. We additionally consider the constraint $\boldsymbol{V} \in \mathrm{St}(d, k)$ to ensure that the resulting coordinate after the transformation is orthogonal and thus amenable to various statistical interpretations (Johnson and Wichern, 2008). Without any fairness constraint, Eckart-Young theorem (Eckart and Young, 1936) implies that the solution is characterized as a matrix whose columns are the top-$k$ eigenvectors of $\boldsymbol{\Sigma}$.

**Fair PCA.**   Recently, it has been suggested that performing vanilla PCA on real-world datasets may exhibit bias, making the final outputted projection "unfair". As is often the case, there can be multiple definitions of fairness in PCA, but the following two are the most popular: equalizing reconstruction losses (Kamani et al., 2022; Samadi et al., 2018; Tantipongpipat et al., 2019; Vu et al., 2022), or equalizing the projected distributions (Kleindessner et al., 2023; Lee et al., 2022; Olfat and Aswani, 2019) from the perspective of fair representation (Zemel et al., 2013); we focus on the latter one.

# 3 An Alternative Approach to Fair PCA

## 3.1 "Null It Out" Formulation of Fair PCA

In this work, we consider fair PCA as learning fair representation (Zemel et al., 2013). The goal is to preserve as much variance as possible while obfuscating any information regarding the sensitive attribute. To this end, we take the "Null It Out" approach as proposed in Ravfogel et al. (2020). Intuitively, we want to nullify the directions in which the sensitive attribute $a$ can be inferred, and in this work, we consider two such **unfair directions**: mean difference $\boldsymbol{f}$ and *eigenvectors* of second moment difference $\boldsymbol{S}$. To give the learner flexibility in choosing the trade-off between fairness and performance (measured in explained variance), let $m \geq 1$ be the number of top eigenvectors of $\boldsymbol{S}$ to nullify. Thus, the learner is nullifying at most $(m+1)$-dimensional subspace that is unfair with respect to $a$, which we refer to as the **unfair subspace**. Precisely, we formulate our fair PCA as follows:

$$\max_{\boldsymbol{V} \in \mathrm{St}(d,k)} \mathrm{tr}(\boldsymbol{V}^{\mathsf{T}} \boldsymbol{\Sigma} \boldsymbol{V}), \quad \text{subject to } \mathrm{col}(\boldsymbol{V}) \subset \mathrm{col}([\boldsymbol{P}_m | \boldsymbol{f}])^{\perp}, \tag{1}$$

where $d$ is the data dimensionality, $k$ is the target dimension, and the columns of $\boldsymbol{P}_m \in \mathrm{St}(d, m)$ is top-$m$ orthonormal eigenvectors of $\boldsymbol{S}$.

## 3.2 An Explicit Characterization for Solution of Fair PCA

To first construct the unfair subspace that is spanned by $\boldsymbol{f}$ as well as $\boldsymbol{P}_m$, let us define $\boldsymbol{U} \in \mathrm{St}(d, m')$ to be the orthogonal matrix whose columns form a basis of $\mathrm{col}([\boldsymbol{P}_m | \boldsymbol{f}])$. Then, $\boldsymbol{U}$ has a closed form as follows: $m' = m$ if $\boldsymbol{f} \in \mathrm{col}(\boldsymbol{P}_m)$ and $m' = m + 1$ otherwise, and

$$\boldsymbol{U} = \begin{cases} \boldsymbol{P}_m, & \text{if } \boldsymbol{f} \in \mathrm{col}(\boldsymbol{P}_m), \\ \mathtt{QR}([\boldsymbol{P}_m | \boldsymbol{f}]) = \left[ \boldsymbol{P}_m \,\middle|\, \frac{\boldsymbol{g}}{\|\boldsymbol{g}\|_2} \right], & \text{otherwise,} \end{cases} \tag{2}$$

where $\boldsymbol{g} = \Pi_{\boldsymbol{P}_m}^{\perp} \boldsymbol{f} \in \mathrm{col}(\boldsymbol{P}_m)^{\perp}$. Note that $\boldsymbol{g}$ is a vector in a direction that $\boldsymbol{f}$ is projected onto $\mathrm{col}(\boldsymbol{P}_m)^{\perp} = \mathrm{null}(\boldsymbol{P}_m^{\mathsf{T}})$. For this $\boldsymbol{U}$, our constraint in (1) can be interpreted as an equivalent nullity constraint $\Pi_{\boldsymbol{U}} \boldsymbol{V} = \boldsymbol{0}$:

$$\max_{\boldsymbol{V} \in \mathrm{St}(d,k)} \mathrm{tr}(\boldsymbol{V}^{\mathsf{T}} \boldsymbol{\Sigma} \boldsymbol{V}), \quad \text{subject to } \Pi_{\boldsymbol{U}} \boldsymbol{V} = \boldsymbol{0}. \tag{3}$$

The above is equivalent to the following problem without any constraint other than semi-orthogonality:

$$\max_{\boldsymbol{V} \in \mathrm{St}(d,k)} \mathrm{tr}\left(\boldsymbol{V}^{\mathsf{T}} \Pi_{\boldsymbol{U}}^{\perp} \boldsymbol{\Sigma} \Pi_{\boldsymbol{U}}^{\perp} \boldsymbol{V}\right), \tag{4}$$

which is basically the vanilla $k$-PCA problem of a matrix $\Pi_{\boldsymbol{U}}^{\perp} \boldsymbol{\Sigma} \Pi_{\boldsymbol{U}}^{\perp}$. Therefore, a top-$k$ orthonormal column basis of this matrix is indeed a solution of our problem (4).

## 3.3 Comparison to the Existing Covariance Matching Constraint

Previous works on fair PCA (Kleindessner et al., 2023; Olfat and Aswani, 2019) consider an exact covariance-matching constraint (namely, $\boldsymbol{V}^{\mathsf{T}}(\boldsymbol{\Sigma}_1 - \boldsymbol{\Sigma}_0)\boldsymbol{V} = \boldsymbol{0}$). In fact, this is equivalent to the condition $\boldsymbol{V}^{\mathsf{T}} \boldsymbol{S} \boldsymbol{V} = \boldsymbol{0}$ under the mean-matching constraint $\boldsymbol{f}^{\mathsf{T}} \boldsymbol{V} = \boldsymbol{0}$, which can be derived as

$$\begin{aligned}
\boldsymbol{V}^{\mathsf{T}}(\boldsymbol{\Sigma}_1 - \boldsymbol{\Sigma}_0)\boldsymbol{V} &= \boldsymbol{V}^{\mathsf{T}}\left(\mathbb{E}[\boldsymbol{x}\boldsymbol{x}^{\mathsf{T}}|a=1] - \mathbb{E}[\boldsymbol{x}\boldsymbol{x}^{\mathsf{T}}|a=0]\right)\boldsymbol{V} - \boldsymbol{V}^{\mathsf{T}}\left(\boldsymbol{\mu}_1\boldsymbol{\mu}_1^{\mathsf{T}} - \boldsymbol{\mu}_0\boldsymbol{\mu}_0^{\mathsf{T}}\right)\boldsymbol{V} \\
&= \boldsymbol{V}^{\mathsf{T}}\boldsymbol{S}\boldsymbol{V} - \boldsymbol{V}^{\mathsf{T}}\left(\boldsymbol{f}\boldsymbol{\mu}_1^{\mathsf{T}} + \boldsymbol{\mu}_0\boldsymbol{f}^{\mathsf{T}}\right)\boldsymbol{V} = \boldsymbol{V}^{\mathsf{T}}\boldsymbol{S}\boldsymbol{V} = \boldsymbol{0}.
\end{aligned}$$

One immediate problem with this is that the constraint may be infeasible depending on the choice of $\boldsymbol{\Sigma}_0, \boldsymbol{\Sigma}_1$, or $\boldsymbol{S}$; e.g., when $\boldsymbol{\Sigma}_1 - \boldsymbol{\Sigma}_0$ is positive definite. For this reason, Kleindessner et al. (2023); Olfat and Aswani (2019) propose relaxations of the fairness constraints but provide no further discussions on its impact on statistical guarantees. On the contrary, our formulation is always feasible without any need for relaxation, allowing us to consider a rigorous definition of fair PCA (Definition 2) for the first time in fair PCA literature.

## 4  Statistical Viewpoint: PAFO-Learnability of PCA

As all the distribution statistics $(\boldsymbol{\Sigma}, p, \cdots)$ are unknown, the learner, given some finite number of samples, must learn all of them *and* solve fair PCA. In supervised learning, such a problem is often formalized in a PAC-learnability framework (Shalev-Schwartz and Ben-David, 2014). In the context of PAC-learnability for unsupervised learning settings, TV-learning, which is the task of learning distribution, has been mainly considered so far (Ananthakrishnan et al., 2021; Hopkins et al., 2023). However, unlike TV-learning, it is unnecessary to learn the whole distribution in fair PCA; moreover, fair PCA has the fairness constraint $\boldsymbol{\Pi}_U\boldsymbol{V} = \boldsymbol{0}$ to be satisfied. Inspired by the unsupervised PAC-learnability as well as constrained PAC-learnability (Chamon and Ribeiro, 2020), we propose a new notion of learnability for fair PCA, called *PAFO (Probably Approximately Fair and Optimal) learnability*, as follows:

**Definition 1** (Projection Learner). *A **projection learner** is a function that takes $k \geq 1$ and $d$-dimensional samples as input and outputs a loading matrix $\boldsymbol{V} \in \mathrm{St}(d, k)$.*

**Definition 2** (PAFO-Learnability of PCA). *Let $d, k, m$ be integers such that $1 \leq k < d$ and $m < d$. We say that $\mathcal{F}_d \subset \mathcal{P}_d \times \mathcal{P}_d \times (0, 1)$ is **PAFO-learnable for PCA** if there exists a function $N_{\mathcal{F}_d} : (0, 1)^3 \to \mathbb{N}$ and a projection learner $\mathcal{A}$ satisfying the following:*

*For every $\varepsilon_{\mathrm{o}}, \varepsilon_{\mathrm{f}}, \delta \in (0, 1)$ and $(\mathcal{D}_0, \mathcal{D}_1, p) \in \mathcal{F}_d$, when running $\mathcal{A}$ on $N \geq N_{\mathcal{F}_d}(\varepsilon_{\mathrm{o}}, \varepsilon_{\mathrm{f}}, \delta)$ i.i.d. samples from $\mathcal{D} := p\mathcal{D}_1 + (1-p)\mathcal{D}_0$ of the form $(a, \boldsymbol{x})$, $\mathcal{A}$ returns $\boldsymbol{V}$ s.t., with probability at least $1 - \delta$ (over the draws of the $N$ samples),*

$$\operatorname{tr}\left(\boldsymbol{V}^{\mathsf{T}}\boldsymbol{\Sigma}\boldsymbol{V}\right) \geq \operatorname{tr}\left(\boldsymbol{V}_{\star}^{\mathsf{T}}\boldsymbol{\Sigma}\boldsymbol{V}_{\star}\right) - \varepsilon_{\mathrm{o}}, \quad \|\boldsymbol{\Pi}_U\boldsymbol{V}\|_2 \leq \varepsilon_{\mathrm{f}}, \tag{5}$$

*where $\boldsymbol{U}$ is as defined in Eqn. (2) and $\boldsymbol{V}_{\star}$ is any solution to Eqn. (4) (with prescribed $k$ and $m$).*

Like in the usual PAC-learnability, $N_{\mathcal{F}_d}$ is referred to as the *sample complexity* of fair PCA. Observe how the optimality is measured w.r.t. the optimal solution of *fair* PCA, not the vanilla PCA. Also, the two conditions are not overlapping: vanilla PCA (overly) satisfies $\varepsilon_{\mathrm{o}}$-optimality in explained variance but does not satisfy $\varepsilon_{\mathrm{f}}$-optimality in fairness, and vice versa for a poorly chosen $\boldsymbol{V} \in St(d, k)$ with $\operatorname{col}(\boldsymbol{V}) \subseteq \operatorname{col}([\boldsymbol{P}_m|\boldsymbol{f}])^{\perp}$.

## 5  Algorithmic Viewpoint: Fair Streaming PCA

We now introduce a new problem setting, *fair streaming PCA*. In this setting, the learner receives a stream of pairs $(a_t, \boldsymbol{x}_t) \in \{0, 1\} \times \mathbb{R}^d$ sequentially. Note that the sensitive attribute information $a_t$ is also available at each time-step; this is commonly assumed when considering fairness in streaming setting (Bera et al., 2022; El Halabi et al., 2020). Precisely, we assume the following model of the data generation process: at each time-step $t$, a sensitive attribute is chosen as $a_t \sim \mathrm{Bernoulli}(p)$, then the data is sampled from the corresponding sensitive group's conditional distribution $\boldsymbol{x}_t \mid a_t \sim \mathcal{D}_{s_t}$. Importantly, as done in previous streaming PCA literature (Mitliagkas et al., 2013), we assume that the learner has only $\mathcal{O}(dk)$ memory, where $d$ is the data dimension and $k$ is the target dimension. We can formally define the PAFO-learnability in this streaming setting:

**Definition 3.** *We say that $\mathcal{F}_d \subseteq \mathcal{P}_d \times \mathcal{P}_d \times (0, 1)$ is **PAFO-learnable for streaming PCA** if the projection learner $\mathcal{A}$ for which Definition 2 holds uses only $\mathcal{O}(dk)$ memory for streaming data.*

### 5.1 Our Algorithm: Fair Noisy Power Method (FNPM)

One only needs to estimate $U$ to use the off-the-shelf streaming PCA algorithm. As $U$ is of size $d \times m$, storing its estimate is no problem for the memory constraint as long as $m = \mathcal{O}(k)$. Naturally, we proceed via a two-stage approach; first, estimate $U$ sufficiently well, then with the fixed estimate of $U$, apply the noisy power method (Hardt and Price, 2014; Mitliagkas et al., 2013) for $V$.

For estimating $U$, one needs to estimate $f$ and $P_m$. Estimating $f$ can be done using the usual cumulative averaging. As for $P_m$, we can consider the two main approaches for streaming PCA: Oja's method (Huang et al., 2021; Oja, 1982; Oja and Karhunen, 1985) and noisy power method (NPM) (Hardt and Price, 2014; Mitliagkas et al., 2013). We first show that Oja's method is *inapplicable* for our purpose, as it may ignore some eigenvectors corresponding to negative (but large in magnitude) eigenvalues of $S$. For instance, if $S = -2e_1e_1^\intercal + e_2e_2^\intercal + 4e_3e_3^\intercal$ with $e_i$ being the standard basis vectors, then Oja's method with $m = 2$ would yield $[e_2|e_3]$ when we actually want $[e_1|e_3]$. For the same reason, simply shifting the eigenvalue spectrum by considering $S + \|S\|_2 I$ does not work. Thus we apply NPM for estimating $P_m$ in our case, which is known to converge as long as the singular value gap of $P_m$ is large enough and norms of the noise matrices at each iterate are properly bounded (Balcan et al., 2016; Hardt and Price, 2014).

---

**Algorithm 1:** `UnfairSubspace`

1 **Input:** $m$, Block size $b$, Number of iterations $T$;
2 **Output:** A matrix $\widehat{U}$ with orthonormal columns;
3 $W_0 = \mathtt{QR}(\mathcal{N}(0,1)^{d \times m})$;
4 $(\overline{m}^{(0)}, \overline{m}^{(1)}, B^{(0)}, B^{(1)}) = (0_d, 0_d, 0, 0)$;
5 **for** $t \in [T]$ **do**
6      Receive $\{(a_i, x_i)\}_{i=(t-1)b+1}^{tb}$;
7      **foreach** $a \in \{0,1\}$ **do**
8          Compute $b_t^{(a)}, m_t^{(a)}, C_t^{(a)}$ as Eqn. (6);
9          $\overline{m}^{(a)} \leftarrow \frac{B^{(a)}}{B^{(a)}+b_t^{(a)}}\overline{m}^{(a)} + \frac{b_t^{(a)}}{B^{(a)}+b_t^{(a)}}m_t^{(a)}$;
10          $B^{(a)} \leftarrow B^{(a)} + b_t^{(a)}$;
11      $W_t = \mathtt{QR}\left(C_t^{(1)} - C_t^{(0)}\right)$;
12 $\widehat{f} \leftarrow \overline{m}^{(1)} - \overline{m}^{(0)}$;
13 $\widehat{g} \leftarrow \widehat{f} - W_T W_T^\intercal \widehat{f}$;
14 **if** $\|\widehat{g}\|_2 = 0$ **then**
15      $\widehat{U} = W_T$
16 **else**
17      $\widehat{U} = \left[W_T \mid \frac{\widehat{g}}{\|\widehat{g}\|_2}\right]$
18 **return** $\widehat{U}$

---

**Algorithm 2:** Fair NPM

1 **Input:** $k$, Block sizes $\mathcal{B}, b$, Numbers of iterations $\mathcal{T}, T$;
2 **Output:** $V_{\mathcal{T}} \in \mathrm{St}(d,k)$;
3 $\widehat{U} \leftarrow \mathtt{UnfairSubspace}(b, T)$;
     // Algorithm 1
4 $V_0 \leftarrow \mathtt{QR}(\mathcal{N}(0,1)^{d \times k})$;
5 **for** $\tau \in [\mathcal{T}]$ **do**
6      $V_\tau \leftarrow V_{\tau-1} - \widehat{U}\widehat{U}^\intercal V_{\tau-1}$;
7      Receive $\{(*, \tilde{x}_j)\}_{j=(\tau-1)\mathcal{B}+1}^{\tau\mathcal{B}}$;
8      $V_\tau \leftarrow \frac{1}{\mathcal{B}}\sum_{j=1}^{\mathcal{B}} \tilde{x}_j \tilde{x}_j^\intercal V_\tau$;
9      $V_\tau \leftarrow \mathtt{QR}\left(V_\tau - \widehat{U}\widehat{U}^\intercal V_\tau\right)$;
10 **return** $V_{\mathcal{T}}$

---

**Description of the algorithms.** The pseudocode of our algorithm is shown in Algorithms 1 and 2. The goal of Algorithm 1 is to estimate $U = \left[P_m \mid \frac{g}{\|g\|_2}\right]$ in Eqn. (2) as accurately as possible. Lines 5–13 do the estimation of $P_m$ and $g = \Pi_{P_m}^\perp f$; line 11 is the NPM to find $P_m$, lines 12 and 13 are the estimation of $f$ and $g$ respectively, and line 17 is the concatenation of the estimates of $P_m$ and $g/\|g\|_2$. Especially at line 17, the algorithm determines whether to incorporate mean difference by checking $\hat{g}$, which can be proved to be correct, *i.e.*, $g = 0$ if and only if $\hat{g} = 0$ with high probability. With the estimated $U$ from Algorithm 1, Algorithm 2 performs the usual NPM on $\Pi_U^\perp \Sigma \Pi_U^\perp$, as in Eqn. (4). The memory complexity of Algorithm 2 is $\mathcal{O}(d \max(m,k))$, since we do not have to store all $b$ or $\mathcal{B}$ data points at each time, and all the operations used can be implemented in a manner that conforms to the memory limitation; the full pseudocodes are provided in Appendix B.

At time step $t$ of Algorithm 1, for each $a \in \{0,1\}$, $b_t^{(a)}$ is the number of data points $x_i$'s such that $a_i = a$; $m_t^{(a)}$ is the term used for estimation of the group-wise sample mean of $x_i$'s; $C_t^{(a)}$ is used for the group-wise sample second moment. Their forms are as follows and can be computed

incrementally in the streaming setup:

$$b_t^{(a)} = \sum_{i=b(t-1)+1}^{bt} \mathbb{1}_{[a_i=a]}, \quad \boldsymbol{m}_t^{(a)} = \sum_{i=b(t-1)+1}^{bt} \frac{\mathbb{1}_{[a_i=a]}}{b_t^{(a)}} \boldsymbol{x}_i, \quad \boldsymbol{C}_t^{(a)} = \sum_{i=b(t-1)+1}^{bt} \frac{\mathbb{1}_{[a_i=a]}}{b_t^{(a)}} \boldsymbol{x}_i \boldsymbol{x}_i^\mathsf{T} \boldsymbol{W}_{t-1},$$

(6)

where we set the last two quantities to $\boldsymbol{0}$ when $b_t^{(a)} = 0$.

Note that as $b_t^{(a)}$ itself is random, this presents some technical challenges in the proofs of the theoretical guarantees. For instance, the above estimators for the mean and covariance are biased. Still, by properly using peeling argument and matrix concentration inequalities as well as perturbation theories (Golub and Loan, 2013; Tropp, 2015), we could sufficiently bound their errors. Informally speaking, we show that line 11 corresponds to the noisy power method for the matrix $\boldsymbol{S}$, which incurs the memory complexity $\mathcal{O}(dm)$.

## 5.2 Previous Approaches are not Suitable for Streaming Setup

All the existing approaches to fair PCA require the full knowledge of $\boldsymbol{\mu}_1 - \boldsymbol{\mu}_0$ and $\boldsymbol{\Sigma}_1 - \boldsymbol{\Sigma}_0$, or even the full data matrix $\boldsymbol{X}$. Olfat and Aswani (2019) need $\boldsymbol{f}$ and $\boldsymbol{S}$ to formulate the convex matrix constraints for their semi-positive definite programming (SDP), which is then solved with commercial SDP solver; Lee et al. (2022) need $\boldsymbol{X}$ to compute the derivative of their maximum mean discrepancy (MMD) penalty term, which requires $\mathcal{O}(d^2)$ memory to compute kernel Gram matrices. One may hope that the PCA-type approach taken by Kleindessner et al. (2023) may be easily extendable to our streaming setting by just using the standard techniques (e.g., using matrix-vector products) from streaming PCA (Mitliagkas et al., 2013). Indeed, they also propose a similar relaxation of the covariance constraint, leading to a closed-form solution. However, their formulation is not memory-efficient and, more importantly, is not as applicable to streaming settings as our formulation; see Appendix D for more discussions.

## 5.3 Extension to Multiple Sensitive Groups/Attributes

So far, we have considered a single and binary-sensitive attribute for simplicity. Here, we briefly discuss extending our framework to multiple non-binary sensitive attributes. Suppose that there are $\ell$ different attributes (e.g., gender and race), and for each $r \in [\ell]$, the $r$-th sensitive attribute has $g_r$ groups (e.g., male, female, and non-binary: $g_{\text{gender}} = 3$ in this case). We first describe the new data generation process: at each time $t$, we obtain $((a_{1,t}, \cdots, a_{\ell,t}), \boldsymbol{x}_t)$, where for each $r$, $a_{r,t} \sim \text{Categorical}(p_{r,1}, \cdots, p_{r,g_r})$ with $\sum_{a=1}^{g_r} p_{r,a} = 1$. Then, our formulation of fair PCA easily extends via a one-versus-all comparison approach. That is to say, for each $r \in [\ell]$, define

$$\boldsymbol{f}_{r,a} = \mathbb{E}[\boldsymbol{x} \mid a_r = a] - \mathbb{E}[\boldsymbol{x} \mid a_r \neq a], \quad \boldsymbol{S}_{r,a} = \mathbb{E}[\boldsymbol{x}\boldsymbol{x}^\mathsf{T} \mid a_r = a] - \mathbb{E}[\boldsymbol{x}\boldsymbol{x}^\mathsf{T} \mid a_r \neq a].$$

Denote the top-$m_r$ column basis of eigenspace of $\boldsymbol{S}_{r,a}$ by $\boldsymbol{P}_{m_r}(\boldsymbol{S}_{r,a})$. Then our notion of fair PCA can be extended as follows: where $d, k, m_1, \cdots, m_n$ are given as $d > k + \ell + \sum_r m_r$,

$$\max_{\boldsymbol{V} \in \text{St}(d,k)} \text{tr}(\boldsymbol{V}^\mathsf{T} \boldsymbol{\Sigma} \boldsymbol{V}), \quad \text{subject to } \text{col}(\boldsymbol{V}) \subset \bigcap_{r \in [\ell]} \bigcap_{a \in [g_r]} \text{col}([\boldsymbol{P}_{m_r}(\boldsymbol{S}_{r,a}) \mid \boldsymbol{f}_{r,a}])^\perp. \quad (7)$$

Our algorithm can also be naturally adapted to this scenario via one-versus-all manner. In Appendix C, we provide the full pseudocode for the general case of multiple and non-binary sensitive attributes.

# 6 FNPM is a PAFO-Learning Algorithm

We now show that our proposed memory-efficient algorithm, FNPM, is actually a PAFO-learning algorithm in that with a certain sample complexity, it satisfies the definition of PAFO-learnability. The proofs of all the theoretical results stated here are deferred to Appendix E.

To use proper matrix concentration inequalities for our error term analysis, various streaming PCA literature adopt some assumption on the underlying data distribution, e.g., sub-Gaussianity (Bienstock et al., 2022; Jain et al., 2016; Yang et al., 2018). Throughout the paper and for our theoretical discussions, we consider a collection of distributions $\mathcal{F}_d \subset \mathcal{P}_d \times \mathcal{P}_d \times (0, 1)$ satisfying the following assumptions to our data generation process in terms of the data point conditioned on the sensitive attribute and its eigenspectrum:

**Assumption 1.** *Consider our data generation process $a \sim \text{Bernoulli}(p)$ and $\boldsymbol{x} \mid a \sim \mathcal{D}_a$. There exists $\sigma, V, \mathcal{M}, \mathcal{V}, g_{\min} > 0$, $f_{\max} \in (g_{\min}, \infty)$, and $p_{\min} \in (0, 0.5]$ such that the followings hold for any $(\mathcal{D}_0, \mathcal{D}_1, p) \in \mathcal{F}_d$: for $a \in \{0, 1\}$, $\mathcal{D}_a \in \text{nSG}(\sigma)$,*[2]

$$\mathbb{P}\left[\left\|\boldsymbol{x}\boldsymbol{x}^\intercal - (\boldsymbol{\Sigma}_a + \boldsymbol{\mu}_a\boldsymbol{\mu}_a^\intercal)\right\|_2 \leq \mathcal{M} \mid a\right] = 1, \quad \left\|\boldsymbol{\Sigma}_a + \boldsymbol{\mu}_a\boldsymbol{\mu}_a^\intercal\right\|_2 \leq V, \quad \text{Var}\left(\boldsymbol{x}\boldsymbol{x}^\intercal \mid a\right) \leq \mathcal{V},$$

*and*

$$\left\|\boldsymbol{g}\right\|_2 = \left\|\boldsymbol{\Pi}_{\boldsymbol{P}_m}^\perp \boldsymbol{f}\right\|_2 \in \{0\} \cup [g_{\min}, f_{\max}], \quad \left\|\boldsymbol{f}\right\|_2 \leq f_{\max}, \quad \text{and } p \in [p_{\min}, 1 - p_{\min}].$$

**Assumption 2.** *Fix $m, k \in \mathbb{N}$. There exist $\Delta_{m,\nu}, \Delta_{k,\kappa}, K_{m,\nu}, K_{k,\kappa} \in (0, \infty)$ such that for any $(\mathcal{D}_0, \mathcal{D}_1, \cdot) \in \mathcal{F}_d$, the followings hold:*

$$\nu_m - \nu_{m+1} \geq \Delta_{m,\nu}, \quad \kappa_k - \kappa_{k+1} \geq \Delta_{k,\kappa}, \quad \nu_m \leq K_{m,\nu}, \quad \text{and } \kappa_k \leq K_{k,\kappa},$$

*where $\nu_1 \geq \cdots \geq \nu_d \geq 0$ and $\kappa_1 \geq \cdots \geq \kappa_d \geq 0$ are the singular values of $\boldsymbol{S}$ and $\boldsymbol{\Pi}_{\boldsymbol{U}}^\perp \boldsymbol{\Sigma} \boldsymbol{\Pi}_{\boldsymbol{U}}^\perp$, respectively.*

We start by establishing the sample complexity bounds of Algorithms 1 and 2 based on the convergence bound for NPM by Hardt and Price (2014). Recall that NPM is an algorithm for finding top-$r$ eigenspace (in magnitude) of a symmetric (but *not necessarily PSD*) matrix $\boldsymbol{A}$ under a random noise $\boldsymbol{Z} \in \mathbb{R}^{d \times k}$, by update $\boldsymbol{V}_t \leftarrow \text{QR}(\boldsymbol{A}\boldsymbol{V}_{t-1} + \boldsymbol{Z}_t)$. We start by recalling their meta-sample complexity result for NPM, which we have slightly reformulated for our convenience:

**Lemma 1** (Corollary 1.1 of Hardt and Price (2014)). *Let $1 \leq r < d$, $\epsilon \in (0, 1/2)$ and $\delta \in (0, 2e^{-cd})$, where $c$ is an absolute constant.[3] Let $\boldsymbol{L}_r$ be the $d \times r$ matrix, whose columns correspond to the top-$r$ eigenvectors (in magnitude) of the symmetric (not necessarily PSD) matrix $\boldsymbol{A} \in \mathbb{R}^{d \times d}$, and let $\xi_1 \geq \cdots \geq \xi_d \geq 0$ be the singular values of $\boldsymbol{A}$. Assume that the noise matrices $\boldsymbol{Z}_t \in \mathbb{R}^{d \times r}$ satisfy*

$$5\left\|\boldsymbol{Z}_t\right\|_2 \leq \epsilon(\xi_r - \xi_{r+1}) \quad \text{and} \quad 5\left\|\boldsymbol{L}_r^\intercal \boldsymbol{Z}_t\right\|_2 \leq \frac{\delta(\xi_r - \xi_{r+1})}{2\sqrt{dr}}, \quad \forall t \geq 1. \tag{8}$$

*Then, after $T = \Theta\left(\frac{\xi_r}{\xi_r - \xi_{r+1}} \log\left(\frac{d}{\epsilon\delta}\right)\right)$ steps of NPM, $\left\|\boldsymbol{\Pi}_{\boldsymbol{V}_T}^\perp \boldsymbol{L}_r\right\|_2 \leq \epsilon$ with probability at least $1 - \delta$.*

First, we prove that the $\boldsymbol{W}_T$ resulting from Algorithm 1 (NPM for the second moment gap) converges to the true value. We can view Algorithm 1's updates as $\boldsymbol{W}_t \leftarrow \text{QR}(\boldsymbol{S}\boldsymbol{W}_{t-1} + \boldsymbol{Z}_{t,1})$, where the noise matrix in this case is $\boldsymbol{Z}_{t,1} := (\boldsymbol{C}_t^{(1)} - \boldsymbol{C}_t^{(0)}) - \boldsymbol{S}\boldsymbol{W}_{t-1}$ and $\boldsymbol{C}_t^{(a)}$ is as defined in Eqn. (6). The following lemma asserts that with large enough block size $b$, the error matrices are sufficiently bounded such that the NPM iterates converge:

**Lemma 2.** *Let $\epsilon, \delta \in (0, 1)$. It is sufficient to choose the block size $b$ in Algorithm 1 as*

$$b = \Omega\left(\frac{\mathcal{V}}{\Delta_{m,\nu}^2 p_{\min}}\left(\frac{dm}{\delta^2}\log\frac{m}{\delta} + \frac{1}{\epsilon^2}\log\frac{d}{\delta}\right) + \frac{\mathcal{M}^2}{\mathcal{V}p_{\min}}\log\frac{d}{\delta}\right) \tag{9}$$

*to make the following hold with probability at least $1 - \delta$:*

$$5\left\|\boldsymbol{Z}_{t,1}\right\|_2 \leq \epsilon\Delta_{m,\nu} \quad \text{and} \quad 5\left\|\boldsymbol{P}_m^\intercal \boldsymbol{Z}_{t,1}\right\|_2 \leq \frac{\delta\Delta_{m,\nu}}{2\sqrt{dm}}, \quad \forall t \geq 1, \tag{10}$$

*where we recall that the columns of $\boldsymbol{P}_m$ are the top-$m$ (in magnitude) orthonormal eigenvectors of $\boldsymbol{S}$.*

Let $\widehat{\boldsymbol{U}}$ be the final estimate of the true $\boldsymbol{U}$ outputted by Algorithm 1. For Algorithm 2, the noise matrix is $\boldsymbol{Z}_{\tau,2} := \left(\boldsymbol{\Pi}_{\widehat{\boldsymbol{U}}}^\perp \widehat{\boldsymbol{\Sigma}}_\tau \boldsymbol{\Pi}_{\widehat{\boldsymbol{U}}}^\perp - \boldsymbol{\Pi}_{\boldsymbol{U}}^\perp \boldsymbol{\Sigma} \boldsymbol{\Pi}_{\boldsymbol{U}}^\perp\right) \boldsymbol{V}_{\tau-1}$, where $\widehat{\boldsymbol{\Sigma}}_\tau := \frac{1}{\mathcal{B}}\sum_j \boldsymbol{x}_j \boldsymbol{x}_j^\intercal$ is the sample covariance at time step $\tau$ of Algorithm 2. Similarly, with large enough block size $\mathcal{B}$, we have the following lemma:

---

[2]$\boldsymbol{y}$ is norm-sub-Gaussian, denoted as $\text{nSG}(\sigma)$, if $\mathbb{P}[\|\boldsymbol{y} - \mathbb{E}\boldsymbol{y}\| \geq t] \leq 2e^{-\frac{t^2}{2\sigma^2}}$. We refer interested readers to Jin et al. (2019, Section 2) for more discussions on norm-sub-Gaussianity.

[3]It depends polynomially only in the sub-Gaussian moment of the data distribution $\mathcal{D}$; see Theorem 1.1 of Rudelson and Vershynin (2009).

**Lemma 3.** *Let $\epsilon, \delta \in (0,1)$. Suppose that $\left\|\mathbf{\Pi}_{\widehat{U}} - \mathbf{\Pi}_U\right\|_2 \leq \frac{\Delta_{k,\kappa}}{20V} \min\left(\epsilon, \frac{\delta}{2\sqrt{dk}}\right)$. Then, it is sufficient to choose the block size $\mathcal{B}$ in Algorithm 2 as*

$$\mathcal{B} = \Omega\left(\frac{\mathcal{V} + V^2}{\Delta_{k,\kappa}^2}\left(\frac{dk}{\delta^2}\log\frac{k}{\delta} + \frac{1}{\epsilon^2}\log\frac{d}{\delta}\right) + \frac{\mathcal{M}^2}{\mathcal{V} + V^2}\log\frac{d}{\delta}\right), \tag{11}$$

*to make the following hold with probability at least $1 - \delta$:*

$$5\left\|\boldsymbol{Z}_{\tau,2}\right\|_2 \leq \epsilon\Delta_{k,\kappa} \quad \text{and} \quad 5\left\|\boldsymbol{Q}_k^{\mathsf{T}}\boldsymbol{Z}_{\tau,2}\right\|_2 \leq \frac{\delta\Delta_{k,\kappa}}{2\sqrt{dk}}, \quad \forall\tau \geq 1, \tag{12}$$

*where the columns of $\boldsymbol{Q}_k$ is the top-$k$ (in magnitude) orthonormal eigenvectors of $\mathbf{\Pi}_U^\perp\boldsymbol{\Sigma}\mathbf{\Pi}_U^\perp$. (We note that $\boldsymbol{Q}_k$ is a solution for Eqn. (4).)*

**Remark 1.** *We emphasize that, regardless of the block size requirement in the above lemmas, all the operations can be implemented within $o(d^2)$ space requirement.*

Combining the convergence results, we can prove the following PAFO-learnability guarantee in the memory-limited, streaming setting:

**Theorem 1.** *Let $d, m, k \in \mathbb{N}$ be fixed. Consider a collection $\mathcal{F}_d \subset \mathcal{P}_d \times \mathcal{P}_d \times (0,1)$ satisfying Assumptions 1 and 2. Then, $\mathcal{F}_d$ is PAFO-learnable for streaming PCA with FNPM, where the* *sufficient* *number of samples is given as $N_{\mathcal{F}_d}(\varepsilon_{\mathrm{o}}, \varepsilon_{\mathrm{f}}, \delta) = N_1 + N_2$, with*

$$N_1 = \tilde{\Omega}\left(\frac{K_{m,\nu}}{p_{\min}}\left\{\frac{\mathcal{V}}{\Delta_{m,\nu}^3}\left(\frac{dm}{\delta^2} + \frac{\alpha}{\eta_k^2}\right) + \frac{\mathcal{M}^2}{\mathcal{V}\Delta_{m,\nu}}\right\} + \frac{\mathbb{1}[\boldsymbol{g} \neq \boldsymbol{0}]\sigma^2}{p_{\min}g_{\min}^2\eta_k^2}\right), \quad \text{(Algorithm 1)}$$

$$N_2 = \tilde{\Omega}\left(\frac{K_{k,\kappa}(\mathcal{V} + V^2)}{\Delta_{k,\kappa}^3}\left(\frac{dk}{\delta^2} + \frac{k^2V^2}{\varepsilon_{\mathrm{o}}^2}\right) + \frac{K_{k,\kappa}\mathcal{M}^2}{\Delta_{k,\kappa}(\mathcal{V} + V^2)}\right), \quad \text{(Algorithm 2)}$$

*where $\eta_k = \Theta\left(\min\left\{\varepsilon_{\mathrm{f}}, \frac{\Delta_{k,\kappa}}{kV^2}\varepsilon_{\mathrm{o}}, \frac{\Delta_{k,\kappa}}{V\sqrt{dk}}\delta\right\}\right)$ and $\alpha = 1 + \frac{\mathbb{1}[\boldsymbol{g} \neq \boldsymbol{0}]f_{\max}^2}{g_{\min}^2}$.*

Let us take a moment to digest the sample complexity in Theorem 1. The second term $N_2$ is determined by the $\varepsilon_{\mathrm{o}}$-optimality requirement in our PAFO-learnability. Note that $N_2$ has no dependencies on fairness-related quantities such as $\varepsilon_{\mathrm{f}}$, $p_{\min}$, and $\Delta_{m,\nu}$. On the other hand, the first term $N_1$ is not only determined by the $\varepsilon_{\mathrm{f}}$-fairness requirement, but also the $\varepsilon_{\mathrm{o}}$-optimality as well. This is the "price" of pursuing fairness; $\boldsymbol{U}$, which encodes the unfair subspace needed to be nullified, is required to be accurately estimated as it impacts not only the level of fairness but also the resulting solution's optimality. This is clear in our formulation of fair PCA; in Eqn. (4), note how the optimal solution depends heavily on $\mathbf{\Pi}_U$.

We further elaborate on the dependencies of $N_1$ on fairness-related quantities, namely $p_{\min}$ and $\boldsymbol{g}$. First, if $p_{\min} \to 0$, i.e., if one of the two groups is never sampled, then the sample complexity tends to infinity, and the learnability does not hold; this aligns with our intuition, as we need samples from *both* of the sensitive groups. Its dependency is also quite natural, as the minimum expected number of samples from either group depends linearly on $\frac{1}{p_{\min}}$. Also, when $\boldsymbol{g} \neq \boldsymbol{0}$, $N_1$ has an additional additive term scaling linearly with $\frac{1}{g_{\min}^2}$. This is because when $\boldsymbol{g} \neq \boldsymbol{0}$, we must explicitly account for the approximation error of $\frac{\boldsymbol{g}}{\|\boldsymbol{g}\|_2}$ due to the QR decomposition at the end of Algorithm 1.

## 7 Experiments on Real-World Datasets

The code for all experiments is available at `github.com/HanseulJo/fair-streaming-pca`.

### 7.1 CelebA Dataset

We evaluate the efficacy of our proposed FNPM on the CelebA dataset (Liu et al., 2015b). It has been considered by Kleindessner et al. (2023) to show the superior efficiency of their fair PCA algorithm compared to previous approaches (Lee et al., 2022; Olfat and Aswani, 2019; Ravfogel et al., 2022a). However, even in Kleindessner et al. (2023), the images were resized and grey-scaled,

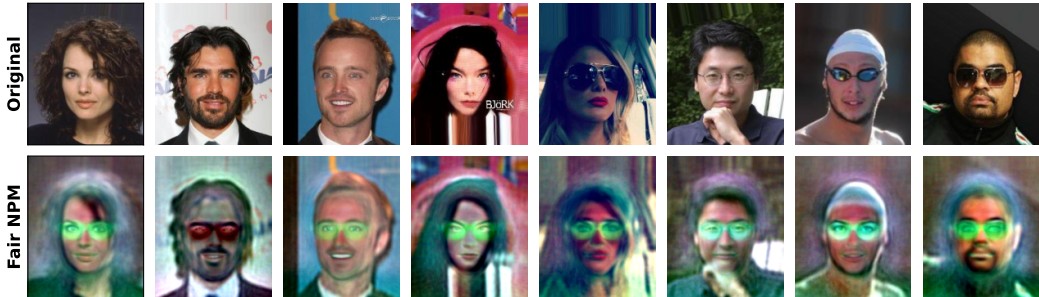

Figure 1: Experimental results on full-resolution **CelebA** dataset. Original image v.s. FNPM output. (Sensitive attribute: "Eyeglasses")

reducing the dimension from the original $218 \times 178 = 38,804$ to $80 \times 80 = 6,400$. Indeed, for a modest-sized computer, it is impossible to load all 162,770 original images in training set to the memory at once, while they require a full dataset to run each step of their algorithm. Here, we use the *original* resolution, full-color CelebA dataset. We implement our FNPM using Python JAX NumPy Module (Bradbury et al., 2023; Harris et al., 2020) and Pytorch (Paszke et al., 2017). All experiments were performed on Apple 2020 Mac mini M1 with 16GB RAM.

Although CelebA is not streaming in nature, we intend to show that transforming it to one and using our memory-efficient approach allow us to *scale up* fair PCA. Since there are three channels of color, we run FNPM channel-wise but in *parallel* as usual in vision tasks (Priorov et al., 2013). For each channel of colors, we project the data onto a $k = 1000$-dimensional subspace while nullifying $m = 2$ leading eigenvectors of covariance difference.

The resulting images are displayed in Figure 1. Here, we consider 'Eyeglasses' as a sensitive attribute to divide groups. We adopt the predefined train-validation split and run our algorithm only on the training set for 5 iterations with block sizes of $b = \mathcal{B} = 32,000$. Then, using the output $V$ of FNPM, we project images selected from the validation set. We observe that we have images of faces wearing colorful glasses by nullifying some of the leading eigenvectors of covariance difference. For the images originally with sunglasses, their glasses get blurred, and "virtual" eyes are added to them. We provide more results on the other attributes and ablation studies on varying $m$ and $b$ in Appendix F.2.

## 7.2   UCI Datasets

For the sake of completeness, we conduct a quantitative evaluation of our algorithm on UCI datasets (Adult Income, COMPAS, German Credit) and compare it with six previous works (Kleindessner et al., 2023; Lee et al., 2022; Olfat and Aswani, 2019; Ravfogel et al., 2020, 2022a; Samadi et al., 2018). The results for the Adult Income dataset are shown in Table 1. We assess several variants of our methods: in Table 1, "mean" is when we match the means and not second moments, "FNPM" is when we run our Algorithm 2 with a block size of full-batch, and "offline" is when we directly solve our "Null It Out" formulation of fair PCA (Eqn. (4)) via offline eigen-decomposition. In the table, we report the explained variance (%Var), representation fairness measured by maximum mean discrepancy ($MMD^2$), downstream task accuracy (%Acc), and downstream task fairness in demographic parity ($\%\Delta_{DP}$). The result showcases that our method yields competitive quantitative performance even for the common tabular datasets while being much more memory efficient. We defer the full results for other UCI datasets to Appendix F.3.

## 8   Other Related Works

**Fairness in ML.**   There are largely two directions in the literature of algorithmic fairness. One direction is to propose a suitable and meaningful fairness definition (Dwork et al., 2012; Feldman et al., 2015; Hardt et al., 2016). The other direction is to develop *efficient* fair algorithms, although often the fairness constraint forces the algorithm to be much more inefficient than its unfair counterpart, or it calls for a need for a completely new algorithmic approach. There are also different ways of

Table 1: Dataset = **Adult Income** [feature dim=102, #(train data)=31,655], $k = 2$

| Method | %Var($\uparrow$) | MMD$^2$($\downarrow$) | %Acc($\uparrow$) kernel SVM | $\Delta_{\mathrm{DP}}$($\downarrow$) |
|---|---|---|---|---|
| PCA | $6.88_{(0.14)}$ | $0.374_{(0.006)}$ | $82.4_{(0.2)}$ | $0.19_{(0.01)}$ |
| Olfat and Aswani (2019) (0.1) | | Memory Out | | |
| Olfat and Aswani (2019) (0.0) | | Memory Out | | |
| Lee et al. (2022) (1e-3) | $5.68_{(0.11)}$ | $0.0_{(0.0)}$ | $80.34_{(0.24)}$ | $0.05_{(0.01)}$ |
| Lee et al. (2022) (1e-6) | $5.42_{(0.11)}$ | $0.0_{(0.0)}$ | $79.41_{(0.23)}$ | $0.02_{(0.01)}$ |
| Kleindessner et al. (2023) (mean) | $5.74_{(0.11)}$ | $0.002_{(0.0)}$ | $80.6_{(0.2)}$ | $0.07_{(0.01)}$ |
| Kleindessner et al. (2023) (0.85) | $4.09_{(0.17)}$ | $0.001_{(0.001)}$ | $75.52_{(0.21)}$ | $0.0_{(0.0)}$ |
| Kleindessner et al. (2023) (0.5) | $2.63_{(0.07)}$ | $0.0_{(0.0)}$ | $75.38_{(0.18)}$ | $0.0_{(0.0)}$ |
| Kleindessner et al. (2023) (kernel) | | Takes too long time | | |
| Ravfogel et al. (2020) | $1.91_{(0.08)}$ | $0.001_{(0.001)}$ | $75.67_{(0.31)}$ | $0.0_{(0.0)}$ |
| Ravfogel et al. (2022a) | $1.91_{(0.09)}$ | $0.006_{(0.011)}$ | $75.59_{(0.34)}$ | $0.0_{(0.0)}$ |
| Samadi et al. (2018) | N/A | N/A | $82.63_{(0.18)}$ | $0.15_{(0.01)}$ |
| **Ours** (offline, mean) | $5.74_{(0.11)}$ | $0.002_{(0.0)}$ | $80.6_{(0.2)}$ | $0.07_{(0.01)}$ |
| **Ours** (FNPM, mean) | $5.74_{(0.11)}$ | $0.002_{(0.0)}$ | $80.6_{(0.2)}$ | $0.07_{(0.01)}$ |
| **Ours** (offline, $m$=15) | $4.04_{(0.14)}$ | $0.001_{(0.001)}$ | $75.51_{(0.23)}$ | $0.0_{(0.0)}$ |
| **Ours** (FNPM, $m$=15) | $4.07_{(0.13)}$ | $0.001_{(0.0)}$ | $75.54_{(0.25)}$ | $0.0_{(0.0)}$ |
| **Ours** (offline, $m$=50) | $2.63_{(0.07)}$ | $0.0_{(0.0)}$ | $75.38_{(0.18)}$ | $0.0_{(0.0)}$ |
| **Ours** (FNPM, $m$=50) | $2.64_{(0.06)}$ | $0.0_{(0.0)}$ | $75.44_{(0.21)}$ | $0.0_{(0.0)}$ |

imposing fairness in an ML pipeline, such as learning fair pre-processing (Biswas and Rajan, 2021), fair in-processing (Roh et al., 2021; Wan et al., 2023; Zafar et al., 2019), and more. The reader is encouraged to check Barocas et al. (2019) for a more comprehensive treatment of this subject.

**Fair online/streaming Learning.** Bechavod et al. (2020); Gillen et al. (2018) study individual fairness in online learning in a learning theoretic framework, even when the underlying metric is unavailable. Stemming from the concept of fair clustering as proposed in Chierichetti et al. (2017), Bera et al. (2022); Schmidt et al. (2020) study imposing demographic parity on clustering in the streaming setting. Such fairness has been considered in various other streaming problems such as online selection (Correa et al., 2021), streaming submodular optimization (El Halabi et al., 2020), and diversity maximization (Wang et al., 2022). Quite surprisingly, demographic parity (or any other concept of fairness) has never been considered in the setting of streaming PCA.

**Streaming PCA.** Without the fairness constraint, streaming PCA has been studied much from statistics and the machine learning community. Two prominent algorithms have been studied; the noisy power method (Mitliagkas et al., 2013) and Oja's method (Oja, 1982). Much work has been done in improving the theoretical guarantees of streaming PCA (Balcan et al., 2016; Hardt and Price, 2014; Jain et al., 2016; Liang, 2023), improving the algorithm itself (Xu, 2023; Yun, 2018), or extending the guarantees to various different settings (Balzano et al., 2018; Bienstock et al., 2022; Kumar and Sarkar, 2023). Memory-limited, streaming versions of somewhat related problems, such as community detection (Yun et al., 2014) and low-rank matrix completion (Yun et al., 2015), have been tackled as well using similar spectral techniques as PCA (e.g., power method). However, to the best of our knowledge, fairness (regardless of the definition) has never been considered in this context of streaming PCA, which we tackle in this work and which we believe is of great importance.

## 9   Conclusion

In this work, we tackled the two outstanding problems of the existing fair PCA literature. From the theoretical side, we illustrated a new formulation of fair PCA based on the "Null It Out" approach and then provided a novel statistical framework called PAFO-learnability of fair PCA. From the practical side, we addressed the memory-limited scenarios by proposing a new problem setting called fair streaming PCA and a memory-efficient two-stage algorithm called FNPM. Based on these, we established a statistical guarantee that our algorithm achieves the PAFO-learnability for fair streaming PCA. Lastly, we ran experiments on the CelebA and UCI datasets to certify the scalability of our method.

## Acknowledgments and Disclosure of Funding

We thank the anonymous reviewers for their helpful comments and suggestions. We also thank Gwangsu Kim (Jeonbuk National University) for the helpful discussions in the initial phase of the research. This work was supported by the Institute of Information & Communications Technology Planning & Evaluation (IITP) grants funded by the Korean government(MSIT) (No.2019-0-00075, Artificial Intelligence Graduate School Program(KAIST); No. 2022-0-00184, Development and Study of AI Technologies to Inexpensively Conform to Evolving Policy on Ethics).

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

# Contents

# Appendix

## A  Broader Impacts, Limitations, and Future Directions

### A.1  Broader Impacts

This work proposes a dimensionality reduction method that addresses memory efficiency *and* fairness while providing a statistical guarantee. By identifying and nullifying the "unfair direction" inherent in the data distribution, we offer an alternative approach to fair PCA. We anticipate that our approach will motivate researchers to explore other dimensionality reduction techniques (*e.g.,* auto-encoder) with fairness constraints. Significantly, our contribution includes a rigorous theoretical framework, PAFO learnability, which enables sample complexity analysis of fair PCA. We envision the potential for our theory to be further generalized to broader contexts, including alternative definitions of fairness and optimality of different algorithms for fair machine learning.

On the application side, the memory efficiency of our method can facilitate the scalability of fair PCA, making it viable for processing high-dimensional datasets, even in scenarios where data points arrive in a streaming fashion. One possible application of our approach is data pre-/post-processing to alleviate unfairness, such as generating fair word embeddings by eliminating sensitive attribute information through orthogonal projection. For more detailed discussions on potential future directions, please refer to below.

### A.2  Limitations and Future Directions

Here, we list some of our work's limitations and possible extensions/future directions.

**Individual Fairness for PCA.**  Our definition of fairness in PCA only covers group fairness via fair presentation learning, which was also the case for the prior works (Kleindessner et al., 2023; Lee et al., 2022; Olfat and Aswani, 2019). None of the works, including ours, have yet to consider the notion of individual fairness (Dwork et al., 2012) in the context of PCA, for which we do not have a definitive answer.

**Knowledge of sensitive attribute.**  Our framework requires the *full* knowledge of the sensitive attribute $s$ for all data points $x$'s, which was also the case for all the previous works (Kleindessner et al., 2023; Lee et al., 2022; Olfat and Aswani, 2019). But, the assumption of such knowledge may not be feasible in the real world due to privacy or legal reasons (Lahoti et al., 2020; Zhao et al., 2022), or even just due to some extrinsic noises. Considering the case where the dataset may lack sensitive attributes for some or all of the data points is an important future direction.

**Making the algorithm anytime.**  In our formulation of fair PCA, our algorithm is two-phase, with the first phase as a "burn-in" period for estimating the unfair subspace. Thus, it is not an anytime algorithm in that if the algorithm stops whilst in the first phase, then the resulting $V$ is random and completely uninformative. Designing an anytime variant of our algorithm is an important future direction. One possible way to achieve that is to consider other streaming PCA algorithms such as Oja's method (Oja and Karhunen, 1985) or accelerated NPM (Xu, 2023; Xu and Li, 2022).

**Improving gap dependence.**  Our algorithm's current sample complexity analysis relies on the analysis of NPM by Hardt and Price (2014), which relies on the immediate singular value gap, $\sigma_k - \sigma_{k+1}$. Balcan et al. (2016) showed that considering greater iteration rank $q \geq k$, *i.e.,* by considering optimization variable of greater size, leads to a better gap dependency: from $\sigma_k - \sigma_{k+1}$ to $\sigma_k - \sigma_{q+1}$. However, this is incompatible with our current definition of PAFO-learnability (Definition 2) because with greater iteration rank, the solution $V^\star$ to which the explained variance should be compared against is not clear. The problem is that the sample complexity guarantee of Balcan et al. (2016); Hardt and Price (2014) is derived in terms of the *sine* angle between the top $k$-eigenspace of the true covariance and the estimated $q$-dimensional subspace. Clearing this up would allow for a better theoretical guarantee in sample complexity and thus an important future direction.

**Kernelizing our framework**  Kernel PCA (Schölkopf et al., 1998) is a cornerstone in modern machine learning that has lent itself to be an inspiration to many applications, both theory and practice-wise. Unlike previous fair PCA works (Kleindessner et al., 2023; Olfat and Aswani, 2019), in which the authors provided a kernelized version of their fair PCA algorithm, both our statistical framework and memory-efficient algorithm for streaming setting do not readily extend to the kernelized version. Algorithmically, to tackle the streaming setting, one may take inspiration from streaming kernel PCA (Ghashami et al., 2016; Ullah et al., 2018), which was in turn inspired by matrix sketching (Liberty, 2013).

**Extending to other settings.**  In a similar spirit, extending our formulation of fair PCA to other streaming settings such as sparse (Wang and Lu, 2016; Yang and Xu, 2015), nonstationary (Bienstock et al., 2022), or even distributionally robust settings (Vu et al., 2022) would also be interesting.

**More experiments**  Last but not least, we've only performed experiments on synthetic, CelebA, and UCI datasets. It would be interesting to try our framework (either the statistical formulation, the streaming setting, or both) on datasets from other domains, such as NLP and graphs. GloVe vectors (Pennington et al., 2014) has been used as a benchmark in a similar task called "concept erasure" (Ravfogel et al., 2020, 2022b); group fairness in spectral clustering of graphs has been studied as well (Kleindessner et al., 2019; Wang et al., 2023).

# B    Full Pseudocodes of Algorithms 1 and 2 for Fair "Streaming" PCA

Throughout the appendices, from hereon, we will refer to our algorithms by Algorithms 3 and 4 instead of Algorithms 1 and 2.

---

**Algorithm 3:** `UnfairSubspace` (= Algorithm 1)

---

1 **Input:** $d$, $m$, Block size $b$, Number of iterations $T$;
2 **Output:** A matrix $\widehat{U}$ with orthonormal columns;
3 $\boldsymbol{W}_0 \leftarrow \mathtt{QR}(\mathcal{N}(0,1)^{d \times m})$;
4 $(\overline{\boldsymbol{m}}^{(0)}, \overline{\boldsymbol{m}}^{(1)}, B^{(0)}, B^{(1)}) \leftarrow (\mathbf{0}_d, \mathbf{0}_d, 0, 0)$;
5 **for** $t \in [T]$ **do**
6      $(b_t^{(0)}, b_t^{(1)}) \leftarrow (0, 0)$;
7      $(\boldsymbol{m}_t^{(0)}, \boldsymbol{m}_t^{(1)}, \boldsymbol{C}_t^{(0)}, \boldsymbol{C}_t^{(1)}) \leftarrow (\mathbf{0}_d, \mathbf{0}_d, \mathbf{0}_{d \times m}, \mathbf{0}_{d \times m})$;
8      **for** $i \in [b]$ **do**
9          Receive $(a, \boldsymbol{x}) = (a_{(t-1)b+i}, \boldsymbol{x}_{(t-1)b+i})$;
10          $b_t^{(a)} \leftarrow b_t^{(a)} + 1$;
11          $\boldsymbol{m}_t^{(a)} \leftarrow \boldsymbol{m}_t^{(a)} + \boldsymbol{x}$;
12          $\boldsymbol{C}_t^{(a)} \leftarrow \boldsymbol{C}_t^{(a)} + \boldsymbol{x}\boldsymbol{x}^{\intercal}\boldsymbol{W}_{t-1}$;
13      **foreach** $a \in \{0, 1\}$ **do**
14          **if** $b_t^{(a)} > 0$ **then**
15              $\boldsymbol{m}_t^{(a)} \leftarrow \frac{1}{b_t^{(a)}}\boldsymbol{m}_t^{(a)}$;
16              $\boldsymbol{C}_t^{(a)} \leftarrow \frac{1}{b_t^{(a)}}\boldsymbol{C}_t^{(a)}$;
17          $\overline{\boldsymbol{m}}^{(a)} \leftarrow \frac{B^{(a)}}{B^{(a)}+b_t^{(a)}}\overline{\boldsymbol{m}}^{(a)} + \frac{b_t^{(a)}}{B^{(a)}+b_t^{(a)}}\boldsymbol{m}_t^{(a)}$;
18          $B^{(a)} \leftarrow B^{(a)} + b_t^{(a)}$;
19      $\boldsymbol{W}_t = \mathtt{QR}\left(\boldsymbol{C}^{(1)} - \boldsymbol{C}^{(0)}\right)$;
20 $\widehat{\boldsymbol{f}} \leftarrow \overline{\boldsymbol{m}}^{(1)} - \overline{\boldsymbol{m}}^{(0)}$;
21 $\widetilde{\boldsymbol{g}} \leftarrow \widehat{\boldsymbol{f}} - \boldsymbol{W}_T\boldsymbol{W}_T^{\intercal}\widehat{\boldsymbol{f}}$;
22 **if** $\|\widehat{\boldsymbol{g}}\|_2 = 0$ **then**
23      $\widehat{\boldsymbol{U}} = \boldsymbol{W}_T$
24 **else**
25      $\widehat{\boldsymbol{U}} = \left[\boldsymbol{W}_T \mid \frac{\widehat{\boldsymbol{g}}}{\|\widehat{\boldsymbol{g}}\|_2}\right]$
26 **return** $\widehat{\boldsymbol{U}}$

---

**Algorithm 4:** Fair NPM (= Algorithm 2)

---

1 **Input:** $d$, $k$, Block sizes $\mathcal{B}$, $b$, Numbers of iterations $\mathcal{T}$, $T$;
2 **Output:** $\boldsymbol{V}_{\mathcal{T}} \in \mathrm{St}(d, k)$;
3 $\widehat{\boldsymbol{U}} \leftarrow \mathtt{UnfairSubspace}(b, T)$;
4 $\boldsymbol{V}_0 \leftarrow \mathtt{QR}(\mathcal{N}(0,1)^{d \times k})$;
5 **for** $\tau \in [\mathcal{T}]$ **do**
6      $\boldsymbol{V}_{\tau} \leftarrow \boldsymbol{V}_{\tau-1} - \widehat{\boldsymbol{U}}\widehat{\boldsymbol{U}}^{\intercal}\boldsymbol{V}_{\tau-1}$;
7      $\boldsymbol{C} \leftarrow \mathbf{0}_{d \times k}$;
8      **for** $j \in [\mathcal{B}]$ **do**
9          Receive $(*, \widetilde{\boldsymbol{x}}) = (*, \widetilde{\boldsymbol{x}}_{(\tau-1)\mathcal{B}+j})$;
10          $\boldsymbol{C} \leftarrow \boldsymbol{C} + \frac{1}{\mathcal{B}}\widetilde{\boldsymbol{x}}\widetilde{\boldsymbol{x}}^{\intercal}\boldsymbol{V}_{\tau}$;
11      $\boldsymbol{V}_{\tau} \leftarrow \mathtt{QR}\left(\boldsymbol{C} - \widehat{\boldsymbol{U}}\widehat{\boldsymbol{U}}^{\intercal}\boldsymbol{C}\right)$;
12 **return** $\boldsymbol{V}_{\mathcal{T}}$

---

## C   Extension of Algorithms 3 to Multiple and Non-binary Attributes

We remark that Algorithm 3 is a special case of the pseudocode below. By applying Algorithm 4 after applying this algorithm, we can handle the multiple non-binary attribute case and approximately and memory-efficiently solve the extended notion of fair PCA elaborated in Section 5.3.

---

**Algorithm 5:** `UnfairSubspace_Extended`

---

1   **Input:** $d, m_1, \cdots, m_\ell$, Block size $b$, Number of iterations $T$;

2   **Output:** A matrix $\widehat{U}$ with orthonormal columns;

3   **for** $r \in [\ell]$ **do**

4      **for** $a \in [g_r]$ **do**

5         $\boldsymbol{W}_0^{(r,a)} = \mathrm{QR}(\mathcal{N}(0,1)^{d \times m_r})$;

6         $(\overline{\boldsymbol{m}}^{(r,a,\mathrm{in})}, \overline{\boldsymbol{m}}^{(r,a,\mathrm{out})}, B^{(r,a)}) \leftarrow (\mathbf{0}_d, \mathbf{0}_d, 0)$

7   **for** $t \in [T]$ **do**

8      Receive $\{((a_{1,i}, \cdots, a_{\ell,i}), \boldsymbol{x}_i)\}_{i=b(t-1)+1}^{bt}$;

9      **for** $r \in [\ell]$ **do**

10         **for** $a \in [g_r]$ **do**

11            $b_t^{(r,a)} = \sum_{i=b(t-1)+1}^{bt} \mathbb{1}_{[a_{r,i}=a]}$;

12            $B^{(r,a)} \leftarrow B^{(r,a)} + b_t^{(r,a)}$;

13            **if** $b_t^{(a)} > 0$ **then**

14                $\boldsymbol{m}_t^{(r,a,\mathrm{in})} = \frac{1}{b_t^{(a)}} \sum_{i=b(t-1)+1}^{bt} \mathbb{1}_{[a_{r,i}=a]} \boldsymbol{x}_i$;

15                $\boldsymbol{C}_t^{(r,a,\mathrm{in})} = \frac{1}{b_t^{(a)}} \sum_{i=b(t-1)+1}^{bt} \mathbb{1}_{[a_{r,i}=a]} \boldsymbol{x}_i \boldsymbol{x}_i^\mathsf{T} \boldsymbol{W}_{t-1}^{(r,a)}$;

16            **else**

17                $(\boldsymbol{m}_t^{(r,a,\mathrm{in})}, \boldsymbol{C}_t^{(r,a,\mathrm{in})}) \leftarrow (\mathbf{0}_d, \mathbf{0}_{d \times m_r})$

18            **if** $b_t^{(a)} < b$ **then**

19                $\boldsymbol{m}_t^{(r,a,\mathrm{out})} = \frac{1}{b-b_t^{(a)}} \sum_{i=b(t-1)+1}^{bt} \mathbb{1}_{[a_{r,i} \neq a]} \boldsymbol{x}_i$;

20                $\boldsymbol{C}_t^{(r,a,\mathrm{out})} = \frac{1}{b-b_t^{(a)}} \sum_{i=b(t-1)+1}^{bt} \mathbb{1}_{[a_{r,i} \neq a]} \boldsymbol{x}_i \boldsymbol{x}_i^\mathsf{T} \boldsymbol{W}_{t-1}^{(r,a)}$;

21            **else**

22                $(\boldsymbol{m}_t^{(r,a,\mathrm{out})}, \boldsymbol{C}_t^{(r,a,\mathrm{out})}) \leftarrow (\mathbf{0}_d, \mathbf{0}_{d \times m_r})$

23            $\overline{\boldsymbol{m}}^{(r,a,\mathrm{in})} \leftarrow \frac{B^{(r,a)} - b_t^{(r,a)}}{B^{(a)}} \overline{\boldsymbol{m}}^{(r,a,\mathrm{in})} + \frac{b_t^{(a)}}{B^{(a)}} \boldsymbol{m}_t^{(r,a,\mathrm{in})}$;

24            $\overline{\boldsymbol{m}}^{(r,a,\mathrm{out})} \leftarrow \frac{(t-1)b - B^{(r,a)} + b_t^{(r,a)}}{tb - B^{(a)}} \overline{\boldsymbol{m}}^{(r,a,\mathrm{out})} + \frac{b - b_t^{(a)}}{tb - B^{(a)}} \boldsymbol{m}_t^{(r,a,\mathrm{out})}$;

25            $\boldsymbol{W}_t^{(r,a)} = \mathrm{QR}\left( \boldsymbol{C}_t^{(r,a,\mathrm{in})} - \boldsymbol{C}_t^{(r,a,\mathrm{out})} \right)$;

26   **for** $r \in [\ell]$ **do**

27      **for** $a \in [g_r]$ **do**

28         $\widehat{\boldsymbol{f}}^{(r,a)} = \overline{\boldsymbol{m}}_t^{(r,a,\mathrm{in})} - \overline{\boldsymbol{m}}_t^{(r,a,\mathrm{out})}$;

29   $\widehat{U} = \mathrm{QR}\left( \left[ \boldsymbol{W}_t^{(1,1)} \mid \cdots \mid \boldsymbol{W}_t^{(\ell,g_\ell)} \mid \widehat{\boldsymbol{f}}^{(1,1)} \mid \cdots \mid \widehat{\boldsymbol{f}}^{(\ell,g_\ell)} \right] \right)$;

30   **return** $\widehat{U}$

---

# D   More Detailed Comparison to Kleindessner et al. (2023)

## D.1   Their Approach

Kleindessner et al. (2023) considered the following formulation of fair PCA:

$$\max_{\boldsymbol{V} \in \mathrm{St}(d,k)} \mathrm{tr}(\boldsymbol{V}^\mathsf{T} \boldsymbol{X}^\mathsf{T} \boldsymbol{X} \boldsymbol{V}), \quad \text{subject to } \boldsymbol{f}^\mathsf{T} \boldsymbol{V} = \boldsymbol{0} \ \wedge \ \boldsymbol{V}^\mathsf{T}(\boldsymbol{\Sigma}_1 - \boldsymbol{\Sigma}_0)\boldsymbol{V} = \boldsymbol{0}, \tag{13}$$

and proposed a reasonable approximation of the covariance constraint, which we briefly describe here. Let us write $\boldsymbol{S}' = \boldsymbol{\Sigma}_1 - \boldsymbol{\Sigma}_0$ for brevity. As implicitly assumed in Kleindessner et al. (2023), let us assume that $\boldsymbol{f} \neq \boldsymbol{0}$. The mean constraint is dealt with first. Denoting $\boldsymbol{U_f} \in \mathrm{St}(d, d-1)$ to be the matrix whose columns form a basis $(d-1)$-dimensional nullspace of $\boldsymbol{f}$, the mean constraint is satisfied if and only if $\boldsymbol{V}$ is of the form $\boldsymbol{U_f}\boldsymbol{U}$ with $\boldsymbol{U} \in \mathrm{St}(d-1, k)$ as the intermediate optimization variable. With this first reparametrization, the optimization now becomes

$$\max_{\boldsymbol{U} \in \mathrm{St}(d-1,k)} \mathrm{tr}(\boldsymbol{U}^\mathsf{T} \boldsymbol{U_f}^\mathsf{T} \boldsymbol{X}^\mathsf{T} \boldsymbol{X} \boldsymbol{U_f} \boldsymbol{U}), \quad \text{subject to } \boldsymbol{U}^\mathsf{T} \boldsymbol{U_f}^\mathsf{T} \boldsymbol{S}' \boldsymbol{U_f} \boldsymbol{U} = \boldsymbol{0}, \tag{14}$$

which now has only the covariance constraint.

To deal with the possible infeasibility of the covariance constraint, Kleindessner et al. (2023) proposed the following approach: letting $\boldsymbol{M_{f,S'}} \in \mathrm{St}(d-1, l)$ be the matrix whose columns are the $l$ smallest eigenvectors of $\boldsymbol{U_f}^\mathsf{T} \boldsymbol{S}' \boldsymbol{U_f}$ (in magnitude), $\boldsymbol{U}$ only needs to nullify the eigenspace spanned by the remaining $d-1-l$ eigenvectors. To see which terms are ignored with the relaxation, let $\sum_{i=1}^{d} q_i' \boldsymbol{v}' \boldsymbol{v}_i'^\mathsf{T}$ be the eigenvalue decomposition of $\boldsymbol{U_f}^\mathsf{T} \boldsymbol{S}' \boldsymbol{U_f}$ with $|q_1'| \geq |q_2'| \geq \cdots |q_d'| \geq 0$. This approach essentially ignores the constraint $\boldsymbol{U}^\mathsf{T} \boldsymbol{E}_l \boldsymbol{U} = \boldsymbol{0}$, where

$$\boldsymbol{E}_l = \boldsymbol{M_{f,S'}}^\mathsf{T} \boldsymbol{U_f}^\mathsf{T} \boldsymbol{S}' \boldsymbol{U_f} \boldsymbol{M_{f,S'}} = \sum_{i=d-l}^{d-1} q_i' \boldsymbol{v}' \boldsymbol{v}_i'^\mathsf{T}. \tag{15}$$

The number $l \in \{k, \cdots, d-1\}$ controls how much the group-conditional covariances will be equalized; a smaller $l$ means that the covariance constraint is enforced more stringently and vice versa. Ultimately, the learner can control $l$ as a hyperparameter to create a trade-off between fairness and the explained variance. Moreover, *if* $|q_i'|$'s are negligible for all $i \geq d-1-l$, then $\boldsymbol{M_{f,S'}}^\mathsf{T} \boldsymbol{U_f}^\mathsf{T} \boldsymbol{S}' \boldsymbol{U_f} \boldsymbol{M_{f,S'}} \approx \boldsymbol{0}$, and the relaxation becomes tighter.

As any $\boldsymbol{U}$ of the form $\boldsymbol{M_{f,S'}} \boldsymbol{\Lambda}$ satisfies the relaxed covariance constraint, a second reparameterization in terms of the new optimization variable $\boldsymbol{\Lambda} \in \mathrm{St}(l, k)$ gives us

$$\max_{\boldsymbol{\Lambda} \in \mathrm{St}(l,k)} \mathrm{tr}(\boldsymbol{\Lambda}^\mathsf{T} \boldsymbol{M_{f,S'}}^\mathsf{T} \boldsymbol{U_f}^\mathsf{T} \boldsymbol{X}^\mathsf{T} \boldsymbol{X} \boldsymbol{U_f} \boldsymbol{M_{f,S'}} \boldsymbol{\Lambda}), \tag{16}$$

which can be solved via the standard SVD-based approach for vanilla PCA. Then, the final solution is obtained as $\boldsymbol{V}^* = \boldsymbol{M_{f,S'}} \boldsymbol{U_f} \boldsymbol{\Lambda}^*$.

## D.2   Unsuitability for the Streaming Setting

In Section 5.2, we provided a rough overview of why existing approaches to fair PCA (Kleindessner et al., 2023; Lee et al., 2022; Olfat and Aswani, 2019) are not amenable to our streaming setting. Especially as the approach of Kleindessner et al. (2023) (described above) is almost like a PCA, one may wonder if standard techniques used in streaming PCA (Mitliagkas et al., 2013) can be used. Here, we argue in detail why that is *not* the case, which is in sharp contrast to our approach and our reformulation of fair PCA that led to a memory-efficient fair streaming PCA algorithm.

**Memory constraint.**   For streaming PCA without fairness constraints, the main objective could be written as $\mathrm{tr}(\boldsymbol{V}^\mathsf{T} \boldsymbol{X}^\mathsf{T} \boldsymbol{X} \boldsymbol{V}) = \sum_{i=1}^{N} \mathrm{tr}(\boldsymbol{V}^\mathsf{T} \boldsymbol{x}_i \boldsymbol{x}_i^\mathsf{T} \boldsymbol{V})$, which is easily amenable to *memory-limited* algorithm such as noisy power method (Mitliagkas et al., 2013) or stochastic optimization (Oja, 1982). Both approaches utilize the fact that instead of storing $d \times d$ matrices, one only needs to store matrix-vector product of size $d \times 1$, e.g., $\boldsymbol{V}^\mathsf{T} \boldsymbol{x}_i$. This is not the case for the approach of Kleindessner et al. (2023). To see this, consider Eqn. (14) without the covariance constraint, i.e., fair PCA with only the mean constraint. Even here, although the objective can be written as $\sum_{i=1}^{N} \mathrm{tr}\left(\boldsymbol{U}^\mathsf{T} \boldsymbol{U_f}^\mathsf{T} \boldsymbol{x}_i \boldsymbol{x}_i^\mathsf{T} \boldsymbol{U_f} \boldsymbol{U}\right),$

we must know $U_f$ in order to proceed further. Moreover, as $U_f$ is of size $\mathcal{O}(d^2)$, it cannot be stored nor estimated explicitly. When the covariance constraint is also taken into account, although the matrix in question $M_{f,S'}$ is of dimension $(d-1) \times l$, which is within the memory constraint if $l = \mathcal{O}(1)$, the computation of $M_{f,S'}$ *requires* the knowledge of $U_f$. This is because Kleindessner et al. (2023) dealt with the two constraints sequentially (mean first, then covariance), which forced the reparametrization to be done twice and, more importantly, coupling the memory requirement for the computation of $U_f$ and $M_{f,S'}$.

**Statistical consideration.**    The statistical guarantee (global convergence, sample complexity) of streaming PCA algorithms is often obtained by appropriately bounding the error term and using proper matrix concentration inequalities (Tropp, 2015). For example, for the sample complexity guarantee of noisy power method (Balcan et al., 2016; Hardt and Price, 2014), as the learner performs power method on the empirical covariance $\sum_i x_i x_i^\intercal$ instead of the true covariance $\Sigma$, the proof proceeds by first showing that the empirical covariance is close enough to the true covariance (i.e., the norm of their error term is sufficiently bounded), which then implies that the variance of the estimation isn't too high. Thus to analyze the error bound of the final iterates $V$ when the approach of Kleindessner et al. (2023) is extended to a streaming setting, one would have to bound the estimation error of $M_{f,S'}^\intercal U_f^\intercal \Sigma U_f M_{f,S'}$. There are three sources of estimation error: $M_{f,S'}, U_f^\intercal, \Sigma$. Recalling that $M_{f,S'}$ consists of the *eigenvectors* of $U_f^\intercal S' U_f$, one can see that the estimation error of $M_{f,S'}$ is actually nontrivially dependent on the estimation quality of *both $U$ and $S'$*. Here, we say nontrivially because the error isn't simply bounded in a linear sense via the usual triangle inequality; it requires rather intricate techniques involving eigenvector perturbation theory (Davis and Kahan, 1969; Golub and Loan, 2013), which may require additional assumptions on eigenvalue gaps. As one can see later in the proof, our approach considers both constraints simultaneously and thus allows for a quite simple theoretical analysis.

# E Proofs of Lemma 2, 3, and Theorem 1

## E.1 Notations and Assumptions

We first recall the data generation process of our interest. At every time step, a pair of a sensitive attribute and a data point is sampled as $a \sim \text{Bernoulli}(p)$ and $\boldsymbol{x}|a \sim \mathcal{D}_a$, respectively, where $\mathcal{D}_a$ has mean $\boldsymbol{\mu}_a$ and covariance $\boldsymbol{\Sigma}_a$. This can be written more compactly as $\boldsymbol{x} \sim \mathcal{D} := p_0\mathcal{D}_0 + p_1\mathcal{D}_1$, where we denote $p_0 := 1 - p$ and $p_1 := p$.

We recall a set of notation needed for the proof. The mean difference is denoted by $\boldsymbol{f} = \boldsymbol{\mu}_1 - \boldsymbol{\mu}_0$, while the second moment difference is $\boldsymbol{S} = \mathbb{E}[\boldsymbol{x}\boldsymbol{x}^\mathsf{T}|a=1] - E[\boldsymbol{x}\boldsymbol{x}^\mathsf{T}|a=0] = \boldsymbol{\Sigma}_1 - \boldsymbol{\Sigma}_0 + \boldsymbol{\mu}_1\boldsymbol{\mu}_1^\mathsf{T} - \boldsymbol{\mu}_0\boldsymbol{\mu}_0^\mathsf{T}$. Let $\boldsymbol{P}_m \in \mathbb{R}^{d \times m}$ be the matrix whose columns are top $m$ eigenvectors of $\boldsymbol{S}$ in magnitude of eigenvalues. Note that $t \in [T]$ is the time variable for Algorithm 3, which we often omit if the context is clear. Moreover, $b$ (without any sub-/superscript) is the block size of Algorithm 3. The remaining notation is listed below; most of them came from the pseudocode of our algorithms.

- Counts of data points for each sensitive attribute $a$:

$$b_t^{(a)} = \sum_{i=(t-1)b+1}^{tb} \mathbb{1}_{[a_i=a]}, \qquad B^{(a)} = \sum_{t=1}^{T} b_t^{(a)} = \sum_{i=1}^{Tb} \mathbb{1}_{[a_i=a]}$$

- Estimates of group-conditional means:

$$\overline{\boldsymbol{m}}^{(a)} = \begin{cases} \dfrac{1}{TB^{(a)}} \displaystyle\sum_{i=1}^{Tb} \mathbb{1}_{[a_i=a]}\boldsymbol{x}_i & \text{if } B^{(a)} > 0, \\ \boldsymbol{0} & \text{if } B^{(a)} = 0 \end{cases}$$

- Estimate of the group-conditional mean gap: $\widehat{\boldsymbol{f}} = \overline{\boldsymbol{m}}^{(1)} - \overline{\boldsymbol{m}}^{(0)}$

- Estimate of the group-conditional second moments:

$$\boldsymbol{A}_t^{(a)} = \begin{cases} \dfrac{1}{b_t^{(a)}} \displaystyle\sum_{i=(t-1)b+1}^{tb} \mathbb{1}_{[a_i=a]}\boldsymbol{x}_i\boldsymbol{x}_i^\mathsf{T} & \text{if } b_t^{(a)} > 0, \\ \boldsymbol{0} & \text{if } b_t^{(a)} = 0 \end{cases}$$

- Estimate of the group-conditional second moment gap: $\widehat{\boldsymbol{S}}_t = \boldsymbol{A}_t^{(1)} - \boldsymbol{A}_t^{(0)}$
- $\boldsymbol{W}_t \in \text{St}(d, n)$ (decision variable of Algorithm 1 and 3)
- $\boldsymbol{C}_t^{(a)} = \boldsymbol{A}_t^{(a)}\boldsymbol{W}_{t-1}$
- $\boldsymbol{F}_t = \boldsymbol{C}_t^{(1)} - \boldsymbol{C}_t^{(0)} = \widehat{\boldsymbol{S}}_t\boldsymbol{W}_{t-1}$
- $\boldsymbol{W}_t = \text{QR}(\boldsymbol{F}_t)$    (i.e., $\boldsymbol{F}_t \overset{\text{QR}}{=} \boldsymbol{W}_t\boldsymbol{R}_t$, $\boldsymbol{R}_t \in \mathbb{R}^{m \times m}$ is a lower triangular matrix)

Recall the assumptions on the data distribution, which we will assume throughout the proof:

**Assumption 1.** *Consider our data generation process $a \sim \text{Bernoulli}(p)$ and $\boldsymbol{x} \mid a \sim \mathcal{D}_a$. There exists $\sigma, V, \mathcal{M}, \mathcal{V}, g_{\min} > 0$, $f_{\max} \in (g_{\min}, \infty)$, and $p_{\min} \in (0, 0.5]$ such that the followings hold for any $(\mathcal{D}_0, \mathcal{D}_1, p) \in \mathcal{F}_d$: for $a \in \{0,1\}$, $\mathcal{D}_a \in \text{nSG}(\sigma)$,[4]*

$$\mathbb{P}\left[\|\boldsymbol{x}\boldsymbol{x}^\mathsf{T} - (\boldsymbol{\Sigma}_a + \boldsymbol{\mu}_a\boldsymbol{\mu}_a^\mathsf{T})\|_2 \leq \mathcal{M} \mid a\right] = 1, \quad \|\boldsymbol{\Sigma}_a + \boldsymbol{\mu}_a\boldsymbol{\mu}_a^\mathsf{T}\|_2 \leq V, \quad \text{Var}\left(\boldsymbol{x}\boldsymbol{x}^\mathsf{T} \mid a\right) \leq \mathcal{V},$$

*and*

$$\|\boldsymbol{g}\|_2 = \left\|\boldsymbol{\Pi}_{\boldsymbol{P}_m}^{\perp}\boldsymbol{f}\right\|_2 \in \{0\} \cup [g_{\min}, f_{\max}], \quad \|\boldsymbol{f}\|_2 \leq f_{\max}, \quad \text{and} \quad p \in [p_{\min}, 1 - p_{\min}].$$

**Assumption 2.** *Fix $m, k \in \mathbb{N}$. There exist $\Delta_{m,\nu}, \Delta_{k,\kappa}, K_{m,\nu}, K_{k,\kappa} \in (0, \infty)$ such that for any $(\mathcal{D}_0, \mathcal{D}_1, \cdot) \in \mathcal{F}_d$, the followings hold:*

$$\nu_m - \nu_{m+1} \geq \Delta_{m,\nu}, \quad \kappa_k - \kappa_{k+1} \geq \Delta_{k,\kappa}, \quad \nu_m \leq K_{m,\nu}, \quad \text{and} \quad \kappa_k \leq K_{k,\kappa},$$

*where $\nu_1 \geq \cdots \geq \nu_d \geq 0$ and $\kappa_1 \geq \cdots \geq \kappa_d \geq 0$ are the singular values of $\boldsymbol{S}$ and $\boldsymbol{\Pi}_{\bar{U}}^{\perp}\boldsymbol{\Sigma}\boldsymbol{\Pi}_{\bar{U}}^{\perp}$, respectively.*

---

[4]$\boldsymbol{y}$ is norm-sub-Gaussian, denoted as $\text{nSG}(\sigma)$, if $\mathbb{P}[\|\boldsymbol{y} - \mathbb{E}\boldsymbol{y}\| \geq t] \leq 2e^{-\frac{t^2}{2\sigma^2}}$. We refer interested readers to Jin et al. (2019, Section 2) for more discussions on norm-sub-Gaussianity.

We provide some intuition on the assumptions that we impose here. Assumption 1 consists of three parts. The first part, which involves $\sigma, V, \mathcal{M}$ and $\mathcal{V}$, ensures that the maximum deviation in mean and covariance of each $\mathcal{D}_a$ are well bounded; this is critical in allowing for us to use proper matrix concentration inequalities (to be described in the next subsection) and has been used in various streaming PCA literature (Bienstock et al., 2022; Huang et al., 2021; Jain et al., 2016). The second part, which involves $g_{\min}$ and $f_{\max}$, imposes a bound on the maximum mean separation, $\boldsymbol{f} = \boldsymbol{\mu}_1 - \boldsymbol{\mu}_0$, $\ell_2$-wise and angle-wise, respectively. If the mean difference can be arbitrarily large, then the $\ell_2$-estimation error that one has to achieve becomes arbitrarily small; precisely speaking, $\|\boldsymbol{f}\|_2$ acts as a Lipschitz constant. On the other hand, if the mean difference can be arbitrarily small, then the angle-wise estimation error becomes arbitrarily large. The last part, which involves $p_{\min}$, ensures that both groups are selected with some positive, nonvanishing probability. Assumption 2 is standard in streaming PCA literature to ensure convergence; indeed, if the singular value gap is zero, a definitive convergence result can never be obtained, as the ground-truth solution becomes vague.

## E.2 Matrix/Vector Concentration Inequalities

Before moving on to our proof, we review some useful concentration inequalities for our theoretical analysis.

**Definition 4** (Variance of random matrix)**.** *For a zero-mean random matrix $\boldsymbol{Z}$, its variance is defined as*
$$\mathrm{Var}(\boldsymbol{Z}) = \max\left(\left\|\mathbb{E}\left[\boldsymbol{Z}\boldsymbol{Z}^\mathsf{T}\right]\right\|_2, \left\|\mathbb{E}\left[\boldsymbol{Z}^\mathsf{T}\boldsymbol{Z}\right]\right\|_2\right).$$
*In general, $\mathrm{Var}(\boldsymbol{Z}) = \mathrm{Var}(\boldsymbol{Z} - \mathbb{E}[\boldsymbol{Z}])$.*

**Proposition 1** (Matrix Bernstein inequality (Theorem 6.6.1 of Tropp (2015)))**.** *Consider a finite collection $\{\boldsymbol{Y}_j\}_{j=1}^b$ of independent matrices with the same size ($d_1 \times d_2$). Suppose they are zero mean and they have uniformly bounded singular values, i.e.,*
$$\mathbb{E}[\boldsymbol{Y}_j] = \boldsymbol{0} \quad and \quad \mathbb{P}(\|\boldsymbol{Y}_j\| \leq \mathcal{M}) = 1 \quad for\ each\ j \in [b].$$
*Let $\mathcal{V}$ be an upper bound of matrix variance, $\mathcal{V} \geq \mathrm{Var}(\boldsymbol{Y}_j)$ for all $j \in [b]$. Then, for all $x \geq 0$,*
$$\mathbb{P}\left(\left\|\frac{1}{b}\sum_{j=1}^b \boldsymbol{Y}_j\right\| \geq x\right) \leq (d_1 + d_2)\exp\left(\frac{-bx^2}{2(\mathcal{V} + \mathcal{M}x/3)}\right).$$
*In particular, if $0 \leq x \leq \frac{3\mathcal{V}}{\mathcal{M}}$,*
$$\mathbb{P}\left(\left\|\frac{1}{b}\sum_{j=1}^b \boldsymbol{Y}_j\right\| \geq x\right) \leq (d_1 + d_2)\exp\left(\frac{-bx^2}{4\mathcal{V}}\right).$$

**Proposition 2** (Vector Hoeffding inequality (Corollary 7 of Jin et al. (2019)))**.** *There exists an absolute constant $\mathfrak{c}$ such that if $\boldsymbol{y}_1, \cdots, \boldsymbol{y}_b$ be independent random vectors with common dimension $d$, and assume the following:*
$$\mathbb{E}[\boldsymbol{y}_i] = 0, \quad \boldsymbol{y}_i \in \mathrm{nSG}(\sigma).$$
*Then, for any $x \geq 0$,*
$$\mathbb{P}\left(\left\|\frac{1}{b}\sum_{j=1}^b \boldsymbol{y}_j\right\|_2 \geq x\right) \leq 2d\exp\left(-\frac{bx^2}{\mathfrak{c}^2\sigma^2}\right).$$

**Remark 2.** *With an additional assumption that the random variables are bounded, we can also consider using vector Bernstein inequality (Lemma 18 of Kohler and Lucchi (2017)), which removes the factor of dimension $d$ from the RHS of the concentration inequality.*

## E.3 Proof of Lemma 2 - Bounding Error in Second Moment Gap

Recall that Algorithm 3 performs NPM on the second moment difference matrix $\boldsymbol{S}$ so that an iteration can be written as
$$\boldsymbol{W}_t = \mathrm{QR}(\widehat{\boldsymbol{S}}_t \boldsymbol{W}_{t-1}) = \mathrm{QR}(\boldsymbol{S}\boldsymbol{W}_{t-1} + \boldsymbol{Z}_{t,1}),$$
where $\boldsymbol{Z}_{t,1} = (\widehat{\boldsymbol{S}}_t - \boldsymbol{S})\boldsymbol{W}_{t-1}$ is the noise matrix.

The lemma below provides a general statement on sufficient size $b$ of blocks to bound the second moment difference error $\widehat{\boldsymbol{S}}_t - \boldsymbol{S}$ multiplied by some fixed matrices.

**Lemma 4.** *Let any $\delta > 0$ and matrices $\boldsymbol{M} \in \mathbb{R}^{m \times d}$ and $\boldsymbol{N} \in \mathbb{R}^{d \times n}$ be given. If we choose $b \geq \frac{4}{p_{\min}} \max\left(1, \frac{8\mathcal{M}^2}{9\mathcal{V}}\right) \log \frac{2(m+n)}{\delta}$, then with probability at least $1 - \delta$,*

$$\left\| \boldsymbol{M}(\widehat{\boldsymbol{S}}_t - \boldsymbol{S})\boldsymbol{N} \right\|_2 \leq \mathcal{E}_{m+n}^{(S)} \triangleq \sqrt{\frac{32\mathcal{V}}{bp_{\min}} \log \frac{2(m+n)}{\delta}}.$$

We now proceed the proof of Lemma 2. To this end, recall that $\boldsymbol{Z}_{t,1} = (\widehat{\boldsymbol{S}}_t - \boldsymbol{S})\boldsymbol{W}_{t-1}$. Firstly, applying Lemma 4 with $\boldsymbol{M} = \boldsymbol{I}_d$ and $\boldsymbol{N} = \boldsymbol{W}_{t-1} \in \mathbb{R}^{d \times m}$, a sufficient condition for having $5\|\boldsymbol{Z}_{t,1}\|_2 \leq \epsilon\Delta_{m,\nu}$ with probability at least $1 - \delta$ is that

$$b \geq \frac{400\mathcal{V}}{\epsilon^2 \Delta_{m,\nu}^2 p_{\min}} \log \frac{2(d+m)}{\delta} + \frac{4}{p_{\min}} \max\left(1, \frac{8\mathcal{M}^2}{9\mathcal{V}}\right) \log \frac{2(d+m)}{\delta}.$$

Secondly, applying Lemma 4 with $\boldsymbol{M} = \boldsymbol{P}_m^\intercal \in \mathbb{R}^{m \times d}$ and $\boldsymbol{N} = \boldsymbol{W}_{t-1} \in \mathbb{R}^{d \times m}$, a sufficient condition for having $5\|\boldsymbol{P}_m^\intercal \boldsymbol{Z}_{t,1}\|_2 \leq \frac{\delta\Delta_{m,\nu}}{2\sqrt{dm}}$ with probability at least $1 - \delta$ is that

$$b \geq \frac{3200dm\mathcal{V}}{\delta^2 \Delta_{m,\nu}^2 p_{\min}} \log \frac{4m}{\delta} + \frac{4}{p_{\min}} \max\left(1, \frac{8\mathcal{M}^2}{9\mathcal{V}}\right) \log \frac{4m}{\delta}.$$

Combining all bounds for $b$, we conclude that a sufficient block size $b$ is

$$b = \Omega\left(\frac{\mathcal{V}}{\Delta_{m,\nu}^2 p_{\min}} \left(\frac{dm}{\delta^2} \log \frac{m}{\delta} + \frac{1}{\epsilon^2} \log \frac{d}{\delta}\right) + \frac{\mathcal{M}^2}{\mathcal{V}p_{\min}} \log \frac{d}{\delta}\right).$$

*Proof of Lemma 4.* Note that $(a_i, \boldsymbol{x}_i)$'s are i.i.d. samples ($i \in \{(t-1)b+1, \ldots, tb\}$). Consider independent random matrices

$$\boldsymbol{Y}_i^{(a)} = \boldsymbol{M}\left(\boldsymbol{x}_i \boldsymbol{x}_i^\intercal - (\boldsymbol{\Sigma}_a + \boldsymbol{\mu}_a \boldsymbol{\mu}_a^\intercal)\right) \boldsymbol{N}.$$

For each set $A \subseteq [b]$, define an event $E_A := \{a_i = \mathbb{1}_{[i \in A]} \forall i \in [b]\}$. Note that $\mathbb{P}[E_A] = p_0^{b-|A|} p_1^{|A|}$. To exploit this, we first apply a peeling argument as follows: for any $c_0, c_1 \in (0,1)$ with $c_0 + c_1 = 1$,

$$\mathbb{P}\left(\left\| \boldsymbol{M}(\widehat{\boldsymbol{S}}_t - \boldsymbol{S})\boldsymbol{N} \right\|_2 \geq x\right)$$

$$= \sum_{r=0}^{b} \sum_{A \in \binom{[b]}{r}} \mathbb{P}\left(\left\| \boldsymbol{M}(\widehat{\boldsymbol{S}}_t - \boldsymbol{S})\boldsymbol{N} \right\|_2 \geq x \,\Big|\, E_A\right) \mathbb{P}(E_A)$$

$$= \sum_{r=1}^{b-1} \sum_{A \in \binom{[b]}{r}} \mathbb{P}\left(\left\| \frac{1}{r} \sum_{i \in A} \boldsymbol{Y}_i^{(1)} - \frac{1}{b-r} \sum_{i \in A^c} \boldsymbol{Y}_i^{(0)} \right\|_2 \geq x \,\Big|\, E_A\right) \mathbb{P}(E_A)$$

$$+ \mathbb{P}\left(\left\| \frac{1}{b} \sum_{i=1}^{b} \boldsymbol{Y}_i^{(0)} \right\|_2 \geq x \,\Big|\, E_\emptyset\right) \mathbb{P}(E_\emptyset) + \mathbb{P}\left(\left\| \frac{1}{b} \sum_{i=1}^{b} \boldsymbol{Y}_i^{(1)} \right\|_2 \geq x \,\Big|\, E_{[b]}\right) \mathbb{P}(E_{[b]})$$

$$\leq \sum_{r=1}^{b-1} \sum_{A \in \binom{[b]}{r}} \left\{ \mathbb{P}\left(\left\| \frac{1}{b-r} \sum_{i \in A^c} \boldsymbol{Y}_i^{(0)} \right\|_2 \geq c_0 x \,\Big|\, E_A\right) + \mathbb{P}\left(\left\| \frac{1}{r} \sum_{i \in A} \boldsymbol{Y}_i^{(1)} \right\|_2 \geq c_1 x \,\Big|\, E_A\right) \right\} p_0^{b-r} p_1^r$$

$$+ \mathbb{P}\left(\left\| \frac{1}{b} \sum_{i=1}^{b} \boldsymbol{Y}_i^{(0)} \right\|_2 \geq c_0 x \,\Big|\, E_\emptyset\right) p_0^b + \mathbb{P}\left(\left\| \frac{1}{b} \sum_{i=1}^{b} \boldsymbol{Y}_i^{(1)} \right\|_2 \geq c_1 x \,\Big|\, E_{[b]}\right) p_1^b,$$

where the last inequality holds for the following fact: if $\|\boldsymbol{a} - \boldsymbol{b}\| \geq x$ then $\|\boldsymbol{a}\| \geq \lambda x$ or $\|\boldsymbol{b}\| \geq (1 - \lambda)x$ for any $\lambda \in (0,1)$.

Now, we make the crucial observation that *conditioned* on the event $E_A$, both $\left\{\boldsymbol{Y}_i^{(1)}\right\}_{i \in A}$ and $\left\{\boldsymbol{Y}_i^{(0)}\right\}_{i \in A^c}$ are sets of i.i.d. random zero-mean $m \times n$ matrices. In addition, with Assumption 1, we have that for each $i \in A$,

$$\mathbb{P}\left[\left\| \boldsymbol{Y}_i^{(1)} \right\|_2 \leq \mathcal{M} \,\Big|\, i \in A\right] = 1, \quad \text{Var}\left(\boldsymbol{Y}_i^{(1)} \,\Big|\, i \in A\right) \leq \mathcal{V}.$$

(and analogously for each $i \in A^c$). Thus, applying the matrix Bernstein inequality[5] (Proposition 1), we obtain the following: when $0 \le x \le \frac{3\mathcal{V}}{\mathcal{M}}$,

$$\mathbb{P}\left(\left\|(\widehat{\boldsymbol{S}}_t - \boldsymbol{S})\boldsymbol{W}_{t-1}\right\|_2 \ge x\right)$$

$$\le (m+n)\sum_{r=1}^{b-1} p_0^{b-r} p_1^r \sum_{A \in \binom{[b]}{r}} \left\{\exp\left(-\frac{(b-r)c_0^2 x^2}{4\mathcal{V}}\right) + \exp\left(-\frac{rc_1^2 x^2}{4\mathcal{V}}\right)\right\}$$

$$+ (m+n)p_0^b \exp\left(-\frac{bc_0^2 x^2}{4\mathcal{V}}\right) + (m+n)p_1^b \exp\left(-\frac{bc_1^2 x^2}{4\mathcal{V}}\right)$$

$$\le (m+n)\sum_{r=0}^{b}\binom{b}{r} p_0^{b-r} p_1^r \left\{\exp\left(-\frac{(b-r)c_0^2 x^2}{4\mathcal{V}}\right) + \exp\left(-\frac{rc_1^2 x^2}{4\mathcal{V}}\right)\right\}$$

$$= (m+n)\left\{\left(p_0 + p_1\exp\left(-\frac{c_1^2 x^2}{4\mathcal{V}}\right)\right)^b + \left(p_1 + p_0\exp\left(-\frac{c_0^2 x^2}{4\mathcal{V}}\right)\right)^b\right\} < \delta.$$

For this to occur, it suffices for both terms to be bounded by $\frac{\delta}{2}$. Let us consider only the first term, as the second term follows from symmetry. Taking the log, we have

$$\log\left((1-p_1) + p_1\exp\left(-\frac{c_1^2 x^2}{4\mathcal{V}}\right)\right) \le \frac{1}{b}\log\frac{\delta}{2(m+n)}.$$

Using $\log(1+y) \le y$, it suffices to have

$$\frac{1}{b}\log\frac{2(m+n)}{\delta} \le p_1\left(1 - \exp\left(-\frac{c_1^2 x^2}{4\mathcal{V}}\right)\right).$$

Using $y \le 1 - e^{-2y}$ for $y \in [0, 1/2]$, it now suffices to have

$$\frac{8\mathcal{V}}{bp_1 c_1^2}\log\frac{2(m+n)}{\delta} \le x^2 \le \min\left(\frac{2\mathcal{V}}{c_1^2}, \frac{9\mathcal{V}^2}{\mathcal{M}^2}\right).$$

Combining this with the other term and, for simplicity[6] choosing $c_0 = c_1 = 1/2$, we have our desired statement. $\qquad\square$

### E.4 Proof of Lemma 3 - Bounding the Final Error

Recall that the iterates are $\boldsymbol{V}_{\tau+1} = \texttt{QR}(\boldsymbol{\Pi}_{\widehat{\boldsymbol{U}}}^{\perp}\widehat{\boldsymbol{\Sigma}}_\tau\boldsymbol{\Pi}_{\widehat{\boldsymbol{U}}}^{\perp}\boldsymbol{V}_\tau)$, where $\widehat{\boldsymbol{U}}$ is the output of Algorithm 3, $\boldsymbol{\Pi}_{\widehat{\boldsymbol{U}}}^{\perp} := \boldsymbol{I}_d - \widehat{\boldsymbol{U}}\widehat{\boldsymbol{U}}^{\intercal}$, and $\widehat{\boldsymbol{\Sigma}}_\tau := \frac{1}{\mathcal{B}}\sum_{j=(\tau-1)\mathcal{B}+1}^{\tau\mathcal{B}}\tilde{\boldsymbol{x}}_j\tilde{\boldsymbol{x}}_j^{\intercal}$ is the sample covariance at time step $\tau$ of Algorithm 4. This can be rewritten as

$$\boldsymbol{V}_{\tau+1} = \texttt{QR}(\boldsymbol{\Pi}_{\boldsymbol{U}}^{\perp}\boldsymbol{\Sigma}\boldsymbol{\Pi}_{\boldsymbol{U}}^{\perp}\boldsymbol{V}_\tau + \boldsymbol{Z}_{\tau,2}),$$

where $\boldsymbol{Z}_{\tau,2} = (\boldsymbol{\Pi}_{\widehat{\boldsymbol{U}}}^{\perp}\widehat{\boldsymbol{\Sigma}}_\tau\boldsymbol{\Pi}_{\widehat{\boldsymbol{U}}}^{\perp} - \boldsymbol{\Pi}_{\boldsymbol{U}}^{\perp}\boldsymbol{\Sigma}\boldsymbol{\Pi}_{\boldsymbol{U}}^{\perp})\boldsymbol{V}_\tau$ is the noise matrix and $\boldsymbol{\Sigma} = \sum_{a \in \{0,1\}} p_a(\boldsymbol{\Sigma}_a + \boldsymbol{\mu}_a\boldsymbol{\mu}_a^{\intercal})$. By Assumption 1, we have that $\|\boldsymbol{\Sigma}\|_2 \le V$.

The lemma below provides a general statement on a sufficient block size $\mathcal{B}$ to bound the covariance error $\widehat{\boldsymbol{\Sigma}} - \boldsymbol{\Sigma}$ multiplied by some fixed matrices.

**Lemma 5.** *Let any $\delta > 0$ and matrices $\boldsymbol{M} \in \mathbb{R}^{m \times d}$ and $\boldsymbol{N} \in \mathbb{R}^{d \times n}$ be given. If we choose $\mathcal{B} \ge \frac{4}{9}\frac{\mathcal{M}^2}{\mathcal{V}+V^2}\log\frac{4(m+n)}{\delta}$, then the following holds with probability at least $1 - \delta$:*

$$\left\|\boldsymbol{M}(\widehat{\boldsymbol{\Sigma}} - \boldsymbol{\Sigma})\boldsymbol{N}\right\|_2 \le \mathcal{E}_{m+n}^{(\Sigma)} \triangleq \sqrt{\frac{4(\mathcal{V}+V^2)}{\mathcal{B}}\log\frac{4(m+n)}{\delta}}.$$

---

[5]Precisely, we use its conditional version where the means and the variances are replaced with their conditional counterparts.

[6]One could try to optimize for $c_0, c_1$ to obtain an "optimal" sample complexities. However, from some preliminary computations, this seems to be not worth pursuing, and we conjecture that it would yield the same asymptotic dependency as when $c_0 = c_1 = 1/2$.

We now proceed the proof of Lemma 3. Fix some $V \in \text{St}(d, k)$, and from the design of our Algorithm 4, $\widehat{U}$ is also fixed. By our given assumption on upper bound of $\left\|\Pi_{\widehat{U}} - \Pi_U\right\|_2$, and the fact that $\|V\|_2 = \|Q_k\|_2 = 1$, we have

$$\|Z_{\tau,2}\|_2 \leq \left\|\Pi_{\widehat{U}}^{\perp}(\widehat{\Sigma} - \Sigma)\Pi_{\widehat{U}}^{\perp}V\right\|_2 + 2\|\Sigma\|_2\|\Pi_{\widehat{U}} - \Pi_U\|_2$$

$$\leq \left\|\Pi_{\widehat{U}}^{\perp}(\widehat{\Sigma} - \Sigma)\Pi_{\widehat{U}}^{\perp}V\right\|_2 + \frac{\epsilon\Delta_{k,\kappa}}{10}$$

and, with a similar logic,

$$\|Q_k^{\mathsf{T}}Z_{\tau,2}\|_2 \leq \left\|Q_k^{\mathsf{T}}\Pi_{\widehat{U}}^{\perp}(\widehat{\Sigma} - \Sigma)\Pi_{\widehat{U}}^{\perp}V\right\|_2 + \frac{\delta\Delta_{k,\kappa}}{20\sqrt{dk}}.$$

Applying Lemma 5 with $M = \Pi_{\widehat{U}}^{\perp} \in \mathbb{R}^{d \times d}$ and $N = \Pi_{\widehat{U}}^{\perp}V \in \mathbb{R}^{d \times k}$, a sufficient condition for having $\left\|\Pi_{\widehat{U}}^{\perp}(\widehat{\Sigma} - \Sigma)\Pi_{\widehat{U}}^{\perp}V\right\|_2 \leq \frac{\epsilon\Delta_{k,\kappa}}{10}$ with probability at least $1 - \delta$ is that

$$\mathcal{B} \geq \frac{400(\mathcal{V} + V^2)}{\epsilon^2\Delta_{k,\kappa}^2}\log\frac{4(d+k)}{\delta} + \frac{4}{9}\frac{\mathcal{M}^2}{\mathcal{V} + V^2}\log\frac{4(d+k)}{\delta}.$$

Furthermore, applying Lemma 5 with $M = Q_k^{\mathsf{T}}\Pi_{\widehat{U}}^{\perp} \in \mathbb{R}^{k \times d}$ and $N = \Pi_{\widehat{U}}^{\perp}V \in \mathbb{R}^{d \times k}$, a sufficient condition for having $\left\|Q_k^{\mathsf{T}}\Pi_{\widehat{U}}^{\perp}(\widehat{\Sigma} - \Sigma)\Pi_{\widehat{U}}^{\perp}V\right\|_2 \leq \frac{\delta\Delta_{k,\kappa}}{20\sqrt{dk}}$ with probability at least $1 - \delta$ is that

$$\mathcal{B} \geq \frac{1600dk(\mathcal{V} + V^2)}{\delta^2\Delta_{k,\kappa}^2}\log\frac{8k}{\delta} + \frac{4}{9}\frac{\mathcal{M}^2}{\mathcal{V} + V^2}\log\frac{8k}{\delta}.$$

Combining all bounds for $\mathcal{B}$, we conclude that a sufficient block size $\mathcal{B}$ to have our desired statement is

$$\mathcal{B} = \Omega\left(\frac{\mathcal{V} + V^2}{\Delta_{k,\kappa}^2}\left(\frac{dk}{\delta^2}\log\frac{k}{\delta} + \frac{1}{\epsilon^2}\log\frac{d}{\delta}\right) + \frac{\mathcal{M}^2}{\mathcal{V} + V^2}\log\frac{d}{\delta}\right).$$

*Proof of Lemma 5.* Note that $\tilde{x}_j$'s are i.i.d. samples from $\mathcal{D}$ ($j \in \{(\tau-1)\mathcal{B} + 1, \ldots, \tau\mathcal{B}\}$). Let

$$Y_j = M\left(\tilde{x}_j\tilde{x}_j^{\mathsf{T}} - \Sigma\right)N.$$

Then, $Y_i$ are i.i.d. random zero-mean $d \times k$ matrices across $j$. In addition, with Assumption 1,

$$\mathbb{P}\left[\|\tilde{x}\tilde{x}^{\mathsf{T}} - \Sigma\|_2 \leq \mathcal{M}\right] = \sum_{a' \in \{0,1\}} \mathbb{P}\left[\|\tilde{x}\tilde{x}^{\mathsf{T}} - \Sigma\|_2 \leq \mathcal{M} \mid a = a'\right]\mathbb{P}[a = a'] = 1,$$

and

$$\text{Var}(\tilde{x}\tilde{x}^{\mathsf{T}}) = \left\|\mathbb{E}\left[\mathbb{E}[(\tilde{x}\tilde{x}^{\mathsf{T}})^2|a]\right] - \mathbb{E}\left[\mathbb{E}[\tilde{x}\tilde{x}^{\mathsf{T}}|a]^2\right]\right\|_2$$

$$\leq \left\|\mathbb{E}\left[\mathbb{E}\left[(\tilde{x}\tilde{x}^{\mathsf{T}})^2|a\right] - \mathbb{E}[\tilde{x}\tilde{x}^{\mathsf{T}}|a]^2\right]\right\|_2 + \left\|\mathbb{E}\left[\mathbb{E}[\tilde{x}\tilde{x}^{\mathsf{T}}|a]^2\right] - \mathbb{E}[\mathbb{E}[\tilde{x}\tilde{x}^{\mathsf{T}}|a]]^2\right\|_2$$

$$\leq \sum_{a \in [0,1]} p_a \underbrace{\left\|\mathbb{E}[(\tilde{x}\tilde{x}^{\mathsf{T}})^2|a] - \mathbb{E}[\tilde{x}\tilde{x}^{\mathsf{T}}|a]^2\right\|_2}_{\text{Var}(\tilde{x}\tilde{x}^{\mathsf{T}}|a)} + \left\|\mathbb{E}\left[(\Sigma_a + \mu_a\mu_a^{\mathsf{T}})^2\right] - \mathbb{E}\left[\Sigma_a + \mu_a\mu_a^{\mathsf{T}}\right]^2\right\|_2$$

$$\leq \mathcal{V} + p_0p_1\left\|\Sigma_0 + \mu_0\mu_0^{\mathsf{T}} - \Sigma_1 + \mu_1\mu_1^{\mathsf{T}}\right\|_2^2$$

$$\leq \mathcal{V} + 4p_0(1 - p_0)V^2$$

$$\leq \mathcal{V} + V^2.$$

Applying the matrix Bernstein inequality (Proposition 1), we obtain the following: when $0 \leq x \leq \frac{3(\mathcal{V}+V^2)}{\mathcal{M}}$,

$$\mathbb{P}\left(\left\|\frac{1}{\mathcal{B}}\sum_{i=1}^{\mathcal{B}} Y_i\right\|_2 \geq x\right) \leq (m+n)\exp\left(-\frac{\mathcal{B}x^2}{4(\mathcal{V} + V^2)}\right) \leq \frac{\delta}{4}.$$

Solving for $x$, we have

$$\frac{4(\mathcal{V} + V^2)}{\mathcal{B}}\log\frac{4(m+n)}{\delta} \leq x^2 \leq 9\left(\frac{(\mathcal{V} + V^2)}{\mathcal{M}}\right)^2,$$

from which the statement naturally follows. $\qquad\square$

## E.5 Bounding the Estimation Error of Mean Difference

**Lemma 6.** *For any $\delta > 0$, choose $b \geq \frac{2}{Tp_{\min}} \log \frac{4d}{\delta}$. Then, the following holds with probability at least $1 - \delta$:*

$$\left\| \widehat{f} - f \right\|_2 \leq \mathcal{E}^{(\mathrm{f})} \triangleq \sqrt{\frac{8\mathfrak{c}^2 \sigma^2}{Tbp_{\min}} \log \frac{4d}{\delta}},$$

*where $\mathfrak{c}$*

*Proof.* Again, note that $(a_i, \boldsymbol{x}_i)$'s are i.i.d. samples ($i \in [Tb]$). Consider independent random vectors

$$\boldsymbol{y}_i^{(a)} = \boldsymbol{x}_i - \boldsymbol{\mu}_a.$$

For each $A \subseteq [Tb]$, $E_A = \{a_i = \mathbb{1}[i \in A] \ \forall i \in [b]\}$ is an event satisfying $\mathbb{P}(E_A) = p_0^{Tb - |A|} p_1^{|A|}$. To exploit this, we now apply the peeling argument as follows: for any $c_0, c_1 \in (0, 1)$ with $c_0 + c_1 = 1$,

$$\mathbb{P}\left( \left\| \widehat{f} - f \right\|_2 \geq x \right)$$

$$= \sum_{r=0}^{Tb} \sum_{A \in \binom{[Tb]}{r}} \mathbb{P}\left( \left\| \widehat{f} - f \right\|_2 \geq x \,\middle|\, E_A \right) \mathbb{P}(E_A)$$

$$= \sum_{r=1}^{Tb-1} \sum_{A \in \binom{[Tb]}{r}} \mathbb{P}\left( \left\| \frac{1}{r} \sum_{i \in A} \boldsymbol{y}_i^{(1)} - \frac{1}{Tb - r} \sum_{i \in A^c} \boldsymbol{y}_i^{(0)} \right\|_2 \geq x \,\middle|\, E_A \right) \mathbb{P}(E_A)$$

$$+ \mathbb{P}\left( \left\| \frac{1}{Tb} \sum_{i=1}^{Tb} \boldsymbol{y}_i^{(0)} \right\|_2 \geq x \,\middle|\, E_\emptyset \right) \mathbb{P}(E_\emptyset) + \mathbb{P}\left( \left\| \frac{1}{Tb} \sum_{i=1}^{Tb} \boldsymbol{y}_i^{(1)} \right\|_2 \geq x \,\middle|\, E_{[Tb]} \right) \mathbb{P}(E_{[Tb]})$$

$$\leq \sum_{r=1}^{Tb-1} \sum_{A \in \binom{[Tb]}{r}} \left\{ \mathbb{P}\left( \left\| \frac{1}{Tb - r} \sum_{i \in A^c} \boldsymbol{y}_i^{(0)} \right\|_2 \geq c_0 x \,\middle|\, E_A \right) + \mathbb{P}\left( \left\| \frac{1}{r} \sum_{i \in A} \boldsymbol{y}_i^{(1)} \right\|_2 \geq c_1 x \,\middle|\, E_A \right) \right\} p_0^{Tb - r} p_1^r$$

$$+ \mathbb{P}\left( \left\| \frac{1}{Tb} \sum_{i=1}^{Tb} \boldsymbol{y}_i^{(0)} \right\|_2 \geq x \,\middle|\, E_\emptyset \right) p_0^{Tb} + \mathbb{P}\left( \left\| \frac{1}{Tb} \sum_{i=1}^{Tb} \boldsymbol{y}_i^{(1)} \right\|_2 \geq x \,\middle|\, E_{[Tb]} \right) p_1^{Tb}$$

Observe that *conditioned* on the event $E_A$, both $\left\{ \boldsymbol{y}_i^{(1)} \right\}_{i \in A}$ and $\left\{ \boldsymbol{y}_i^{(0)} \right\}_{i \in A^c}$ are sets of i.i.d. random zero-mean $d$-dimensional vectors. In addition, with Assumption 1, we have that for each $i \in A$, $\boldsymbol{y}_i^{(1)} \in \mathrm{nSG}(\sigma_1)$ (and analogously for each $i \in A^c$). Thus, applying the vector Bernstein inequality (Proposition 2), we obtain the following: for $x > 0$,

$$\mathbb{P}\left( \left\| \widehat{f} - f \right\|_2 \geq x \right) \leq 2d \sum_{r=1}^{Tb-1} p_0^{Tb - r} p_1^r \sum_{A \in \binom{[Tb]}{r}} \left\{ \exp\left( -\frac{(Tb - r)c_0^2 x^2}{\mathfrak{c}^2 \sigma_0^2} \right) + \exp\left( -\frac{rc_1^2 x^2}{\mathfrak{c}^2 \sigma_1^2} \right) \right\}$$

$$+ 2dp_0^{Tb} \exp\left( -\frac{Tbx^2}{\mathfrak{c}^2 \sigma_0^2} \right) + 2dp_1^{Tb} \exp\left( -\frac{Tbx^2}{\mathfrak{c}^2 \sigma_1^2} \right)$$

$$\leq 2d \sum_{r=0}^{Tb} \binom{Tb}{r} p_0^{Tb - r} p_1^r \left\{ \exp\left( -\frac{(Tb - r)c_0^2 x^2}{\mathfrak{c}^2 \sigma_0^2} \right) + \exp\left( -\frac{rc_1^2 x^2}{\mathfrak{c}^2 \sigma_1^2} \right) \right\}$$

$$= 2d \left( p_0 + p_1 \exp\left( -\frac{c_1^2 x^2}{\mathfrak{c}^2 \sigma_1^2} \right) \right)^{Tb} + 2d \left( p_1 + p_0 \exp\left( -\frac{c_0^2 x^2}{\mathfrak{c}^2 \sigma_1^2} \right) \right)^{Tb} < \delta.$$

It suffices for each of both terms to be bounded by $\frac{\delta}{2}$. Let us consider only the first term, as the second term follows from symmetry. Taking the log, we have

$$\log\left( (1 - p_1) + p_1 \exp\left( -\frac{c_1^2 x^2}{\mathfrak{c}^2 \sigma_1^2} \right) \right) \leq \frac{1}{Tb} \log \frac{\delta}{4d}.$$

Using $\log(1 + y) \le y$, it suffices to have

$$\frac{1}{Tb} \log \frac{4d}{\delta} \le p_1 \left( 1 - \exp\left( -\frac{c_1^2 x^2}{\mathfrak{c}^2 \sigma_1^2} \right) \right).$$

Using $y \le 1 - e^{-2y}$ for $y \le 1/2$, it now suffices to have

$$\frac{2\mathfrak{c}^2 \sigma_1^2}{Tb p_1 c_1^2} \log \frac{4d}{\delta} \le x^2 \le \frac{\mathfrak{c}^2 \sigma_1^2}{c_1^2}.$$

Combining this with the other term and, for simplicity (see the footnote on pg. 25), choosing $c_0 = c_1 = 1/2$, we have our desired statement. $\qquad\square$

### E.6 Proof of Theorem 1 - Sample Complexity for PAFO-learnability

Let us fix $\varepsilon_{\mathrm{f}}, \varepsilon_{\mathrm{o}}, \delta \in (0,1)$. From the definition of PAFO-learnability (Definition 2), with a large enough sample size, we must guarantee the following with probability at least $1 - \delta$:

$$\mathrm{tr}\left( V^{\mathsf{T}} \Sigma V \right) \ge \mathrm{tr}\left( V^{\star \mathsf{T}} \Sigma V^\star \right) - \varepsilon_{\mathrm{f}}, \quad \| \Pi_U V \|_2 \le \varepsilon_{\mathrm{o}}.$$

Let $\widehat{U}$ be the final estimate of the unfair subspace $U$ from Algorithm 1, and let $V$ be the final estimate of $Q_k = V^\star$, whose columns are top $k$ eigenvectors of $\Pi_{\widehat{U}}^{\perp} \Sigma \Pi_{\widehat{U}}^{\perp}$.

We first recall the von Neumann trace inequality (Mirsky, 1975; von Neumann, 1937):

**Proposition 3** (Theorem H.1.g of Marshall et al. (2011)). *If $A, B$ are two $n \times n$ Hermitian matrices, then*

$$\mathrm{tr}(AB) \le \sum_{i=1}^{n} \lambda_i(A) \lambda_i(B),$$

*where $\lambda_i(\cdot)$ is the $i$-th smallest eigenvalue.*

Also, the following lemma implies that the sine angle between subspaces (with the same dimension) and the corresponding Grassmannian (projection) distance are the same.

**Lemma 7.** *Let $A_1 \in \mathbb{R}^{d \times k}$ and $B_1 \in \mathbb{R}^{d \times \ell}$ be semi-orthogonal matrices: $A_1^{\mathsf{T}} A_1 = I_k$ and $B_1^{\mathsf{T}} B_1 = I_\ell$. Define*

$$\mathrm{dist}(\mathrm{col}(A_1), \mathrm{col}(B_1)) \triangleq \left\| \Pi_{A_1} - \Pi_{B_1} \right\|_2,$$
$$\sin\left( \mathrm{col}(A_1), \mathrm{col}(B_1) \right) \triangleq \left\| \Pi_{A_1}^{\perp} B_1 \right\|_2.$$

*Then, (i) if $k = \ell$,*

$$\mathrm{dist}(\mathrm{col}(A_1), \mathrm{col}(B_1)) = \sin\left( \mathrm{col}(A_1), \mathrm{col}(B_1) \right)$$
$$= \sin\left( \mathrm{col}(B_1), \mathrm{col}(A_1) \right) = \sqrt{1 - \sigma_{\min}(A_1^{\mathsf{T}} B_1)^2},$$

*where $\sigma_{\min}(M)$ is the minimum singular value of $M$. On the other hand, (ii) if $k < \ell$,*

$$\mathrm{dist}(\mathrm{col}(A_1), \mathrm{col}(B_1)) = \sin\left( \mathrm{col}(A_1), \mathrm{col}(B_1) \right) = 1$$
$$\ge \sin\left( \mathrm{col}(B_1), \mathrm{col}(A_1) \right) = \sqrt{1 - \sigma_{\min}(A_1^{\mathsf{T}} B_1)^2}.$$

Now we can write the optimality (the first inequality) as follows:

$$\mathrm{tr}\left( Q_k^{\mathsf{T}} \Sigma Q_k \right) - \mathrm{tr}\left( V^{\mathsf{T}} \Sigma V \right) = \mathrm{tr}\left( \Sigma (Q_k Q_k^{\mathsf{T}} - V V^{\mathsf{T}}) \right)$$

$$\overset{(a)}{\le} \sum_{i=1}^{d} \lambda_i(\Sigma) \lambda_i(Q_k Q_k^{\mathsf{T}} - V V^{\mathsf{T}})$$

$$\overset{(b)}{\le} 2k \| \Sigma \|_2 \| Q_k Q_k^{\mathsf{T}} - V V^{\mathsf{T}} \|_2$$

$$\overset{(c)}{\le} 2kV \| \Pi_V^{\perp} Q_k \|_2 \le \varepsilon_{\mathrm{f}}.$$

Here, $(a)$ follows from the von Neumann trace inequality, $(b)$ follows from the fact that $\boldsymbol{Q}_k\boldsymbol{Q}_k^\mathsf{T}-\boldsymbol{V}\boldsymbol{V}^\mathsf{T}$ is a symmetric matrix of rank at most $2k$, and $(c)$ follows from Lemma 7. Thus, for the optimality, it suffices to ensure that $\|\boldsymbol{\Pi}_{\boldsymbol{V}}^\perp\boldsymbol{Q}_k\|_2 \leq \frac{\varepsilon_\mathrm{f}}{2kV}$.

Similarly, for the fairness (the second inequality), it suffices to ensure that $\|\boldsymbol{\Pi}_{\boldsymbol{U}}\boldsymbol{\Pi}_{\widehat{U}}^\perp\|_2 \leq \varepsilon_\mathrm{o}$ or $\|\boldsymbol{\Pi}_{\widehat{U}} - \boldsymbol{\Pi}_{\boldsymbol{U}}\|_2 \leq \varepsilon_\mathrm{o}$, as

$$\|\boldsymbol{\Pi}_{\boldsymbol{U}}\boldsymbol{V}\|_2 \overset{(*)}{=} \left\|\boldsymbol{\Pi}_{\boldsymbol{U}}\boldsymbol{\Pi}_{\widehat{U}}^\perp\boldsymbol{V}\right\|_2 \leq \left\|\boldsymbol{\Pi}_{\boldsymbol{U}}\boldsymbol{\Pi}_{\widehat{U}}^\perp\right\|_2 \overset{(\star)}{\leq} \left\|\boldsymbol{\Pi}_{\widehat{U}} - \boldsymbol{\Pi}_{\boldsymbol{U}}\right\|_2 \leq \varepsilon_\mathrm{o},$$

where $(*)$ follows from $\boldsymbol{V} = \boldsymbol{\Pi}_{\widehat{U}}^\perp\boldsymbol{V}$ (due to the design of Algorithm 4) and $(\star)$ follows from Lemma 7.

From Lemma 1 (letting $\mathcal{T} = \Theta\left(\frac{K_{k,\kappa}}{\Delta_{k,\kappa}}\log\frac{d}{\epsilon\delta}\right)$) and 3 (substituting $\epsilon \mapsto \frac{\varepsilon_\mathrm{f}}{2kV}$), *given* that

$$\left\|\boldsymbol{\Pi}_{\widehat{U}} - \boldsymbol{\Pi}_{\boldsymbol{U}}\right\|_2 \leq \eta_k = \eta_k(\varepsilon_\mathrm{f}, \varepsilon_\mathrm{o}, \delta) \triangleq \min\left(\varepsilon_\mathrm{o}, \frac{\Delta_{k,\kappa}\varepsilon_\mathrm{f}}{40V^2k}, \frac{\Delta_{k,\kappa}\delta}{160V\sqrt{dk}}\right), \tag{17}$$

a total of $N_2 = \mathcal{T}\mathcal{B}$ samples are sufficient in Algorithm 4 for ensuring $\varepsilon_\mathrm{f}$-optimality with probability at least $1 - \frac{\delta}{2}$, where

$$N_2 \gtrsim \left(\frac{K_{k,\kappa}(\mathcal{V} + V^2)}{\Delta_{k,\kappa}^3}\left(\frac{dk}{\delta^2}\log\frac{k}{\delta} + \frac{k^2V^2}{\varepsilon_\mathrm{f}^2}\log\frac{d}{\delta}\right) + \frac{K_{k,\kappa}\mathcal{M}^2}{\Delta_{k,\kappa}(\mathcal{V} + V^2)}\log\frac{d}{\delta}\right)\log\frac{dkV}{\varepsilon_\mathrm{f}\delta}. \tag{18}$$

We now focus on obtaining the sample complexity for Algorithm 3 to satisfy Eqn. (17) with probability at least $1 - \frac{\delta}{2}$. Then combining those with a union bound gives us the desired statement.

From hereon and forth, we write

$$\boldsymbol{g} \triangleq \boldsymbol{\Pi}_{\boldsymbol{P}_m}^\perp\boldsymbol{f}, \quad \boldsymbol{h} \triangleq \frac{1}{\|\boldsymbol{g}\|_2}\boldsymbol{g},$$

$$\widehat{\boldsymbol{g}} \triangleq \boldsymbol{\Pi}_{\boldsymbol{W}_T}^\perp\widehat{\boldsymbol{f}}, \quad \widehat{\boldsymbol{h}} \triangleq \frac{1}{\|\widehat{\boldsymbol{g}}\|_2}\widehat{\boldsymbol{g}}.$$

Also, we introduce the following lemma that provides a general bound of sine angle between a pair of vectors with a small $\ell_2$-distance.

**Lemma 8.** *Consider two vectors $\boldsymbol{a}, \boldsymbol{b} \in \mathbb{R}^d$. If we denote the (acute) angle between $\boldsymbol{a}$ and $\boldsymbol{b}$ as $\theta(\boldsymbol{a}, \boldsymbol{b})$,*

$$\sin\theta(\boldsymbol{a}, \boldsymbol{b}) = \sin(\mathrm{span}(\boldsymbol{a}), \mathrm{span}(\boldsymbol{b})) = \left\|\frac{1}{\|\boldsymbol{a}\|_2^2}\boldsymbol{a}\boldsymbol{a}^\mathsf{T} - \frac{1}{\|\boldsymbol{b}\|_2^2}\boldsymbol{b}\boldsymbol{b}^\mathsf{T}\right\|_2.$$

*Consider any $\epsilon \in \left(0, \frac{\|\boldsymbol{a}\|_2}{2}\right]$ and suppose $\|\boldsymbol{a} - \boldsymbol{b}\|_2 \leq \epsilon$. Then,*

$$\sin\theta(\boldsymbol{a}, \boldsymbol{b}) \leq \frac{\sqrt{2}\epsilon}{\|\boldsymbol{a}\|_2}.$$

**Case 1.** Assume $\boldsymbol{g} \neq \boldsymbol{0}$. In this case, $\boldsymbol{U} = [\boldsymbol{P}_m|\boldsymbol{h}]$, $\|\boldsymbol{g}\|_2 \geq g_{\min}$, and $\|\boldsymbol{f}\|_2 \leq f_{\max}$ by Assumption 1. We want to upper-bound the following probability to be $< \delta/2$:

$$\mathbb{P}\left[\left\|\boldsymbol{\Pi}_{\boldsymbol{U}}\boldsymbol{\Pi}_{\widehat{U}}^\perp\right\|_2 > \eta_k\right]$$

$$= \mathbb{P}\left[\left(\widehat{\boldsymbol{g}} \neq \boldsymbol{0} \text{ and } \left\|\boldsymbol{\Pi}_{\boldsymbol{U}} - \boldsymbol{\Pi}_{\widehat{U}}\right\|_2 > \eta_k\right) \text{ or } \left(\widehat{\boldsymbol{g}} = \boldsymbol{0} \text{ and } \left\|\boldsymbol{\Pi}_{\boldsymbol{U}}\boldsymbol{\Pi}_{\widehat{U}}^\perp\right\|_2 > \eta_k\right)\right]$$

$$\leq \mathbb{P}\left[\left\|\boldsymbol{\Pi}_{[\boldsymbol{P}_m|\boldsymbol{h}]} - \boldsymbol{\Pi}_{[\boldsymbol{W}_T|\widehat{\boldsymbol{h}}]}\right\|_2 > \eta_k \,\middle|\, \widehat{\boldsymbol{g}} \neq \boldsymbol{0}\right] + \mathbb{P}\left[\widehat{\boldsymbol{g}} = \boldsymbol{0}\right].$$

Let us obtain a sample complexity to bound the first probability term to be $< \frac{\delta}{4}$. Note that

$$\left\|\boldsymbol{\Pi}_{[\boldsymbol{P}_m|\boldsymbol{h}]} - \boldsymbol{\Pi}_{[\boldsymbol{W}_T|\widehat{\boldsymbol{h}}]}\right\|_2 \leq \|\boldsymbol{\Pi}_{\boldsymbol{P}_m} - \boldsymbol{\Pi}_{\boldsymbol{W}_T}\|_2 + \left\|\boldsymbol{h}\boldsymbol{h}^\mathsf{T} - \widehat{\boldsymbol{h}}\widehat{\boldsymbol{h}}^\mathsf{T}\right\|_2.$$

Firstly, to yield

$$\left\|\mathbf{\Pi}_{\boldsymbol{P}_m} - \mathbf{\Pi}_{\boldsymbol{W}_T}\right\|_2 = \left\|\mathbf{\Pi}_{\boldsymbol{W}_T}^{\perp}\boldsymbol{P}_m\right\|_2 \leq \frac{\eta_k}{2}$$

with probability at least $1 - \frac{\delta}{12}$, it is sufficient that

$$T = \Theta\left(\frac{K_{m,\nu}}{\Delta_{m,\nu}}\log\left(\frac{d}{\eta_k\delta}\right)\right), \tag{19}$$

$$b = \Omega\left(\frac{\mathcal{V}}{\Delta_{m,\nu}^2 p_{\min}}\left(\frac{dm}{\delta^2}\log\frac{m}{\delta} + \frac{1}{\eta_k^2}\log\frac{d}{\delta}\right) + \frac{\mathcal{M}^2}{\mathcal{V}p_{\min}}\log\frac{d}{\delta}\right), \tag{20}$$

due to application of Lemma 1 and 2.

Secondly, we aim to make the second term $\left\|\boldsymbol{h}\boldsymbol{h}^{\intercal} - \widehat{\boldsymbol{h}}\widehat{\boldsymbol{h}}^{\intercal}\right\|_2$ small with high probability. We first claim that, if $\|\boldsymbol{g} - \widehat{\boldsymbol{g}}\|_2 \leq \frac{g_{\min}}{2}\min\left\{\frac{\eta_k}{\sqrt{2}}, 1\right\}$, then $\left\|\boldsymbol{h}\boldsymbol{h}^{\intercal} - \widehat{\boldsymbol{h}}\widehat{\boldsymbol{h}}^{\intercal}\right\|_2 \leq \frac{\eta_k}{2}$. Here is the proof of the claim: Since $\|\boldsymbol{g} - \widehat{\boldsymbol{g}}\|_2 \leq \frac{\|\boldsymbol{g}\|_2}{2}$, we have

$$\left\|\boldsymbol{h}\boldsymbol{h}^{\intercal} - \widehat{\boldsymbol{h}}\widehat{\boldsymbol{h}}^{\intercal}\right\|_2 \overset{\text{Lem. 8}}{\leq} \frac{\sqrt{2}}{\|\boldsymbol{g}\|_2}\|\boldsymbol{g} - \widehat{\boldsymbol{g}}\|_2 \leq \frac{\sqrt{2}}{g_{\min}}\|\boldsymbol{g} - \widehat{\boldsymbol{g}}\|_2 \leq \frac{\eta_k}{2}.$$

Thus, since

$$\|\boldsymbol{g} - \widehat{\boldsymbol{g}}\|_2 = \left\|\left(\mathbf{\Pi}_{\boldsymbol{P}_m}^{\perp} - \mathbf{\Pi}_{\boldsymbol{W}_T}^{\perp}\right)\boldsymbol{f} + \mathbf{\Pi}_{\boldsymbol{W}_T}^{\perp}(\boldsymbol{f} - \widehat{\boldsymbol{f}})\right\|_2$$

$$\leq \left\|\mathbf{\Pi}_{\boldsymbol{P}_m} - \mathbf{\Pi}_{\boldsymbol{W}_T}\right\|_2 f_{\max} + \left\|\boldsymbol{f} - \widehat{\boldsymbol{f}}\right\|_2, \tag{21}$$

we have

$$\mathbb{P}\left[\left\|\boldsymbol{h}\boldsymbol{h}^{\intercal} - \widehat{\boldsymbol{h}}\widehat{\boldsymbol{h}}^{\intercal}\right\|_2 > \frac{\eta_k}{2}\,\middle|\,\widehat{\boldsymbol{g}} \neq \boldsymbol{0}\right]$$

$$\leq \mathbb{P}\left[\|\boldsymbol{g} - \widehat{\boldsymbol{g}}\|_2 > \frac{g_{\min}}{2}\min\left\{\frac{\eta_k}{\sqrt{2}}, 1\right\}\,\middle|\,\widehat{\boldsymbol{g}} \neq \boldsymbol{0}\right]$$

$$\leq \mathbb{P}\left[\left\|\mathbf{\Pi}_{\boldsymbol{P}_m} - \mathbf{\Pi}_{\boldsymbol{W}_T}\right\|_2 > \frac{g_{\min}}{4f_{\max}}\min\left\{\frac{\eta_k}{\sqrt{2}}, 1\right\}\,\middle|\,\widehat{\boldsymbol{g}} \neq \boldsymbol{0}\right] + \mathbb{P}\left[\left\|\boldsymbol{f} - \widehat{\boldsymbol{f}}\right\|_2 > \frac{g_{\min}}{4}\min\left\{\frac{\eta_k}{\sqrt{2}}, 1\right\}\,\middle|\,\widehat{\boldsymbol{g}} \neq \boldsymbol{0}\right].$$

To have

$$\left\|\mathbf{\Pi}_{\boldsymbol{P}_m} - \mathbf{\Pi}_{\boldsymbol{W}_T}\right\|_2 \leq \frac{g_{\min}}{4f_{\max}}\min\left\{\frac{\eta_k}{\sqrt{2}}, 1\right\}$$

with probability at least $1 - \frac{\delta}{12}$, it is sufficient that

$$T = \Theta\left(\frac{K_{m,\nu}}{\Delta_{m,\nu}}\log\left(\frac{d}{\eta_k\delta}\frac{f_{\max}}{g_{\min}}\right)\right), \tag{22}$$

$$b = \Omega\left(\frac{\mathcal{V}}{\Delta_{m,\nu}^2 p_{\min}}\left(\frac{dm}{\delta^2}\log\frac{m}{\delta} + \frac{f_{\max}^2}{\eta_k^2 g_{\min}^2}\log\frac{d}{\delta}\right) + \frac{\mathcal{M}^2}{\mathcal{V}p_{\min}}\log\frac{d}{\delta}\right). \tag{23}$$

Moreover, to have

$$\left\|\boldsymbol{f} - \widehat{\boldsymbol{f}}\right\|_2 \leq \frac{g_{\min}}{4}\min\left\{\frac{\eta_k}{\sqrt{2}}, 1\right\}$$

with probability at least $1 - \frac{\delta}{12}$, it is sufficient that

$$Tb \geq \frac{128\mathfrak{c}^2\sigma^2}{p_{\min}g_{\min}^2}\max\left\{\frac{2}{\eta_k^2}, 1\right\}\log\frac{48d}{\delta}. \tag{24}$$

Next, we obtain a sample complexity to bound the second probability term $\mathbb{P}\left[\widehat{\boldsymbol{g}} = \boldsymbol{0}\right] < \frac{\delta}{4}$. Because of Eqn. (21),

$$\mathbb{P}\left[\widehat{\boldsymbol{g}} = \boldsymbol{0}\right] \leq \mathbb{P}\left[\|\boldsymbol{g} - \widehat{\boldsymbol{g}}\|_2 \geq \|\boldsymbol{g}\|_2\right]$$

$$\leq \mathbb{P}\left[\|\mathbf{\Pi}_{\boldsymbol{P}_m} - \mathbf{\Pi}_{\boldsymbol{W}_T}\|_2 \geq \frac{g_{\min}}{2f_{\max}}\right] + \mathbb{P}\left[\left\|\boldsymbol{f} - \widehat{\boldsymbol{f}}\right\|_2 \geq \frac{g_{\min}}{2}\right].$$

To have $\|\mathbf{\Pi}_{\boldsymbol{P}_m} - \mathbf{\Pi}_{\boldsymbol{W}_T}\|_2 < \frac{g_{\min}}{2f_{\max}}$ with probability at least $1 - \frac{\delta}{8}$, it suffices to have a sample complexity in Eqn. (22) and (23). Also, to have $\left\|\boldsymbol{f} - \widehat{\boldsymbol{f}}\right\|_2 < \frac{g_{\min}}{2}$ with probability at least $1 - \frac{\delta}{8}$, it suffices to have a sample complexity in Eqn. (24).

Combining the bounds in the **Case 1** with a union bound, we obtain a full sample complexity for Algorithm 3 as follows: a total of $N_1^{(1)} = Tb$ samples are sufficient for ensuring $(\varepsilon_{\mathrm{f}}, \varepsilon_{\mathrm{o}})$-PAFO-learnability with probability at least $1 - \delta$, where

$$\begin{aligned} N_1^{(1)} &= \Omega\left(\frac{K_{m,\nu}}{p_{\min}}\left\{\frac{\mathcal{V}}{\Delta_{m,\nu}^3}\left(\frac{dm}{\delta^2}\log\frac{m}{\delta} + \frac{f_{\max}^2}{\eta_k^2 g_{\min}^2}\log\frac{d}{\delta}\right) + \frac{\mathcal{M}^2}{\mathcal{V}\Delta_{m,\nu}}\log\frac{d}{\delta}\right\}\log\left(\frac{d}{\eta_k\delta}\frac{f_{\max}}{g_{\min}}\right)\right. \\ &\quad \left. + \frac{\sigma^2}{p_{\min}g_{\min}^2\eta_k^2}\log\frac{d}{\delta}\right) \\ &= \tilde{\Omega}\left(\frac{K_{m,\nu}}{p_{\min}}\left\{\frac{\mathcal{V}}{\Delta_{m,\nu}^3}\left(\frac{dm}{\delta^2} + \frac{f_{\max}^2}{\eta_k^2 g_{\min}^2}\right) + \frac{\mathcal{M}^2}{\mathcal{V}\Delta_{m,\nu}}\right\} + \frac{\sigma^2}{p_{\min}g_{\min}^2\eta_k^2}\right). \end{aligned} \quad (25)$$

**Case 2.** Assume $\boldsymbol{g} = \boldsymbol{0}$. In this case, $\boldsymbol{U} = \boldsymbol{P}_m$. We want to upper-bound the follow probability to be $< \delta/2$:

$$\begin{aligned} &\mathbb{P}\left[\left\|\mathbf{\Pi}_{\boldsymbol{U}}\mathbf{\Pi}_{\widehat{\boldsymbol{U}}}^{\perp}\right\|_2 > \eta_k\right] \\ &= \mathbb{P}\left[\left(\widehat{\boldsymbol{g}} = \boldsymbol{0} \text{ and } \left\|\mathbf{\Pi}_{\boldsymbol{U}} - \mathbf{\Pi}_{\widehat{\boldsymbol{U}}}\right\|_2 > \eta_k\right) \text{ or } \left(\widehat{\boldsymbol{g}} \neq \boldsymbol{0} \text{ and } \left\|\mathbf{\Pi}_{\boldsymbol{U}}\mathbf{\Pi}_{\widehat{\boldsymbol{U}}}^{\perp}\right\|_2 > \eta_k\right)\right] \\ &\leq \mathbb{P}\left[\left.\left\|\mathbf{\Pi}_{\boldsymbol{P}_m} - \mathbf{\Pi}_{\boldsymbol{W}_T}\right\|_2 > \eta_k\right| \widehat{\boldsymbol{g}} = \boldsymbol{0}\right] + \mathbb{P}\left[\left.\left\|\mathbf{\Pi}_{\boldsymbol{P}_m}\mathbf{\Pi}_{[\boldsymbol{W}_T|\widehat{\boldsymbol{h}}]}^{\perp}\right\|_2 > \eta_k\right| \widehat{\boldsymbol{g}} \neq \boldsymbol{0}\right]. \end{aligned}$$

The first probability term is bounded by $\frac{\delta}{4}$ given that a sufficient number of samples just like Eqn. (19) and (20). To bound the second one, note that $\widehat{\boldsymbol{h}}$ is orthogonal to every column of $\boldsymbol{W}_T$ (*i.e.*, $\mathbf{\Pi}_{\boldsymbol{W}_T}\widehat{\boldsymbol{h}} = \boldsymbol{0}$). Thus,

$$\begin{aligned} \left\|\mathbf{\Pi}_{\boldsymbol{P}_m}\mathbf{\Pi}_{[\boldsymbol{W}_T|\widehat{\boldsymbol{h}}]}^{\perp}\right\|_2 &= \left\|\mathbf{\Pi}_{\boldsymbol{P}_m} - \mathbf{\Pi}_{\boldsymbol{P}_m}\left(\mathbf{\Pi}_{\boldsymbol{W}_T} + \widehat{\boldsymbol{h}}\widehat{\boldsymbol{h}}^{\intercal}\right)\right\|_2 \\ &= \left\|\mathbf{\Pi}_{\boldsymbol{P}_m}\mathbf{\Pi}_{\boldsymbol{W}_T}^{\perp} - (\mathbf{\Pi}_{\boldsymbol{P}_m} - \mathbf{\Pi}_{\boldsymbol{W}_T})\widehat{\boldsymbol{h}}\widehat{\boldsymbol{h}}^{\intercal}\right\|_2 \\ &\leq 2\left\|\mathbf{\Pi}_{\boldsymbol{P}_m} - \mathbf{\Pi}_{\boldsymbol{W}_T}\right\|_2, \end{aligned}$$

which can be bounded by $< \eta_k$ with probability at least $1 - \frac{\delta}{4}$ given that a sufficient number of samples just like Eqn. (19) and (20) again. Therefore, a total of $N_1^{(2)} = Tb$ samples are sufficient for ensuring $(\varepsilon_{\mathrm{f}}, \varepsilon_{\mathrm{o}})$-PAFO-learnability with probability at least $1 - \delta$, where

$$\begin{aligned} N_1^{(1)} &= \Omega\left(\frac{K_{m,\nu}}{p_{\min}}\left\{\frac{\mathcal{V}}{\Delta_{m,\nu}^3}\left(\frac{dm}{\delta^2}\log\frac{m}{\delta} + \frac{1}{\eta_k^2}\log\frac{d}{\delta}\right) + \frac{\mathcal{M}^2}{\mathcal{V}\Delta_{m,\nu}}\log\frac{d}{\delta}\right\}\log\left(\frac{d}{\eta_k\delta}\right)\right) \\ &= \tilde{\Omega}\left(\frac{K_{m,\nu}}{p_{\min}}\left\{\frac{\mathcal{V}}{\Delta_{m,\nu}^3}\left(\frac{dm}{\delta^2} + \frac{1}{\eta_k^2}\right) + \frac{\mathcal{M}^2}{\mathcal{V}\Delta_{m,\nu}}\right\}\right). \end{aligned} \quad (26)$$

Combining **Case 1** (Eqn. (25)) and **Case 2** (Eqn. (26)) as $N_1 = \mathbb{1}[\boldsymbol{g} \neq \boldsymbol{0}]N_1^{(1)} + \mathbb{1}[\boldsymbol{g} = \boldsymbol{0}]N_1^{(2)}$, we have our desired statement.

Lastly, we provide the missing proof for our lemmas:

*Proof of Lemma 7.* Let

$$\boldsymbol{A} = \underbrace{\left[\boldsymbol{A}_1\right.}_{k} | \underbrace{\left.\boldsymbol{A}_2\right]}_{d-k} \quad \text{and} \quad \boldsymbol{B} = \underbrace{\left[\boldsymbol{B}_1\right.}_{\ell} | \underbrace{\left.\boldsymbol{B}_2\right]}_{d-\ell}$$

be $d \times d$ orthogonal matrices. We first show that $\sin\left(\mathrm{col}(\boldsymbol{A}_1), \mathrm{col}(\boldsymbol{B}_1)\right) = \|\boldsymbol{A}_2^\intercal \boldsymbol{B}_1\|_2$. It can be proven as

$$\left\|\boldsymbol{\Pi}_{\boldsymbol{A}_1}^\perp \boldsymbol{B}_1\right\|_2 = \|\boldsymbol{A}_2 \boldsymbol{A}_2^\intercal \boldsymbol{B}_1\|_2 \le \|\boldsymbol{A}_2^\intercal \boldsymbol{B}_1\|_2 = \|\boldsymbol{A}_2^\intercal \boldsymbol{A}_2 \boldsymbol{A}_2^\intercal \boldsymbol{B}_1\|_2 \le \|\boldsymbol{A}_2 \boldsymbol{A}_2^\intercal \boldsymbol{B}_1\|_2 \,.$$

Thus, from now on, we will work with $\|\boldsymbol{A}_2^\intercal \boldsymbol{B}_1\|_2$ instead of $\sin\left(\mathrm{col}(\boldsymbol{A}_1), \mathrm{col}(\boldsymbol{B}_1)\right)$.

The rest of the proof borrows the idea from the proof of Golub and Loan (2013, Theorem 2.5.1). Observe that

$$\begin{aligned}
\mathrm{dist}(\mathrm{col}(\boldsymbol{A}_1), \mathrm{col}(\boldsymbol{B}_1)) &= \|\boldsymbol{A}^\intercal (\boldsymbol{A}_1 \boldsymbol{A}_1^\intercal - \boldsymbol{B}_1 \boldsymbol{B}_1^\intercal) \boldsymbol{B}\|_2 \\
&= \left\| \begin{bmatrix} \boldsymbol{0} & \boldsymbol{A}_1^\intercal \boldsymbol{B}_2 \\ -\boldsymbol{A}_2^\intercal \boldsymbol{B}_1 & \boldsymbol{0} \end{bmatrix} \right\|_2 \\
&= \max\left\{ \|\boldsymbol{B}_2^\intercal \boldsymbol{A}_1\|_2, \|\boldsymbol{A}_2^\intercal \boldsymbol{B}_1\|_2 \right\}.
\end{aligned}$$

Note that $\boldsymbol{A}_2^\intercal \boldsymbol{B}_1$ and $\boldsymbol{A}_2^\intercal \boldsymbol{B}_1$ are submatrices of a $d \times d$ orthogonal matrix $\boldsymbol{Q}$ defined as

$$\boldsymbol{Q} \triangleq \boldsymbol{A}^\intercal \boldsymbol{B} = \begin{bmatrix} \boldsymbol{A}_1^\intercal \boldsymbol{B}_1 & \boldsymbol{A}_1^\intercal \boldsymbol{B}_2 \\ \boldsymbol{A}_2^\intercal \boldsymbol{B}_1 & \boldsymbol{A}_2^\intercal \boldsymbol{B}_2 \end{bmatrix}.$$

Readers might notice that $\boldsymbol{A}_1^\intercal \boldsymbol{B}_1 \in \mathbb{R}^{k \times \ell}$. For any unit vector (in 2-norm) $\boldsymbol{x} \in \mathbb{R}^k$,

$$1 = \left\| \boldsymbol{Q} \begin{bmatrix} \boldsymbol{x} \\ \boldsymbol{0} \end{bmatrix} \right\|_2^2 = \left\| \begin{bmatrix} \boldsymbol{A}_1^\intercal \boldsymbol{B}_1 \boldsymbol{x} \\ \boldsymbol{A}_2^\intercal \boldsymbol{B}_1 \boldsymbol{x} \end{bmatrix} \right\|_2^2 = \|\boldsymbol{A}_1^\intercal \boldsymbol{B}_1 \boldsymbol{x}\|_2^2 + \|\boldsymbol{A}_2^\intercal \boldsymbol{B}_1 \boldsymbol{x}\|_2^2.$$

Thus,

$$\begin{aligned}
\|\boldsymbol{A}_2^\intercal \boldsymbol{B}_1\|_2^2 &= \max_{\boldsymbol{x} \in \mathbb{R}^k, \|\boldsymbol{x}\|_2 = 1} \|\boldsymbol{A}_2^\intercal \boldsymbol{B}_1 \boldsymbol{x}\|_2^2 \\
&= 1 - \min_{\boldsymbol{x} \in \mathbb{R}^k, \|\boldsymbol{x}\|_2 = 1} \|\boldsymbol{A}_1^\intercal \boldsymbol{B}_1 \boldsymbol{x}\|_2^2 \\
&= \begin{cases} 1 - \sigma_{\min}(\boldsymbol{A}_1^\intercal \boldsymbol{B}_1)^2, & \text{if } k \ge \ell, \\ 1, & \text{if } k < \ell. \end{cases}
\end{aligned}$$

Likewise, working with $\boldsymbol{Q}^\intercal$, one can deduce that

$$\begin{aligned}
\|\boldsymbol{B}_2^\intercal \boldsymbol{A}_1\|_2^2 &= \max_{\boldsymbol{y} \in \mathbb{R}^\ell, \|\boldsymbol{y}\|_2 = 1} \|\boldsymbol{B}_2^\intercal \boldsymbol{A}_1 \boldsymbol{y}\|_2^2 \\
&= 1 - \min_{\boldsymbol{y} \in \mathbb{R}^\ell, \|\boldsymbol{y}\|_2 = 1} \|\boldsymbol{B}_1^\intercal \boldsymbol{A}_1 \boldsymbol{y}\|_2^2 \\
&= \begin{cases} 1 - \sigma_{\min}(\boldsymbol{B}_1^\intercal \boldsymbol{A}_1)^2, & \text{if } k \le \ell, \\ 1, & \text{if } k > \ell. \end{cases}
\end{aligned}$$

Therefore, our desired statement naturally follows. $\qquad\square$

*Proof of Lemma 8.* To prove the first claim, let $\boldsymbol{u}$ and $\boldsymbol{v}$ be unit ($\ell_2$ norm) vectors in direction of $\boldsymbol{a}$ and $\boldsymbol{b}$, respectively. Then, by Lemma 7,

$$\|\boldsymbol{u}\boldsymbol{u}^\intercal - \boldsymbol{v}\boldsymbol{v}^\intercal\|_2^2 = 1 - \sigma_{\min}(\boldsymbol{u}^\intercal \boldsymbol{v})^2 = 1 - |\boldsymbol{u}^\intercal \boldsymbol{v}|^2 = 1 - \cos^2 \theta(\boldsymbol{u}, \boldsymbol{v}) = \sin^2 \theta(\boldsymbol{a}, \boldsymbol{b}).$$

Now, observe that $\|\boldsymbol{b}\|_2 \ge \|\boldsymbol{a}\|_2 - \epsilon \ge \|\boldsymbol{a}\|_2 / 2$ and

$$2(\|\boldsymbol{a}\|_2 \|\boldsymbol{b}\|_2 - \langle \boldsymbol{a}, \boldsymbol{b}\rangle) \le \|\boldsymbol{a}\|_2^2 + \|\boldsymbol{b}\|_2^2 - 2\langle \boldsymbol{a}, \boldsymbol{b}\rangle = \|\boldsymbol{a} - \boldsymbol{b}\|_2^2 \le \epsilon^2.$$

Thus,

$$\begin{aligned}
\sin^2 \theta(\boldsymbol{a}, \boldsymbol{b}) &= (1 + \cos\theta(\boldsymbol{a}, \boldsymbol{b}))(1 - \cos\theta(\boldsymbol{a}, \boldsymbol{b})) \le 2(1 - \cos\theta(\boldsymbol{a}, \boldsymbol{b})) \\
&= 2\frac{\|\boldsymbol{a}\|_2 \|\boldsymbol{b}\|_2 - \langle \boldsymbol{a}, \boldsymbol{b}\rangle}{\|\boldsymbol{a}\|_2 \|\boldsymbol{b}\|_2} \le \frac{2\epsilon^2}{\|\boldsymbol{a}\|_2^2}.
\end{aligned}$$

$\qquad\square$

# F More Experiments

## F.1 Synthetic Example

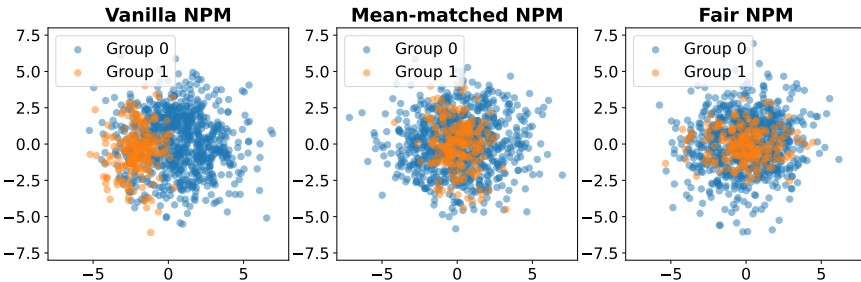

Figure 2: **Synthetic Example**: Vanilla NPM v.s. Mean-matched NPM v.s. FNPM (ours).

We randomly generated two different group-conditional distributions as 10-dimensional multivariate Gaussians with different mean vectors $\boldsymbol{\mu}_0$ and $\boldsymbol{\mu}_1$, satisfying $\boldsymbol{\mu} = (1-p)\boldsymbol{\mu}_0 + p\boldsymbol{\mu}_1 = \mathbf{0}$, and different covariance matrices $\boldsymbol{\Sigma}_0$ and $\boldsymbol{\Sigma}_1$. We choose the sampling probability parameter $p$ as 0.2, which induces asymmetric sampling between two sensitive attributes. The covariance matrices are designed so that both of their eigenvalue spectra, $\{\sigma_1, \ldots, \sigma_d\}$, have *power-law* decay as many practical datasets do (Liu et al., 2015a), *i.e.,* $\sigma_j = \Theta(j^{-\alpha})$ for some decay parameter $\alpha \geq 1$.

We first run and compare three different algorithms: vanilla NPM (without any fairness constraint), mean-matched NPM (with only constraint $\boldsymbol{V}^\mathsf{T}\boldsymbol{f} = \mathbf{0}$), and FNPM with $m = 3$. To ease the visualization, we project the sampled distributions onto a 2-dimensional subspace (*i.e.,* running 2-PCA). After running three algorithms for ten iterations and with a block size of $b = B = 1000$, we randomly sample 1000 data points and visualize the results of projecting the data points in Figure 2. In particular, for FNPM, we run 50 iterations for unfair subspace estimation (Algorithm 3) and run the other 50 iterations for PCA (Algorithm 4). We observe that FNPM does indeed enforce both mean-matching and (partial) covariance-matching, despite the setting being streaming and having asymmetric sampling probability.

## F.2 Additional Results on CelebA Dataset

In Figures 3-5, we provide additional experimental results on the CelebA dataset (Liu et al., 2015b). We consider three sensitive attributes in the CelebA dataset: "Eyeglasses", "Mouth Slightly Open", and "Goattee". In all the figures, False is when the sensitive attribute is absent; True is otherwise.

Figure 3 presents the main results for the new attributes. The first row shows the original images, the second row shows the images after projecting them to $\mathrm{col}(\boldsymbol{V}^\star)$, and the third row shows the images after projecting them to the estimated unfair subspace, $\mathrm{col}(\widehat{\boldsymbol{U}})$. Here, both $\widehat{\boldsymbol{U}}$ and $\boldsymbol{V}^\star$ are obtained from our FNPM; specifically speaking, they are obtained from Algorithm 3 and Algorithm 4, respectively. While we've used $m = 2$ for Figure 1 in the main text, we use $m = 5$ here.

We then perform the two ablation studies as described in the main text, one on the dimension $m$ of unfair subspace and another on the block size $b$ of Algorithm 3, for the additional sensitive attributes considered. In Figure 4, we vary $m \in \{1, 2, 5, 10\}$: as $m$ increases, more features of images are "erased", making the images from the two sensitive groups less distinguishable. At the same time, more semantically meaningful features are erased as well, resulting in rather "alien-like" images. In Figure 5, we vary $b \in \{32000, 8000, 3200, 1600\}$: as $b$ increases, we have a more accurate estimation of unfair subspace $\mathrm{col}(\boldsymbol{U})$, resulting in more indistinguishable images (in sensitive attributes). As soon as the batch size $b$ exceeds a certain threshold (e.g., 8000 for "Goattee"), $\mathrm{col}(\boldsymbol{U})$ is estimated very well, and the unfair features are cleanly recovered, as it can be seen in the bottom rows.

## F.3 Full Results on UCI Dataset

The full experiment results of fair PCA and downstream tasks on UCI datasets are provided. Again, our method is competitive to the existing fair PCA algorithms across the considered UCI datasets.

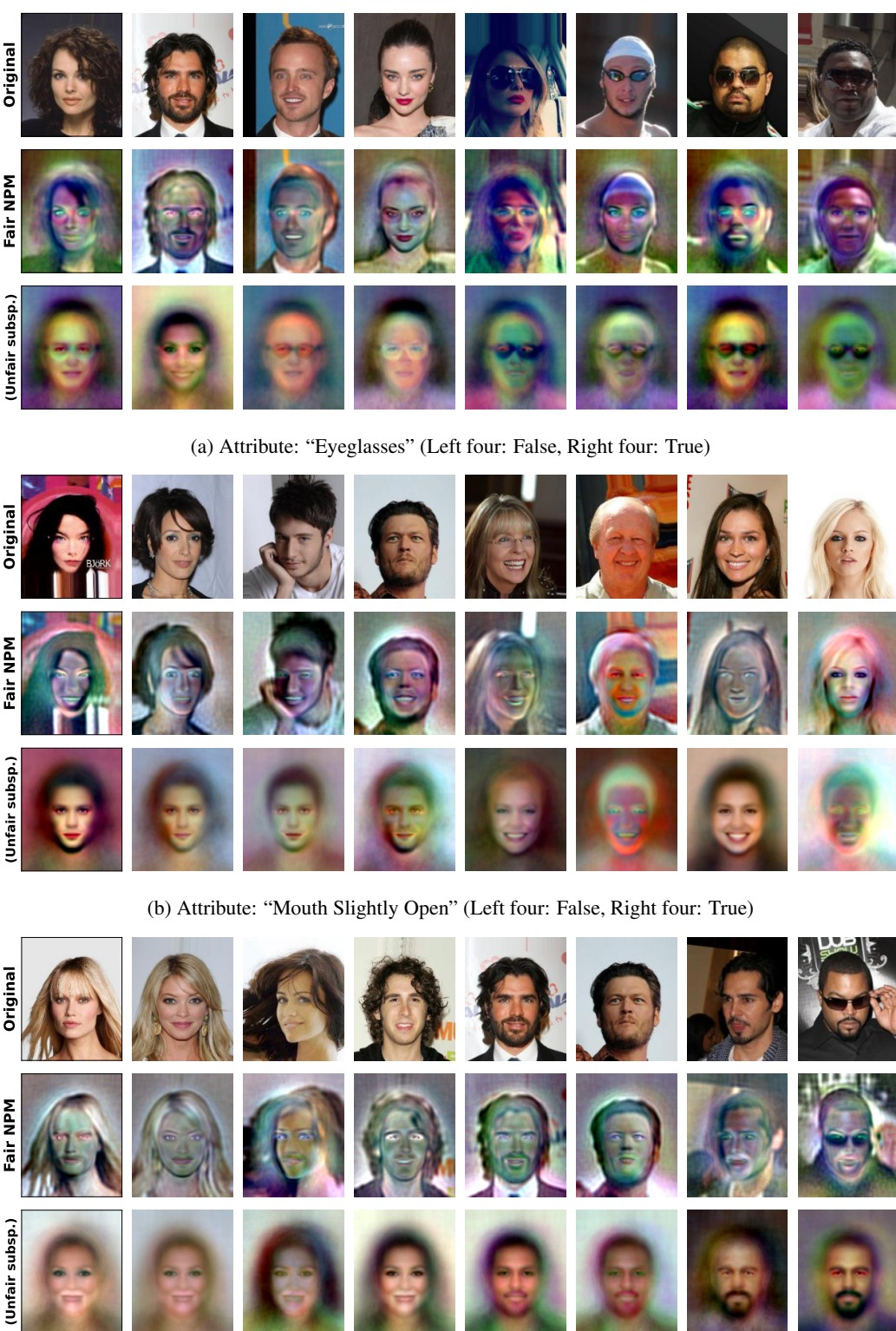

(a) Attribute: "Eyeglasses" (Left four: False, Right four: True)

(b) Attribute: "Mouth Slightly Open" (Left four: False, Right four: True)

(c) Attribute: "Goatee" (Left four: False, Right four: True)

Figure 3: CelebA, Additional results ($m = 5$).

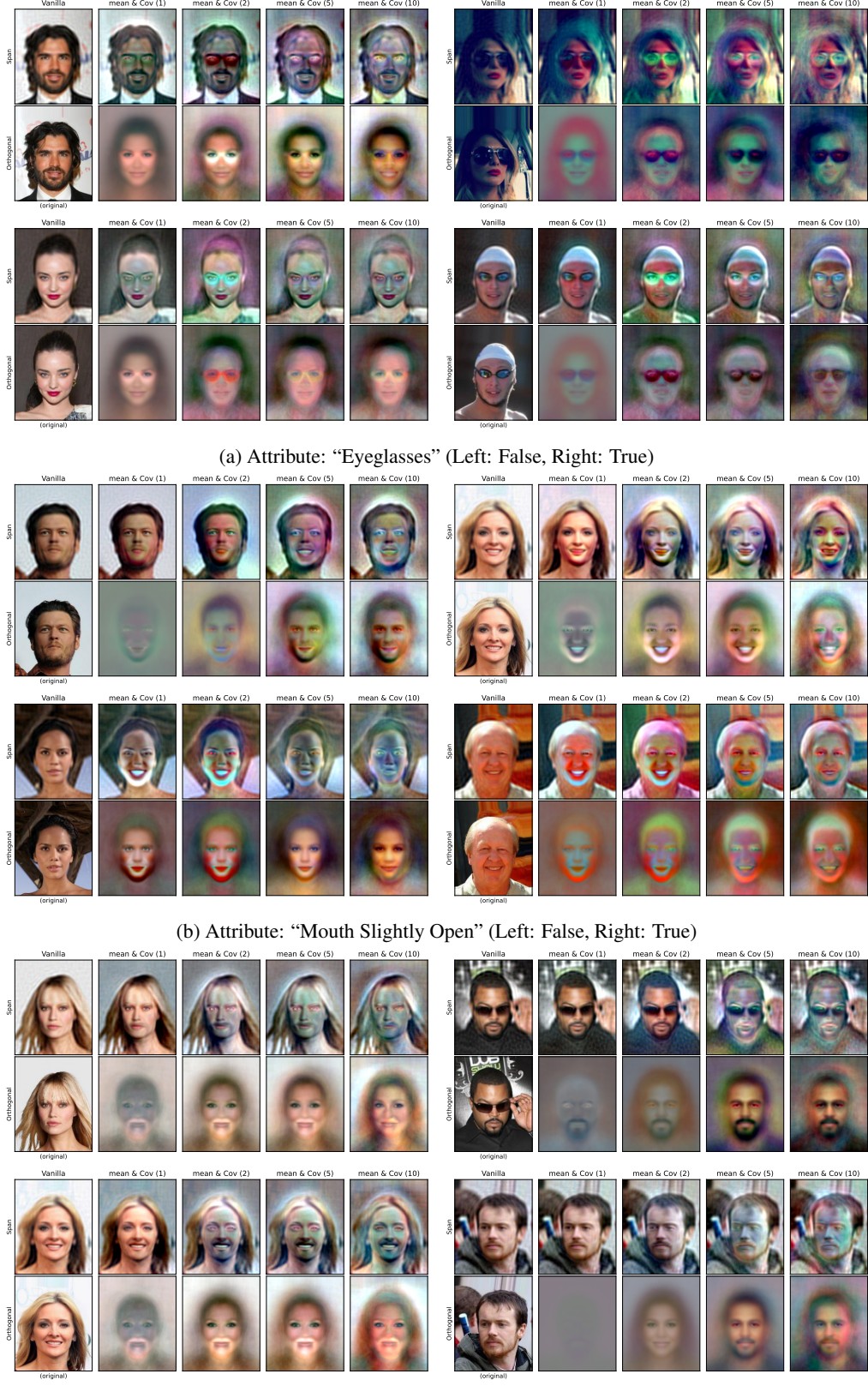

(a) Attribute: "Eyeglasses" (Left: False, Right: True)

(b) Attribute: "Mouth Slightly Open" (Left: False, Right: True)

(c) Attribute: "Goatee" (Left: False, Right: True)

Figure 4: CelebA, Additional ablation on $m \in \{1, 2, 5, 10\}$

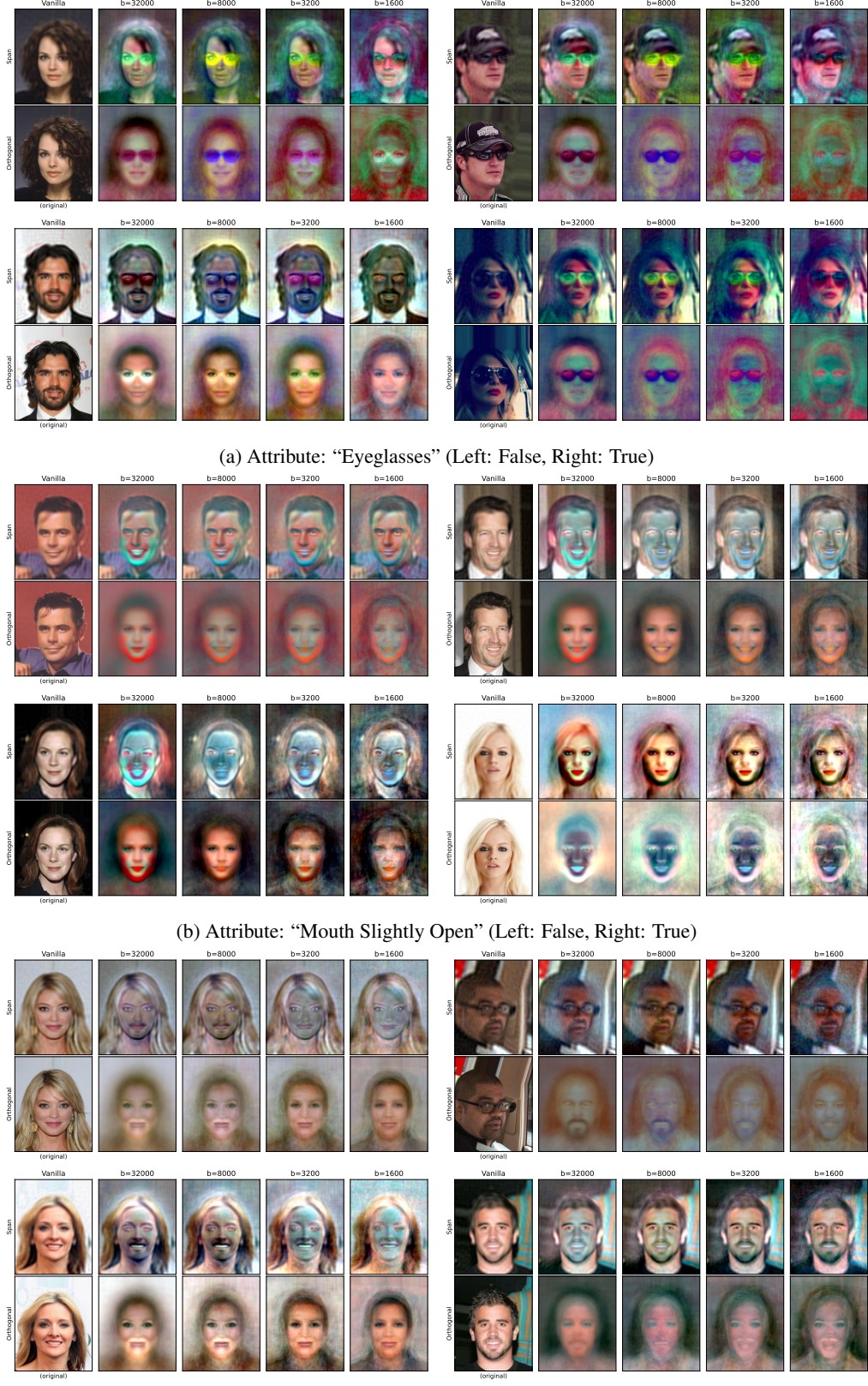

(a) Attribute: "Eyeglasses" (Left: False, Right: True)

(b) Attribute: "Mouth Slightly Open" (Left: False, Right: True)

(c) Attribute: "Goatee" (Left: False, Right: True)

Figure 5: CelebA, Additional ablation on $b \in \{32000, 8000, 3200, 1600\}$

Table 2: Dataset = **Adult Income** [feature dim=102, #(train data)=31,655]

| k | Method | %Var(↑) | MMD²(↓) | kernel SVM %Acc(↑) | kernel SVM $\Delta_{\mathrm{DP}}$(↓) | linear SVM %Acc(↑) | linear SVM $\Delta_{\mathrm{DP}}$(↓) | MLP %Acc(↑) | MLP $\Delta_{\mathrm{DP}}$(↓) |
|---|---|---|---|---|---|---|---|---|---|
| 2 | PCA | $6.88_{(0.14)}$ | $0.374_{(0.006)}$ | $82.4_{(0.2)}$ | $0.19_{(0.01)}$ | $81.23_{(0.16)}$ | $0.2_{(0.0)}$ | $82.31_{(0.18)}$ | $0.19_{(0.01)}$ |
| | Olfat and Aswani (2019) (0.1) | | | Memory Out | | Memory Out | | | |
| | Olfat and Aswani (2019) (0.0) | | | Memory Out | | Memory Out | | | |
| | Lee et al. (2022) (1e-3) | $5.68_{(0.11)}$ | $0.0_{(0.0)}$ | $80.34_{(0.24)}$ | $0.05_{(0.01)}$ | $78.09_{(0.36)}$ | $0.01_{(0.0)}$ | $80.31_{(0.26)}$ | $0.06_{(0.01)}$ |
| | Lee et al. (2022) (1e-6) | $5.42_{(0.11)}$ | $0.0_{(0.0)}$ | $79.41_{(0.23)}$ | $0.02_{(0.01)}$ | $77.52_{(0.69)}$ | $0.01_{(0.0)}$ | $79.48_{(0.29)}$ | $0.02_{(0.01)}$ |
| | Kleindessner et al. (2023) (mean) | $5.74_{(0.11)}$ | $0.002_{(0.0)}$ | $80.6_{(0.2)}$ | $0.07_{(0.01)}$ | $78.57_{(0.26)}$ | $0.01_{(0.0)}$ | $80.42_{(0.17)}$ | $0.07_{(0.01)}$ |
| | Kleindessner et al. (2023) (0.85) | $4.09_{(0.17)}$ | $0.001_{(0.001)}$ | $75.52_{(0.21)}$ | $0.0_{(0.0)}$ | $75.33_{(0.19)}$ | $0.0_{(0.0)}$ | $75.37_{(0.23)}$ | $0.0_{(0.0)}$ |
| | Kleindessner et al. (2023) (0.5) | $2.63_{(0.07)}$ | $0.0_{(0.0)}$ | $75.38_{(0.18)}$ | $0.0_{(0.0)}$ | $75.33_{(0.19)}$ | $0.0_{(0.0)}$ | $75.33_{(0.19)}$ | $0.0_{(0.0)}$ |
| | Kleindessner et al. (2023) (kernel) | | | Takes too long time | | | | | |
| | Ravfogel et al. (2020) | $1.91_{(0.08)}$ | $0.001_{(0.001)}$ | $75.67_{(0.31)}$ | $0.0_{(0.0)}$ | $75.33_{(0.19)}$ | $0.0_{(0.0)}$ | $75.54_{(0.39)}$ | $0.0_{(0.01)}$ |
| | Ravfogel et al. (2022a) | $1.91_{(0.09)}$ | $0.006_{(0.011)}$ | $75.59_{(0.34)}$ | $0.0_{(0.0)}$ | $75.33_{(0.19)}$ | $0.0_{(0.0)}$ | $75.48_{(0.35)}$ | $0.0_{(0.0)}$ |
| | Samadi et al. (2018) | N/A | N/A | $82.63_{(0.18)}$ | $0.15_{(0.01)}$ | $82.21_{(0.2)}$ | $0.17_{(0.0)}$ | $77.62_{(3.46)}$ | $0.06_{(0.09)}$ |
| | **Ours** (offline, mean) | $5.74_{(0.11)}$ | $0.002_{(0.0)}$ | $80.6_{(0.2)}$ | $0.07_{(0.01)}$ | $78.57_{(0.26)}$ | $0.01_{(0.0)}$ | $80.31_{(0.22)}$ | $0.07_{(0.01)}$ |
| | **Ours** (FNPM, mean) | $5.74_{(0.11)}$ | $0.002_{(0.0)}$ | $80.6_{(0.2)}$ | $0.07_{(0.01)}$ | $78.57_{(0.26)}$ | $0.01_{(0.0)}$ | $80.33_{(0.25)}$ | $0.07_{(0.01)}$ |
| | **Ours** (offline, $m=15$) | $4.04_{(0.14)}$ | $0.001_{(0.001)}$ | $75.51_{(0.23)}$ | $0.0_{(0.0)}$ | $75.33_{(0.19)}$ | $0.0_{(0.0)}$ | $75.39_{(0.24)}$ | $0.0_{(0.0)}$ |
| | **Ours** (FNPM, $m=15$) | $4.07_{(0.13)}$ | $0.001_{(0.0)}$ | $75.54_{(0.25)}$ | $0.0_{(0.0)}$ | $75.33_{(0.19)}$ | $0.0_{(0.0)}$ | $75.58_{(0.26)}$ | $0.01_{(0.01)}$ |
| | **Ours** (offline, $m=50$) | $2.63_{(0.07)}$ | $0.0_{(0.0)}$ | $75.38_{(0.18)}$ | $0.0_{(0.0)}$ | $75.33_{(0.19)}$ | $0.0_{(0.0)}$ | $75.39_{(0.25)}$ | $0.0_{(0.0)}$ |
| | **Ours** (FNPM, $m=50$) | $2.64_{(0.06)}$ | $0.0_{(0.0)}$ | $75.44_{(0.21)}$ | $0.0_{(0.0)}$ | $75.33_{(0.19)}$ | $0.0_{(0.0)}$ | $75.37_{(0.2)}$ | $0.0_{(0.0)}$ |
| 10 | PCA | $20.82_{(0.38)}$ | $0.195_{(0.002)}$ | $88.58_{(0.21)}$ | $0.18_{(0.01)}$ | $83.14_{(0.21)}$ | $0.19_{(0.0)}$ | $83.88_{(0.23)}$ | $0.2_{(0.01)}$ |
| | Olfat and Aswani (2019) (0.1) | | | Memory Out | | Memory Out | | | |
| | Olfat and Aswani (2019) (0.0) | | | Memory Out | | Memory Out | | | |
| | Lee et al. (2022) (1e-3) | | | Takes too long time | | Takes too long time | | | |
| | Lee et al. (2022) (1e-6) | | | Takes too long time | | Takes too long time | | | |
| | Kleindessner et al. (2023) (mean) | $18.84_{(0.35)}$ | $0.01_{(0.0)}$ | $88.43_{(0.18)}$ | $0.18_{(0.01)}$ | $81.71_{(0.29)}$ | $0.03_{(0.01)}$ | $83.83_{(0.32)}$ | $0.17_{(0.01)}$ |
| | Kleindessner et al. (2023) (0.85) | $14.54_{(0.23)}$ | $0.002_{(0.0)}$ | $86.99_{(0.32)}$ | $0.16_{(0.0)}$ | $75.33_{(0.19)}$ | $0.0_{(0.0)}$ | $81.92_{(0.56)}$ | $0.14_{(0.01)}$ |
| | Kleindessner et al. (2023) (0.5) | $11.34_{(0.2)}$ | $0.0_{(0.0)}$ | $83.01_{(0.28)}$ | $0.12_{(0.01)}$ | $75.33_{(0.19)}$ | $0.0_{(0.0)}$ | $79.13_{(0.77)}$ | $0.08_{(0.02)}$ |
| | Kleindessner et al. (2023) (kernel) | | | Takes too long time | | | | | |
| | Ravfogel et al. (2020) | $9.36_{(0.26)}$ | $0.001_{(0.0)}$ | $85.71_{(0.78)}$ | $0.14_{(0.02)}$ | $75.33_{(0.19)}$ | $0.0_{(0.0)}$ | $80.18_{(0.68)}$ | $0.1_{(0.02)}$ |
| | Ravfogel et al. (2022a) | $9.6_{(0.25)}$ | $0.001_{(0.0)}$ | $87.61_{(0.57)}$ | $0.16_{(0.01)}$ | $75.37_{(0.22)}$ | $0.0_{(0.0)}$ | $80.72_{(0.9)}$ | $0.11_{(0.02)}$ |
| | Samadi et al. (2018) | N/A | N/A | $85.69_{(0.24)}$ | $0.18_{(0.01)}$ | $83.25_{(0.18)}$ | $0.17_{(0.01)}$ | $84.52_{(0.29)}$ | $0.19_{(0.01)}$ |
| | **Ours** (offline, mean) | $18.84_{(0.35)}$ | $0.01_{(0.0)}$ | $88.43_{(0.18)}$ | $0.18_{(0.01)}$ | $81.71_{(0.29)}$ | $0.03_{(0.01)}$ | $83.7_{(0.24)}$ | $0.17_{(0.01)}$ |
| | **Ours** (FNPM, mean) | $18.83_{(0.35)}$ | $0.009_{(0.0)}$ | $88.53_{(0.22)}$ | $0.18_{(0.01)}$ | $81.7_{(0.29)}$ | $0.03_{(0.01)}$ | $83.85_{(0.28)}$ | $0.18_{(0.01)}$ |
| | **Ours** (offline, $m=15$) | $14.48_{(0.22)}$ | $0.002_{(0.0)}$ | $86.86_{(0.3)}$ | $0.16_{(0.0)}$ | $75.33_{(0.19)}$ | $0.0_{(0.0)}$ | $81.84_{(0.54)}$ | $0.13_{(0.01)}$ |
| | **Ours** (FNPM, $m=15$) | $14.38_{(0.2)}$ | $0.001_{(0.0)}$ | $86.64_{(0.29)}$ | $0.16_{(0.0)}$ | $75.33_{(0.19)}$ | $0.0_{(0.0)}$ | $80.96_{(0.56)}$ | $0.12_{(0.02)}$ |
| | **Ours** (offline, $m=50$) | $11.34_{(0.2)}$ | $0.0_{(0.0)}$ | $83.0_{(0.25)}$ | $0.12_{(0.01)}$ | $75.33_{(0.19)}$ | $0.0_{(0.0)}$ | $78.94_{(0.7)}$ | $0.07_{(0.02)}$ |
| | **Ours** (FNPM, $m=50$) | $11.34_{(0.2)}$ | $0.0_{(0.0)}$ | $82.85_{(0.22)}$ | $0.11_{(0.01)}$ | $75.33_{(0.19)}$ | $0.0_{(0.0)}$ | $78.59_{(0.63)}$ | $0.06_{(0.02)}$ |

Table 3: Dataset = **COMPAS** [feature dim= 11, #(train data)=4,316]

| $k$ | Method | %Var(↑) | MMD$^2$(↓) | kernel SVM %Acc(↑) | kernel SVM $\Delta_{\text{DP}}$(↓) | linear SVM %Acc(↑) | linear SVM $\Delta_{\text{DP}}$(↓) | MLP %Acc(↑) | MLP $\Delta_{\text{DP}}$(↓) |
|---|---|---|---|---|---|---|---|---|---|
| 2 | PCA | $2.09_{(0.11)}$ | $0.057_{(0.006)}$ | $64.84_{(0.78)}$ | $0.28_{(0.03)}$ | $58.91_{(0.52)}$ | $0.2_{(0.03)}$ | $64.16_{(1.01)}$ | $0.3_{(0.03)}$ |
| | Olfat and Aswani (2019) (0.1) | | | Memory Out | | Memory Out | | | |
| | Olfat and Aswani (2019) (0.0) | | | Memory Out | | Memory Out | | | |
| | Lee et al. (2022) (1e-3) | $1.8_{(0.1)}$ | $0.003_{(0.002)}$ | $62.41_{(1.09)}$ | $0.03_{(0.02)}$ | $57.57_{(1.18)}$ | $0.04_{(0.03)}$ | $60.79_{(1.59)}$ | $0.03_{(0.01)}$ |
| | Lee et al. (2022) (1e-6) | $1.78_{(0.11)}$ | $0.003_{(0.002)}$ | $62.65_{(1.54)}$ | $0.03_{(0.03)}$ | $57.56_{(2.06)}$ | $0.04_{(0.03)}$ | $61.09_{(1.53)}$ | $0.03_{(0.01)}$ |
| | Kleindessner et al. (2023) (mean) | $1.97_{(0.1)}$ | $0.008_{(0.003)}$ | $61.94_{(0.59)}$ | $0.07_{(0.02)}$ | $56.03_{(1.09)}$ | $0.05_{(0.04)}$ | $60.79_{(0.8)}$ | $0.07_{(0.04)}$ |
| | Kleindessner et al. (2023) (0.85) | $1.78_{(0.13)}$ | $0.005_{(0.002)}$ | $60.64_{(1.08)}$ | $0.08_{(0.05)}$ | $56.16_{(1.21)}$ | $0.01_{(0.02)}$ | $58.8_{(1.74)}$ | $0.07_{(0.04)}$ |
| | Kleindessner et al. (2023) (0.5) | $1.62_{(0.09)}$ | $0.004_{(0.001)}$ | $59.67_{(0.89)}$ | $0.1_{(0.03)}$ | $54.81_{(1.04)}$ | $0.0_{(0.0)}$ | $57.11_{(1.35)}$ | $0.08_{(0.05)}$ |
| | Kleindessner et al. (2023) (kernel) | N/A | N/A | $57.8_{(1.82)}$ | $0.08_{(0.06)}$ | $54.74_{(1.21)}$ | $0.02_{(0.04)}$ | $55.91_{(1.27)}$ | $0.02_{(0.03)}$ |
| | Ravfogel et al. (2020) | $0.62_{(0.18)}$ | $0.0_{(0.0)}$ | $56.35_{(0.71)}$ | $0.01_{(0.01)}$ | $54.81_{(1.04)}$ | $0.0_{(0.0)}$ | $55.38_{(0.66)}$ | $0.01_{(0.01)}$ |
| | Ravfogel et al. (2022a) | $0.49_{(0.03)}$ | $0.002_{(0.002)}$ | $57.54_{(0.74)}$ | $0.03_{(0.03)}$ | $54.81_{(1.04)}$ | $0.0_{(0.0)}$ | $56.42_{(1.29)}$ | $0.03_{(0.03)}$ |
| | Samadi et al. (2018) | N/A | N/A | $64.19_{(1.07)}$ | $0.15_{(0.03)}$ | $59.15_{(0.59)}$ | $0.13_{(0.03)}$ | $64.4_{(1.32)}$ | $0.15_{(0.03)}$ |
| | **Ours** (offline, mean) | $1.97_{(0.1)}$ | $0.008_{(0.003)}$ | $61.94_{(0.59)}$ | $0.07_{(0.02)}$ | $56.03_{(1.09)}$ | $0.05_{(0.04)}$ | $60.61_{(1.5)}$ | $0.06_{(0.05)}$ |
| | **Ours** (FNPM, mean) | $1.97_{(0.1)}$ | $0.008_{(0.003)}$ | $61.94_{(0.59)}$ | $0.07_{(0.02)}$ | $56.03_{(1.09)}$ | $0.05_{(0.04)}$ | $60.5_{(1.48)}$ | $0.07_{(0.05)}$ |
| | **Ours** (offline, $m$=2) | $1.92_{(0.11)}$ | $0.006_{(0.002)}$ | $61.22_{(0.85)}$ | $0.11_{(0.04)}$ | $55.43_{(0.9)}$ | $0.04_{(0.04)}$ | $59.53_{(1.35)}$ | $0.12_{(0.06)}$ |
| | **Ours** (FNPM, $m$=2) | $1.92_{(0.11)}$ | $0.006_{(0.002)}$ | $61.08_{(0.9)}$ | $0.09_{(0.04)}$ | $55.51_{(1.16)}$ | $0.03_{(0.04)}$ | $59.56_{(1.48)}$ | $0.12_{(0.07)}$ |
| | **Ours** (offline, $m$=5) | $1.89_{(0.1)}$ | $0.006_{(0.002)}$ | $61.71_{(0.78)}$ | $0.1_{(0.04)}$ | $55.49_{(0.79)}$ | $0.03_{(0.04)}$ | $59.74_{(2.0)}$ | $0.14_{(0.06)}$ |
| | **Ours** (FNPM, $m$=5) | $1.89_{(0.1)}$ | $0.006_{(0.002)}$ | $61.67_{(0.76)}$ | $0.11_{(0.05)}$ | $55.55_{(0.9)}$ | $0.03_{(0.04)}$ | $59.63_{(1.84)}$ | $0.13_{(0.07)}$ |
| 10 | PCA | $5.85_{(0.36)}$ | $0.089_{(0.002)}$ | $82.78_{(0.43)}$ | $0.15_{(0.04)}$ | $65.39_{(1.15)}$ | $0.16_{(0.05)}$ | $70.01_{(0.93)}$ | $0.2_{(0.04)}$ |
| | Olfat and Aswani (2019) (0.1) | | | Memory Out | | Memory Out | | | |
| | Olfat and Aswani (2019) (0.0) | | | Memory Out | | Memory Out | | | |
| | Lee et al. (2022) (1e-3) | $5.36_{(0.31)}$ | $0.002_{(0.001)}$ | $80.76_{(0.46)}$ | $0.1_{(0.03)}$ | $63.89_{(1.15)}$ | $0.04_{(0.03)}$ | $68.97_{(1.46)}$ | $0.03_{(0.02)}$ |
| | Lee et al. (2022) (1e-6) | $4.83_{(0.45)}$ | $0.002_{(0.001)}$ | $78.93_{(0.84)}$ | $0.09_{(0.03)}$ | $61.68_{(1.54)}$ | $0.04_{(0.03)}$ | $67.23_{(1.66)}$ | $0.04_{(0.02)}$ |
| | Kleindessner et al. (2023) (mean) | $5.67_{(0.37)}$ | $0.004_{(0.001)}$ | $80.87_{(0.54)}$ | $0.09_{(0.03)}$ | $64.76_{(1.04)}$ | $0.02_{(0.02)}$ | $69.83_{(0.83)}$ | $0.04_{(0.03)}$ |
| | Kleindessner et al. (2023) (0.85) | $5.41_{(0.33)}$ | $0.004_{(0.001)}$ | $80.64_{(0.82)}$ | $0.09_{(0.04)}$ | $63.63_{(1.19)}$ | $0.02_{(0.01)}$ | $68.69_{(1.28)}$ | $0.05_{(0.04)}$ |
| | Kleindessner et al. (2023) (0.5) | $5.29_{(0.31)}$ | $0.003_{(0.001)}$ | $79.22_{(0.44)}$ | $0.1_{(0.03)}$ | $61.62_{(0.63)}$ | $0.02_{(0.02)}$ | $68.63_{(0.92)}$ | $0.07_{(0.04)}$ |
| | Kleindessner et al. (2023) (kernel) | N/A | N/A | $65.96_{(1.12)}$ | $0.26_{(0.07)}$ | $64.35_{(0.8)}$ | $0.05_{(0.04)}$ | $64.93_{(1.49)}$ | $0.04_{(0.03)}$ |
| | Ravfogel et al. (2020) | $2.64_{(0.47)}$ | $0.001_{(0.0)}$ | $64.17_{(1.46)}$ | $0.04_{(0.02)}$ | $54.85_{(1.03)}$ | $0.0_{(0.0)}$ | $62.56_{(1.79)}$ | $0.04_{(0.03)}$ |
| | Ravfogel et al. (2022a) | $2.45_{(0.06)}$ | $0.001_{(0.0)}$ | $71.75_{(0.69)}$ | $0.07_{(0.04)}$ | $55.26_{(1.25)}$ | $0.0_{(0.0)}$ | $66.66_{(1.08)}$ | $0.06_{(0.03)}$ |
| | Samadi et al. (2018) | N/A | N/A | $69.75_{(1.0)}$ | $0.17_{(0.02)}$ | $65.54_{(0.72)}$ | $0.14_{(0.03)}$ | $69.97_{(0.81)}$ | $0.16_{(0.03)}$ |
| | **Ours** (offline, mean) | $5.67_{(0.37)}$ | $0.004_{(0.001)}$ | $80.87_{(0.54)}$ | $0.09_{(0.03)}$ | $64.78_{(1.04)}$ | $0.02_{(0.02)}$ | $69.16_{(0.5)}$ | $0.04_{(0.01)}$ |
| | **Ours** (FNPM, mean) | $5.68_{(0.37)}$ | $0.004_{(0.001)}$ | $80.87_{(0.49)}$ | $0.09_{(0.03)}$ | $64.84_{(1.01)}$ | $0.02_{(0.02)}$ | $68.86_{(0.82)}$ | $0.04_{(0.03)}$ |
| | **Ours** (offline, $m$=2) | $5.57_{(0.35)}$ | $0.004_{(0.001)}$ | $80.93_{(0.63)}$ | $0.09_{(0.03)}$ | $64.72_{(0.93)}$ | $0.02_{(0.02)}$ | $69.25_{(1.27)}$ | $0.04_{(0.04)}$ |
| | **Ours** (FNPM, $m$=2) | $5.58_{(0.35)}$ | $0.004_{(0.001)}$ | $80.82_{(0.48)}$ | $0.09_{(0.03)}$ | $64.65_{(1.1)}$ | $0.02_{(0.02)}$ | $68.97_{(1.31)}$ | $0.04_{(0.02)}$ |
| | **Ours** (offline, $m$=5) | $5.55_{(0.35)}$ | $0.004_{(0.001)}$ | $81.03_{(0.61)}$ | $0.09_{(0.03)}$ | $64.47_{(1.01)}$ | $0.02_{(0.02)}$ | $68.92_{(1.05)}$ | $0.03_{(0.02)}$ |
| | **Ours** (FNPM, $m$=5) | $5.55_{(0.34)}$ | $0.004_{(0.001)}$ | $80.9_{(0.49)}$ | $0.09_{(0.03)}$ | $64.54_{(1.23)}$ | $0.02_{(0.02)}$ | $69.05_{(1.09)}$ | $0.04_{(0.03)}$ |

Table 4: Dataset = **German Credit** [feature dim=59, #(train data)=700]

| k | Method | %Var(↑) | MMD²(↓) | %Acc(↑) kernel SVM | $\Delta_{DP}$(↓) kernel SVM | %Acc(↑) linear SVM | $\Delta_{DP}$(↓) linear SVM | %Acc(↑) MLP | $\Delta_{DP}$(↓) MLP |
|---|---|---|---|---|---|---|---|---|---|
| | PCA | 11.13(0.32) | 0.293(0.054) | 75.47(0.67) | 0.15(0.09) | 70.3(1.59) | 0.03(0.08) | 71.43(0.98) | 0.15(0.09) |
| | Olfat and Aswani (2019) (0.1) | 7.25(0.65) | 0.024(0.01) | 71.53(1.85) | 0.02(0.02) | 69.97(1.93) | 0.0(0.0) | 0.0(0.0) | 0.0(0.0) |
| | Olfat and Aswani (2019) (0.0) | 7.28(0.52) | 0.022(0.009) | 72.03(1.99) | 0.02(0.02) | 69.97(1.93) | 0.0(0.0) | 0.0(0.0) | 0.0(0.0) |
| | Lee et al. (2022) (1e-3) | 9.82(0.45) | 0.027(0.018) | 74.87(1.76) | 0.05(0.04) | 69.97(1.93) | 0.0(0.0) | 71.6(1.58) | 0.05(0.04) |
| | Lee et al. (2022) (1e-6) | 8.89(0.5) | 0.027(0.015) | 73.23(2.27) | 0.03(0.03) | 69.97(1.93) | 0.0(0.0) | 70.73(1.98) | 0.03(0.04) |
| | Kleindessner et al. (2023) (mean) | 10.39(0.62) | 0.031(0.018) | 76.77(1.47) | 0.04(0.03) | 69.97(1.93) | 0.0(0.0) | 72.13(1.69) | 0.05(0.03) |
| | Kleindessner et al. (2023) (0.85) | 6.97(0.41) | 0.021(0.008) | 73.17(2.03) | 0.03(0.03) | 69.97(1.93) | 0.0(0.0) | 70.7(2.65) | 0.02(0.02) |
| | Kleindessner et al. (2023) (0.5) | 4.66(0.22) | 0.017(0.013) | 71.6(2.84) | 0.02(0.02) | 69.97(1.93) | 0.0(0.0) | 71.17(2.95) | 0.01(0.01) |
| | Kleindessner et al. (2023) (kernel) | N/A | N/A | 69.8(1.21) | 0.0(0.0) | 69.8(1.21) | 0.0(0.0) | 69.97(1.93) | 0.0(0.0) |
| 2 | Ravfogel et al. (2020) | 3.25(0.38) | 0.007(0.004) | 71.33(2.32) | 0.02(0.02) | 69.97(1.93) | 0.0(0.0) | 70.13(2.03) | 0.01(0.01) |
| | Ravfogel et al. (2022a) | 3.19(0.36) | 0.04(0.024) | 71.37(2.5) | 0.03(0.04) | 69.97(1.93) | 0.0(0.0) | 70.47(2.06) | 0.02(0.03) |
| | Samadi et al. (2018) | N/A | N/A | 74.2(2.15) | 0.06(0.04) | 69.93(1.99) | 0.0(0.01) | 76.57(2.6) | 0.08(0.06) |
| | **Ours** (offline, mean) | 10.39(0.62) | 0.031(0.018) | 76.77(1.47) | 0.04(0.03) | 69.97(1.93) | 0.0(0.0) | 71.8(1.59) | 0.06(0.05) |
| | **Ours** (FNPM, mean) | 10.39(0.62) | 0.031(0.018) | 76.7(1.43) | 0.04(0.03) | 69.97(1.93) | 0.0(0.0) | 72.63(1.78) | 0.08(0.05) |
| | **Ours** (offline, $m$=10) | 6.36(0.51) | 0.017(0.008) | 72.6(2.37) | 0.03(0.02) | 69.97(1.93) | 0.0(0.0) | 71.2(2.14) | 0.02(0.03) |
| | **Ours** (FNPM, $m$=10) | 6.55(0.44) | 0.019(0.01) | 73.0(2.36) | 0.02(0.02) | 69.97(1.93) | 0.0(0.0) | 70.83(2.25) | 0.02(0.02) |
| | **Ours** (offline, $m$=25) | 4.89(0.24) | 0.032(0.018) | 72.77(2.57) | 0.04(0.04) | 69.97(1.93) | 0.0(0.0) | 71.27(2.72) | 0.03(0.04) |
| | **Ours** (FNPM, $m$=25) | 4.91(0.29) | 0.028(0.018) | 72.53(2.37) | 0.03(0.02) | 69.97(1.93) | 0.0(0.0) | 70.9(2.33) | 0.02(0.02) |
| | PCA | 38.19(0.85) | 0.137(0.012) | 99.93(0.13) | 0.09(0.06) | 74.43(0.68) | 0.14(0.14) | 95.7(1.93) | 0.11(0.07) |
| | Olfat and Aswani (2019) (0.1) | 29.1(0.98) | 0.022(0.006) | 99.97(0.1) | 0.1(0.06) | 71.43(2.47) | 0.03(0.04) | 0.0(0.0) | 0.0(0.0) |
| | Olfat and Aswani (2019) (0.0) | 29.0(0.98) | 0.021(0.005) | 99.97(0.1) | 0.1(0.06) | 71.3(2.44) | 0.02(0.04) | 0.0(0.0) | 0.0(0.0) |
| | Lee et al. (2022) (1e-3) | 33.04(1.11) | 0.022(0.006) | 100.0(0.0) | 0.1(0.06) | 74.3(1.66) | 0.09(0.02) | 95.9(1.67) | 0.09(0.07) |
| | Lee et al. (2022) (1e-6) | 16.6(1.08) | 0.014(0.003) | 94.43(1.55) | 0.07(0.06) | 69.97(1.93) | 0.0(0.0) | 83.1(2.93) | 0.06(0.06) |
| | Kleindessner et al. (2023) (mean) | 35.72(0.83) | 0.024(0.004) | 99.97(0.1) | 0.1(0.06) | 74.1(2.09) | 0.09(0.05) | 96.77(2.16) | 0.09(0.06) |
| | Kleindessner et al. (2023) (0.85) | 27.69(0.82) | 0.02(0.004) | 100.0(0.0) | 0.1(0.06) | 72.97(3.06) | 0.05(0.05) | 97.8(1.29) | 0.09(0.06) |
| | Kleindessner et al. (2023) (0.5) | 19.87(0.61) | 0.016(0.004) | 99.9(0.15) | 0.1(0.06) | 70.83(2.87) | 0.01(0.02) | 94.67(2.11) | 0.08(0.05) |
| | Kleindessner et al. (2023) (kernel) | N/A | N/A | 70.1(1.18) | 0.0(0.01) | 69.8(1.21) | 0.0(0.0) | 81.7(5.49) | 0.08(0.05) |
| 10 | Ravfogel et al. (2020) | 15.0(0.88) | 0.01(0.002) | 99.1(0.47) | 0.1(0.06) | 70.7(2.39) | 0.01(0.02) | 93.27(2.32) | 0.07(0.06) |
| | Ravfogel et al. (2022a) | 16.41(1.1) | 0.022(0.011) | 99.9(0.15) | 0.1(0.06) | 71.8(3.08) | 0.03(0.04) | 95.9(2.3) | 0.09(0.04) |
| | Samadi et al. (2018) | N/A | N/A | 99.07(0.57) | 0.1(0.06) | 75.87(0.92) | 0.11(0.1) | 98.8(0.62) | 0.09(0.06) |
| | **Ours** (offline, mean) | 35.72(0.83) | 0.024(0.004) | 99.97(0.1) | 0.1(0.06) | 74.1(2.09) | 0.09(0.05) | 97.5(0.85) | 0.1(0.07) |
| | **Ours** (FNPM, mean) | 35.73(0.84) | 0.024(0.004) | 99.97(0.1) | 0.1(0.06) | 74.07(2.12) | 0.09(0.05) | 95.53(1.46) | 0.1(0.07) |
| | **Ours** (offline, $m$=10) | 26.24(0.94) | 0.018(0.004) | 99.9(0.21) | 0.1(0.06) | 72.83(3.07) | 0.04(0.04) | 96.73(1.21) | 0.1(0.07) |
| | **Ours** (FNPM, $m$=10) | 26.23(0.81) | 0.018(0.004) | 99.9(0.21) | 0.1(0.06) | 71.37(3.41) | 0.02(0.04) | 96.4(1.33) | 0.08(0.06) |
| | **Ours** (offline, $m$=25) | 21.1(0.48) | 0.027(0.005) | 99.87(0.22) | 0.1(0.06) | 71.77(3.56) | 0.02(0.03) | 93.5(3.54) | 0.08(0.04) |
| | **Ours** (FNPM, $m$=25) | 21.06(0.52) | 0.026(0.005) | 99.9(0.15) | 0.1(0.06) | 71.77(3.41) | 0.03(0.04) | 94.97(2.55) | 0.08(0.05) |

