# OpenReview forum: "Fair Streaming Principal Component Analysis: Statistical and Algorithmic Viewpoint"
_NeurIPS.cc/2023/Conference — NeurIPS 2023 poster_

### Official Review · Reviewer_GuCQ · 2023-07-06

**Soundness:** 3 good
**Presentation:** 3 good
**Contribution:** 3 good
**Rating:** 6
**Confidence:** 3

**Summary:**

This paper studies fair PCA problem. The authors provide a novel formulation of fair PCA based on the "Null it Out" approach and propose the corresponding criterion called PAFO-learnability. The authors also present a streaming algorithm for fair PCA, which has low memory complexity. Experimental results verify the scalability of the proposed method.

**Strengths:**

1. The paper is well-written and mostly clear.
2. The studied problem is interesting and important. Streaming algorithms are very useful in the limited-memory setting.

**Weaknesses:**

1. Assumption 6.1 is a bit strong to me. The paper may show that when D_s is a common distribution such as sub-Gaussian distribution, the generated data satisfies assumption 6.1.
2. I'm worried about the tightness of theorems in section 6. The theoretical results show that Alg 1 and Alg 2 may require very large block sizes, which reduces the contirbution of saving memory.
3. The experiments part may present some quantitative results rather than just show the images. The expriments should verify that proposed FNPM algorithm is PAFO-learnability.

**Questions:**

1. Why this paper proposes a new formulation of fair PCA rather than use a existing one? What's the advantage of "Null it out" formulation?
2. Do existing fair PCA algorithms satisfy PAFO-Learnability?

**Limitations:**

The paper has no potential negative societal impact.

---

> ### Author Rebuttal · Authors · 2023-08-09
>
> We thank the reviewer’s valuable review and questions. Here, we respond to each point raised by the reviewer:
>
> > **W1. Strong assumption**
>
> Please refer to our general response.
>
>
> > **W2. Tightness of the theorems & large block size**
>
> We understand your concern about the tightness of the theorems, particularly as they assert that one needs a large block size. However, we want to emphasize that **the memory complexity is independent of the size of the blocks.** Indeed, our algorithm only requires O(d max(m, k)) *storage* as all other computations are handled in a running-average-type manner, with the primary computational cost being matrix-vector multiplications.
>
> It is important to note that the dependencies on epsilon, delta, and the singular value gap in our results align with those found in previous works on the (vanilla, non-fair) noisy power method [8,9], which also relied on Bernstein concentrations. In practice, using large block sizes helps mitigate large variances and is a common theoretical and empirical approach for the noisy power method.
>
>
> > **W3. Experiments verifying that the FNPM algorithm is PAFO-learnable**
>
> Thank you for suggesting experiments to verify the PAFO-learnability of our algorithm. We have done two sets of additional experiments that would address this point. **Due to space constraints, we report only partial results, but we emphasize that we will report the full results in our revised manuscript.**
>
> First, we conducted experiments on UCI datasets (COMPAS, German Credit, & Adult Income) to compare our algorithm's *quantitative* performance against previous approaches. We evaluated our approach based on metrics such as explained variance, distributional fairness (measured by MMD distance), downstream task accuracy, and downstream task fairness (measured by demographic parity, or DP). The experimental protocols were adopted from prior fair PCA literature [2,3] to ensure a fair comparison. The results demonstrate that our alternative formulation of fair PCA and its streaming variant exhibit comparable performance in both runtime and overall performance.
>
> Second, we conducted synthetic experiments varying the block sizes (and thus the overall sample complexity). With a fixed confidence level ($1-\delta = 0.9$), we reported the maximum possible resulting $\varepsilon_1$ (for optimality) and $\varepsilon_2$ (for fairness constraint). To disentangle the effect of these two error terms, we fix one of either $\boldsymbol{V}$ or $\boldsymbol{N}$ and train the other one with our NPM-based algorithm. As confirmed by Figure 1 in the additional supplementary pdf file, the blocks of sizes $\approx O(\epsilon_1^{-2} + \epsilon_2^{-2})$ are sufficient to achieve PAFO-learnability with errors $\varepsilon_1$ and $\varepsilon_2$ with probability $0.9$. Our obtained sample complexity dependencies seem to match the experiments.
>
> > **Q1. Regarding the significance of our new formulation**
>
> Please refer to our global response.
>
>
> > **Q2. Do existing fair PCA algorithms satisfy PAFO-Learnability?**
>
> We appreciate your comment and bringing up this interesting and important point. We agree that investigating whether existing algorithms satisfy PAFO-learnability is an interesting future topic, but it is a nontrivial task. However, we want to reiterate that none of the existing fair PCA algorithms satisfy PAFO-learnability for *streaming* PCA (Definition 5.1), where the crucial part is whether the memory limitation can be satisfied; this is discussed in Section 5.2.
>
> Based on our understanding, we suspect that the fair PCA algorithms proposed in [2,3] may be PAFO-learnable in the *non-streaming* setting (with $\Omega(d^2)$ memory). The reason for excluding [1] (SDP-based approach) is because, empirically, the algorithm proposed by [1] consistently yielded significantly sub-optimal performance regarding explained variance and downstream task accuracy compared to other fair PCA algorithms [2,3]. For further details and discussions, please refer to the experimental sections of [2,3].
>
> In conclusion, we believe that our current approach does indeed address the memory limitation, even with a large block size (which is unavoidable for noisy power method-type algorithms), and that our theoretical guarantees will hold even with a relaxed assumption. We are open to addressing any further questions or concerns the reviewer may have. We hope that our response, including the additional experiments verifying PAFO-learnability, has properly addressed the reviewer’s concerns, and we hope that the reviewer would reconsider the score. Thank you again for your insightful reviews and comments.

---

> > ### Comment · Reviewer_GuCQ · 2023-08-15
> > **Response**
> >
> > Thank you for the rebuttal, which address part of my concerns. However, I still have some additional questions:
> > 1. It is not clear to me whether Alg.1 is online or offline.
> > 2. The iteration number U in Alg.1 does not seems to appear in the theorems in section 6. How does it affect the performance of the algorithm?

---

> > > ### Author Response · Authors · 2023-08-17
> > >
> > > We thank the reviewer for their attention to our work and the insightful questions they have raised. Below, we provide responses to each of the questions.
> > >
> > > > **Question 1: It is not clear whether Alg. 1 is online or offline.**
> > >
> > > Our Alg.1 is an online algorithm that takes a data point one by one, and performs learning in $O(dm)$ space complexity to output $N$, an approximate unfair subspace (see Section 3.2). Although we have shortened our pseudocode due to space constraints, the main estimators used in Alg. 1 (see Eqn. (5) in our draft) are updated in an online manner using only vector-vector additions and vector-matrix multiplications. We included the complete pseudocode of Alg.1 in Appendix B, where the reviewer can find a more clarified version of our Alg. 1.
> > >
> > > In the offline setting, as we can fully compute $\mathbf{Q}$ and $\mathbf{f}$ from the given offline data, the unfair subspace $\mathbf{N}$ can be computed via SVD of $[\mathbf{Q} | \mathbf{f}]$, as discussed in Section 3.2. Depending on the problem setting, one can still transform an offline setting into an online setting by going through the data points one by one. This would help alleviate memory limitations or other issues, as we’ve done in our CelebA dataset experiment.
> > >
> > > > **Question 2: Effect of the iteration number $U$ on the theorems in Section 6 and the performance of Alg. 1**
> > >
> > > Thank you for pointing this out. Our final sample complexities (Theorem 6.3) are derived by multiplying the iteration number by the batch size for each phase, which is why there isn’t an explicit mention of the iteration number $U$ in the theorem statements; for the proof we have chosen a suitable $U$, and it is included in our Theorem 6.3. Precisely, Theorem 6.1 and 6.2 characterize the sufficient block size for ensuring small noise terms in the noisy power iterations, and Lemma 6.1 (which is taken from [8]) universally characterizes the iteration number that ensures a small final error, given that the iteration errors are small.
> > >
> > > Let us further elaborate on the effect of the block size $b$ and the iteration number $U$ on the convergence rate of the noisy power method. With a closer look at the convergence result by Hardt & Price [8], especially their Lemma 2.2 and Theorem 2.3 (Lemma 2.3 and Theorem 2.4 of their arXiv version, resp.), the distance between the noisy power method iterates and the ground truth decays roughly as $\varepsilon + C^U$, where $\varepsilon$ scales inversely with the square root of the block size $\sqrt{b}$, $U$ is the iteration number, and $C$ is a problem-dependent quantity that depends on the singular value gap and $\varepsilon$. Thus, for a fixed block size $b$, our choice of $U$ is a minimal choice (and thus “tight”) such that the second term becomes negligible compared to the first term, resulting in the error $\varepsilon \sim \frac{1}{\sqrt{b}}$. In other words, even though $U$ is increased far beyond our choice, the final error will still be $\varepsilon$, i.e., a much higher number of iterations does not lead to an error less than $\varepsilon$.
> > >
> > > Again, we emphasize that our choice of $U$ is *sufficient* to ensure the final error is small. In practice, a lesser iteration number may be sufficient for good performance. Indeed, for our CelebA dataset experiments and additional UCI/synthetic dataset experiments, we have observed that a moderate number of iterations (10~20) is enough.
> > >
> > > For completeness, lastly, we provide here the precise form for the iteration number in Alg. 1 (For the notation, please refer to Assumption 6.2.):
> > > $$ U = O\left(\frac{\nu_m}{\nu_m-\nu_{m+1}}\log\frac{d}{\epsilon\delta}\right) = O\left(\frac{K_{m,\nu}}{\Delta_{m,\nu}}\log\frac{d}{\epsilon\delta}\right) $$
> > >
> > > We hope these responses resolve the reviewer’s concerns, and we are happy to answer any more questions or concerns that the reviewer may have.

---

> > > > ### Comment · Reviewer_GuCQ · 2023-08-19
> > > >
> > > > Thank you for the response which addresses my concerns. I have raised my score from 4 to 6. I think the complete pseudocode of Alg.1 should be presented in the main paper (at least the paper should indicate the full version is in appendix B). Also, I hope the author can add the discussion on the iteration number U to the revision.

---

> > > > > ### Author Response · Authors · 2023-08-19
> > > > >
> > > > > Thank you for your constructive feedbacks and for revising the score. We will ensure that the discussion and clarification you suggested are reflected in our final manuscript.

---

### Official Review · Reviewer_6AMc · 2023-07-07

**Soundness:** 3 good
**Presentation:** 3 good
**Contribution:** 3 good
**Rating:** 6
**Confidence:** 4

**Summary:**

This paper proposes a new approach for Fair PCA algorithms that is scalable and fair at the same time. The main contributions of the paper are as follows:

- A new formulation of fair PCA based on the "Null It Out" approach. The goal is to maximize explained variance while nullifying the subspace spanned by the mean difference and leading eigenvectors of the covariance difference between groups. This formulation leads to a closed-form solution and avoids infeasibility issues in previous covariance matching approaches. Their approach removes the unfair subspaces using a noisy power method.

- A new notion of learnability for fair PCA called Probably Approximately Fair and Optimal (PAFO)-learnability. This provides a statistical framework to analyze some of the fair PCA algorithms.

- A new setting called fair streaming PCA, which addresses practical memory limitations. The authors propose an algorithm called Fair Noisy Power Method (FNPM) which only requires O(dk) memory, where d is data dimension and k is the target PCA dimension.

- The empirical study of this method using a vision task is intuitive.

**Strengths:**

I believe the strengths of this paper can be summarized in three main key points:

- Theoretical rigor. The paper provides the first statistical framework for analyzing fair PCA in terms of PAFO-learnability. This gives theoretical guarantees on the solution quality of algorithms like FNPM. Previous works mainly lacked such a framework.

- They propose  FNPM for this problem, which is quite simple to implement, building on standard tools like cumulative averaging and the noisy power method. This makes it easy to apply in practice.

- Their approach is scalable and intuitive. They have validated that with a vision task to demonstrate the scalability. Also, the formulation based on nullifying the "unfair" subspace gives flexibility in how much fairness to impose by choosing m. This can be tuned based on the use case. The analysis provides insights into how properties like the singular value gaps and mean difference norm affect the sample complexity and solution quality.

**Weaknesses:**

There are some main concerns I have regarding this paper:

- The results are limited to binary sensitive attributes and two groups. Extending the approach to handle more complex, multi-group scenarios with sensitive feature interactions would strengthen the paper. Some of the main previous approaches can easily handle this problem as well [A,B].

- Certain assumptions made, like those in Assumptions 6.1 and 6.2, are quite strong and restrictive. Relaxing these assumptions, or testing how sensitive the approach is to their violation would improve the robustness.

- The experiments focus on how much FNPM removes features visually related to sensitive attributes. Evaluating the fairness of solutions in a more quantitative, metric-based fashion would provide a more objective assessment. This quantity is defined differently in various approaches.

- Lack of comparison with the other branch of fair PCA methods. Although the goal of equalizing losses is different than what presented here, it would be more beneficial to better understand how different methods would compare.

- Comparing the effects of this fair PCA on downstream tasks like classification can be beneficial to better understand the effects of a fair PCA approach.



[A] Morgenstern, Jamie, et al. "Fair dimensionality reduction and iterative rounding for sdps." arXiv 2019 (2019).

[B] Kamani, Mohammad Mahdi, et al. "Efficient fair principal component analysis." Machine Learning (2022): 1-32.

**Questions:**

See the previous part

**Limitations:**

Some of the limitations I discussed in the weakness section are not clearly discussed in the paper.

---

> ### Author Rebuttal · Authors · 2023-08-09
>
> Thank you for your detailed review, for recognizing the significance of our contributions, and for providing valuable feedback. Let us address the raised points below:
>
> > **W1. Dealing with multiple sensitive groups**
>
> We appreciate the reviewer for bringing up the issue of focusing on two sensitive groups in our paper. While we agree that the simplicity of the discussions led us to consider two groups initially, we acknowledge the importance of extending our work to handle multiple sensitive groups. We agree with the reviewer that this direction would significantly enhance the strength and impact of our paper.
>
> We first emphasize that the definition of fair PCA considered in references [A, B] mentioned by the reviewer is entirely different from ours as their definition is such that the reconstruction losses across groups are equalized, while ours is in terms of fair representation learning. Thus, their approaches are not readily applicable to our definition of fair PCA as a straightforward extension to multiple groups. Among the ones that tackle fair PCA from the fair representation learning perspective, [1,3] explicitly consider dealing with multiple groups in the fair PCA (fair representation). However, the approach of [1] is not scalable to higher dimensions (as also discussed in [2,3]), and [3]’s approach only deals with mean-matching, is not memory-efficient, and has no theoretical guarantees.
>
> For completeness, we outline how our formulation can be extended to handle multiple sensitive groups. In the streaming setting, we propose to sample the sensitive attribute from a multinomial distribution over the $G$ sensitive groups, where each group corresponds to a separate data distribution. The fair PCA formulation would then involve nullifying the projected mean differences and the top-m eigenvectors of the projected covariance differences for all possible pairwise group comparisons.
>
> To address theoretical analyses, we assume that $G = O(1)$ (or even up to $G = o(\sqrt{d/m})$, as further explained later). For the mean difference, we construct a $d \times (G-1)$ matrix whose $j$-th column is the (estimated) mean difference between group $j$ and $j+1$. For the covariance difference, we construct $\binom{G}{2}$ number of $d \times m$ matrices, each corresponding to top-m eigenvectors of the (estimated) covariance difference between two groups for all possible pairwise comparisons. These matrices are to be nullified, forming a $d \times O(mG^2)$ matrix $\boldsymbol{N}$ used in Phase 2. Estimating $\boldsymbol{N}$ in Phase 1 can be achieved in $o(d^2)$ space, even when $G= o(\sqrt{d/m})$ as mentioned at the beginning. With minor adjustments, we believe our algorithm can effectively satisfy PAFO-learnability for multiple groups with similar assumptions.
>
>
> > **W2. Strong Assumptions**
>
> Please refer to the general response 2.
>
> > **W3-5. Lack of quantitative evaluation, downstream tasks, and comparison with the other branch of fair PCA**
>
> We appreciate the reviewer's suggestion and acknowledge the importance of quantitative evaluation in providing an objective assessment. As showcased in **Table 1** of our supplementary attachment, we have already conducted experiments on UCI datasets (German Credit, Adult Income, COMPAS) to compare the performance of our algorithm against previous approaches. We evaluated our approach based on metrics such as explained variance, PCA fairness (measured by MMD distance), downstream task accuracy, and downstream task fairness (measured by demographic parity). The experimental protocols were adopted from prior fair PCA literature [2,3] to ensure a fair comparison.
>
> Additionally, we performed additional experiments comparing our approach with the algorithm presented in reference [11], which focuses on equalizing reconstruction loss across groups. The results (shown in Table 1) demonstrate that our alternative formulation of fair PCA and its streaming variant exhibit comparable performance. Moreover, as already reported in [2,3], in general, the fair PCA algorithms that focus on fair representation learning, including ours, outperform [11] in both PCA fairness (in the context of fair representation) and downstream task fairness. For further information, we refer the reviewer to Appendix A of [2], where the conceptual difference between the two different fair PCA formulations is well explained.

---

> > ### Comment · Reviewer_6AMc · 2023-08-19
> >
> > Thanks to the authors for the responses. I will keep my score.

---

> > > ### Author Response · Authors · 2023-08-19
> > >
> > > We are also grateful for your thorough examination of our response. If you have any remaining questions or additional remarks, please let us know without hesitation.

---

### Official Review · Reviewer_VcC6 · 2023-07-10

**Soundness:** 3 good
**Presentation:** 3 good
**Contribution:** 3 good
**Rating:** 7
**Confidence:** 2

**Summary:**

This research paper focuses on the fair principal component analysis (PCA) problem using streaming data while requiring low memory. The authors introduce a new formulation for fair PCA, which involves optimizing the vanilla PCA objective with a linear "fair" constraint. In the oracle setting, where the true parameter is known, the problem has a closed-form solution. The authors define the concept of "PAFO-learnable" to quantify the sample complexity of learning a semi-orthogonal matrix V, which is an approximately optimal solution to the oracle problem. They present streaming algorithms and prove that such an algorithm scheme has finite sample complexity according to the proposed PAFO-learnable notion. The main idea is to utilize the noisy power method framework to estimate the unfair subspace (the linear constraint) and subsequently employ this estimation in the NPM (Noisy Power Method) to estimate fair PCA. The effectiveness of the proposed method is evaluated using real-world data.

**Strengths:**

Overall, the contribution of the paper is well-motivated and aligns with the ongoing development of methods for problems involving fair constraints. The paper is well-written, and the investigation is quite extensive. However, I must admit that I am not familiar with recent developments in fairness in machine learning and cannot provide an assessment of the significance of the new formulation Equation (1) and the concept of learnability (Definition 4.2), although they appear reasonable and interesting.


**Weaknesses:**

I have a few minor comments:

* L68: It might be more appropriate to use the term "semi-orthogonal matrix" instead of "orthogonal" to distinguish between O(d) and St(d,k).
* L69, L116, L171: QR decomposition typically yields two outputs, namely the (semi-)orthogonal part and the upper triangular part.
* L164: Regarding F_d, it differs from the one defined in Definition 4.2, where learnability is defined for a different quantity denoted as F_d.
* L262: Once again, F_{d,m,k} is inconsistent with the definition provided in Definition 4.2.

**Questions:**

See above.

**Limitations:**

Yes.

---

> ### Author Rebuttal · Authors · 2023-08-09
>
> We thank the reviewer for providing a detailed and insightful review of our paper. We are glad that the reviewer recognized the significance of our work and appreciated our contributions. We provide our answers below.
>
> > **S1. Recent developments in fairness in ML**
>
> The field of fairness has seen considerable progress in recent years in proposing sensible and new definitions of fairness and new algorithms with the fairness constraint. However, scalability remains a crucial challenge in the latter part (so-called algorithmic fairness), particularly for fair PCA. Indeed only recently has the issue of scalability begun to be studied, focusing on fair clustering [14,15,16]. One of our primary focuses was addressing the memory limitation associated with fair PCA by proposing its streaming variant. Especially as PCA is one of the standard tools used for high dimensional data analysis (see our Introduction section) and previous approaches to fair PCA [1,2,3] are still not so scalable (see our Experiments section), we believe that making fair PCA further scalable is timely and important.
>
> > **S2. Regarding the significance of our new formulation Equation (1) and the concept of learnability (Definition 4.2)**
>
> Again, we appreciate the reviewer's keen interest in our paper. We would like to take a moment to delve deeper into the significance of our work. Our alternative formulation of fair PCA offers two crucial advantages: **feasibility** and **scalability**. To begin, our formulation of fair PCA as a constrained optimization problem is always feasible so that there is no need for further relaxation, which is sometimes not the case for the previous formulations under certain group-wise distributions [1,2,3]. This, in turn, has paved the way for the rigorous establishment of the novel concept of statistical sample complexity in terms of PAFO-learnability. Indeed, this achievement stands as a first in the field of fair PCA literature. Moreover, our formulation is scalable, particularly in the context of streaming scenarios, allowing us to develop the FNPM algorithm for fair streaming PCA. For a more detailed elaboration on this, we kindly direct the reviewer to our general response.
>
> > **W1. Minor comments on the writing**
>
> We genuinely appreciate the reviewer's comments on the writing and assure you that we will incorporate all the suggested fixes in our revised manuscript. To be more specific, we will reflect the following points:
> * L68: We will clarify by using the term ‘semi-orthogonal’ rectangular matrices, as per the reviewer’s suggestion.
> * QR decomposition (L69, L116, L171): We would like to remark that the symbol “$QR(\cdot)$” itself is quite widely used in (streaming) PCA literature, which is generally used as an orthogonal projection operator onto the Stiefel manifold (for the case of vectors (i.e., $k=1$), $QR(v)$ is defined to be the same as the normalization $v/\|v\|$, an orthogonal projection onto the unit sphere). Nevertheless, we will elaborate more clearly on this in our revised manuscript.
> * L164 & L262: We greatly appreciate the reviewer’s comment on our notation. We correct the notation such that for fixed integers $d, k, m$, we define PAFO-learnability for a collection $\mathcal{F}_d \subset \mathcal{P}_d \times \mathcal{P}_d \times (0,1)$. We will make this clear in our upcoming manuscript.

---

### Official Review · Reviewer_SSmx · 2023-07-10

**Soundness:** 2 fair
**Presentation:** 2 fair
**Contribution:** 2 fair
**Rating:** 4
**Confidence:** 4

**Summary:**

The paper defines a new notion for fair pca. It is assumed that the data comes from  mixture of (two) distributions and the goal is to find a subspace for it such that the solution subspace is perpendicular to

1) the difference in the difference vector of the mean
2) the top m eigen vectors of the difference of the variance

This problem can simply be computed by computing PCA on a projected space. This paper looks for an algorithm that is

1) with probability 1-delta, reports an approximate solution.
2) Works in the streaming setting where once gets samples from the mixture of the distributions and uses space which depends on O(kd).

The algorithm is to use the samples to estimate the parameters of the two distributions and thus approximating the orthogonal constraint and then run the standard SVD algorithms.

**Strengths:**

- The paper defines a new notion for fair PCA.

**Weaknesses:**

- The paper does not discuss why this particular notion captures fairness.
- The amount of technical novelty in the paper is limited.


**Questions:**

NA

---

> ### Author Rebuttal · Authors · 2023-08-09
>
> Thank you for your review and comments. We assure the reviewer in advance that all the answers and discussions provided here will be incorporated into our revised manuscript. Below, we respond to each point raised in your review:
>
> > **W1. Why does this particular notion capture fairness?**
>
> To the best of our understanding, your question can be divided into two parts: 1) ‘Why does our “Null it out” formulation of fair PCA capture fairness?’ 2) ‘Why is our PAFO learnability an appropriate statistical framework for fair PCA?’
>
> First, we emphasize that our intuitive notion of fairness in PCA, where the projected distributions are approximately matched, has already been well studied [1,2,3]. Including our paper, this line of work builds upon the foundation of fair representation learning [4], a seminal work in fairness that recently received the test-of-the-time award at ICML 2023. The main idea is to learn a low-dimensional representation that retains as much information as possible about the high-dimensional data while ensuring fairness, such that any vanilla downstream task learner can achieve fairness without explicit regularization. This matches the intuition that if the appropriate (conditional) distributions match, any vanilla supervised learners would be fair.
>
> Second, we elaborate on why our proposed PAFO learnability aligns well with fair PCA. We emphasize that no existing literature on fair PCA for fair representation learning has provided statistical frameworks to analyze the problem setting. Our new formulation defines optimality and fairness criteria. Optimality refers to how suboptimal our solution's explained variance is compared to the optimal fair solution, which is always well-defined, a key characteristic of our new formulation. Fairness is defined based on how much component is left in the "unfair" directions.
>
> > **W2-1. Amount of technical novelty**
>
> We believe that our paper contributes significant technical novelty in making and analyzing scalable (memory-efficient) fair algorithms. While these points are summarized in our contributions (pg. 2) and throughout the paper, we reiterate them for clarity:
>
> > **W2-2. The novelty of our proposed new streaming setting.**
>
> The problem of fair PCA for fair representation learning has been well-studied [1,2,3]. However, previous approaches cannot handle memory limitations, where the algorithm is restricted to using only $o(d^2)$ (or $O(dk)$) space. To handle this, we introduce the new fair streaming PCA setting, where data points arrive sequentially from an "unfair" distribution, and the algorithm must learn under memory constraints. Even without fairness, such a streaming setting has received considerable interest from the stat/ML community due to its potential in processing data streams and dealing with memory limitation, especially in streaming PCA [5,6,7,8,9,10]. **We thoroughly discussed in Section 5.2 and Appendix C why existing approaches/formulations of fair PCA [1,2,3] can*not* be trivially extended to this streaming setting.**
>
> > **W2-3. The novelty of our statistical framework.**
>
> None of the previous works on fair PCA [1,2,3] had a statistical framework in which the performance of their algorithms could be rigorously shown. By statistical framework, we mean a learnability framework (similar to PAC-learning) in which the number of samples sufficient to solve the problem of fair PCA can be formalized. Indeed, due to several approximations that [1,2,3] had done to make their algorithm and/or optimization problem feasible, it is hard to see exactly which part of the approximation causes the bottleneck in the sample complexity. **As discussed in Section 3**, by considering a new “Null It Out” formulation of fair PCA, we could overcome the infeasibility issues and allow us to develop the PAFO-learnability framework.
>
> > **W2-4. Technical difficulties in the analysis.**
>
> None of the previous works on fair PCA [1,2,3] provide statistical guarantees for their algorithms. While our algorithm is based on the well-known noisy power method [8,9], **the analysis is significantly more challenging because there are two sources of randomness: group membership described by Bernoulli random variables and sampling from a group-wise distribution.** We had to modify the given random variables to apply the existing Bernstein concentration results during Phase 1's convergence proof.
>
> > **W2-5. Relevance of our setting to real-life: Experimental results.**
>
> In Section 7, we experimentally demonstrate the significance of memory limitations when performing a fair PCA on real-world datasets (full resolution, full colored CelebA dataset) using previously proposed algorithms; none of them could run on this dataset with our local machine. By transforming the problem into a streaming setting and applying our algorithm, we show that such memory limitation can be circumvented, making fair PCA *scalable*. Additional quantitative results on UCI datasets, parts of which we report in Table 1 of our attached supplementary pdf, demonstrated that our new formulation achieves similar performance as previous algorithms, showing that our formulation and algorithm are a strict improvement over the previous ones. Furthermore, the simplicity of our algorithm enhances its applicability to real-world datasets; this is in contrast to some of the previous fair PCA algorithms [1,2] as they require external libraries such as SDP solver [1] or manifold optimization package [2].
>
> In conclusion, we believe that our paper offers substantial technical novelty in both algorithm design and theoretical analysis, as well as a new problem setting of fair streaming PCA. We are open to addressing any further questions or concerns the reviewer may have. We hope that our response has properly addressed the reviewer’s concerns and that the reviewer would reconsider the score. Thank you again for your helpful reviews and comments.

---

> > ### Comment · Reviewer_SSmx · 2023-08-21
> >
> > Thank you for providing a response. I am retaining my score.

---

### Author Rebuttal · Authors · 2023-08-09

We sincerely thank all the reviewers for providing detailed and insightful reviews/comments/questions about our paper. We assure the reviewers in advance that all the answers and discussions provided here will be incorporated into our revised manuscript.
We are encouraged to see that the reviewers recognize the relevance of our newly proposed problem setting in fairness (SSmx, VcC6, 6AMc, GuCQ), *scalable yet simple* algorithm for our setting (6AMc), theoretical rigor (6AMc), clarity of our exposition (VcC6, 6AMc, GuCQ), and extensive investigation into the effectiveness of our algorithm (VcC6, 6AMc).

From now on, let us first provide our responses to three commonly raised questions:

### **Regarding the significance of our new formulation (VcC6, GuCQ)**

We start by elaborating on the justification of our new formulation of fair PCA (Eqn. (1)). Intuitively, the PCA projection that nullifies the mean difference and top eigenvectors of the covariance difference would result in an orthogonal representation from which any linear (or stronger) adversaries have difficulty in distinguishing between the sensitive groups. We acknowledge that this notion of fair PCA has been previously discussed in [1, 3], as well as recent advances in guarding protected attributes of word embeddings via the “Null It Out” approach [12,13]. Despite the existence of similar prior formulations, we clarify why we needed to propose another new formulation.

Our new formulation, compared to the prior ones, yields two main benefits. One is that our formulation is always feasible, unlike previous formulations [1,2,3], it allows us to define the notion of learnability for fair PCA rigorously. Another is that this makes the problem amenable to a memory-efficient streaming algorithm for fair PCA. On a more technical side, our formulation is quite similar to that of [3], but the key difference lies in the order in which the nullification is applied. [3] applies mean difference nullification first, then it applies the covariance nullification on the subspace resulting after the mean difference nullification. On the other hand, we apply both simultaneously. In Section 5.2 and Appendix C.2, we provide extensive discussions on why previous approaches to fair PCA [1,2,3] CANNOT be easily adapted to our memory-limited & streaming setting.

Especially for the rigorous definition of *learnability* in the context of fair PCA, we start by remarking that *no prior work has approached fair PCA from this perspective*, emphasizing the importance of establishing a solid statistical foundation. We firmly believe that defining learnability in this context will shed light on the number of samples required to achieve a desired output, providing valuable insights for researchers and practitioners in the field, as the PAC-learning framework has done since its introduction.

### **Strong assumptions (6AMc, GuCQ)**

While our assumptions may seem stringent, they are essential to highlight that no previous works on fair PCA (for fair representation learning) [1,2,3] have provided statistical guarantees. Consequently, these assumptions were indispensable in achieving rigorous results for our streaming fair PCA formulation.

Assumption 6.1 enables us to utilize the simpler version of vector/matrix Bernstein concentrations, which require bounded random variables. This was also the case in various streaming PCA literature [5,6,7] for similar reasons, where using Bernstein concentrations was necessary. Indeed, we are sure that our sample complexity results will be retained with relaxed (but similar) assumptions by using appropriate variants of the concentration inequalities (e.g., [10]).

Regarding Assumption 6.2, it is crucial to assume a singular value gap for the convergence of the noisy power method. Without this assumption, the eigenvectors lose uniqueness, hindering the overall convergence.

### **Experiments (6AMc, GuCQ)**

We have also attached a supplementary pdf containing additional experimental results that support our rebuttal. Figure 1 shows synthetic results that verify our sample complexity result w.r.t. block sizes. Also, Table 2 showcases the quantitative results of comparing several fair PCA methods on the UCI Adult Income dataset, showing the efficacy of our proposed FNPM.
Due to space constraints, we report only partial results, but we will report the full results in the upcoming revised manuscript.

Lastly, we put all the relevant references for our rebuttal below.

---
[1] Olfat & Aswani, “Convex Formulations for Fair Principal Component Analysis.” AAAI 2019.
[2] Lee et al., “Fast and Efficient MMD-based Fair PCA via Optimization over Stiefel Manifold.” AAAI 2022.
[3] Kleindessner et al., “Efficient fair PCA for fair representation learning.” AISTATS 2023.
[4] Zemel et al., “Learning Fair Representations.” ICML 2013.
[5] Bienstock et al., “Robust Streaming PCA.” NeurIPS 2022.
[6] Jain et al., “Streaming PCA: Matching Matrix Bernstein and Near-Optimal Finite Sample Guarantees for Oja’s Algorithm.” COLT 2016.
[7] Huang et al., “Streaming k-PCA: Efficient guarantees for Oja’s algorithm, beyond rank-one updates.” COLT 2021.
[8] Hardt & Price, “The Noisy Power Method: A Meta Algorithm with Applications.” NIPS 2014.
[9] ​​Balcan et al., “An Improved Gap-Dependency Analysis of the Noisy Power Method.” COLT 2016.
[10] C. Jin et al., “A Short Note on Concentration Inequalities for Random Vectors with SubGaussian Norm.” arXiv 2019.
[11] Samadi et al., “The Price of Fair PCA: One Extra Dimension.” NeurIPS 2018.
[12] Ravfogel et al., “Null It Out: Guarding Protected Attributes by Iterative Nullspace Projection.” ACL 2020.
[13] Ravfogel et al., “Linear Adversarial Concept Erasure.” ICML 2022.
[14] Backurs​​ et al., “Scalable Fair Clustering.” ICML 2019.
[15] Ziko et al., “Variational Fair Clustering.” AAAI 2021.
[16] ​​Wang et al., “Scalable Spectral Clustering with Group Fairness Constraints.” AISTATS 2023.

---

### Decision · Program_Chairs · 2023-09-21

**Decision:**

Accept (poster)

**Comment:**

The paper concerns fair PCA in the streaming setup. It received mostly positive evaluation from the reviewers, except for one borderline reject score.

Summary of the strengths:
- The paper introduces a new formulation of fair PCA based on the _Null It Out_ approach
- It provides a statistical framework for analyzing fair PCA in terms of PAFO-learnability
- It presents a streaming algorithm for fair PCA that addresses memory limitations. This algorithm, called Fair Noisy Power Method (FNPM), is thoroughly analyzed and shown to be PAFO-learnable.
- Reviewers generally found the paper well-written and clear in its presentation.

Summary of the weaknesses:
- The paper lacks a clear explanation of why the new formulation of fair PCA, based on _Null It Out_, is significant or how it captures fairness compared to existing formulations.
- Some reviewers expressed concerns about the limited technical novelty of the paper - mainly due to the fact that `offline' fair PCA turns out to be a straightforward application of the SVD. Still, developing a streaming version of fair PCA requires non trivial modifications of the original noisy power method.
- Some of the assumptions, such as Assumption 6.1, were found to be restrictive by one of the reviewers.
- Reviewers note that the experiments primarily focus on visual aspects related to sensitive attributes, while a more quantitative evaluation of fairness would provide a more objective assessment. Also, they pointed out the lack of comparison with other fair PCA methods
- Theoretical results suggesting very large block sizes.

I believe some of the weakness were resolved in the author-reviewer discussion phase, but some of the remained - in particular, the lack of motivation for the new problem formulation.

It is clear to me that there are some concerns and limitations with the paper's contribution. Still, I think overall the paper is above the acceptance bar. I hope the authors will address all the reviewers' concerns in the final version of the manuscript.